# A hazard model of subfreezing temperatures in the United Kingdom using vine copulas

Symeon Koumoutsaris

Guy Carpenter, Tower Place, London, EC3R 5BU, UK

*Correspondence to:* Symeon Koumoutsaris (symeon.koumoutsaris@guycarp.com)

**Abstract.** Extreme cold weather events, such as the winter of 1962/63, the third coldest winter ever recorded to the Central England Temperature record, or more recently the winter of 2010/11, have significant consequences for the society and economy. This paper assesses the probability of such extreme cold weather across the United Kingdom, as part of a probabilistic catastrophe model for insured losses caused by the bursting of pipes. A statistical model is developed in order to model the
extremes of the Air Freezing Index (AFI), which is a common measure of magnitude and duration of freezing temperatures. A novel approach in the modelling of the spatial dependence of the hazard has been followed which takes advantage of the vine copula methodology. The method allows to model complex dependencies especially between the tails of the AFI distributions which is important to assess the extreme behaviour of such events. The influence of North Atlantic Oscillation and of anthropogenic climate change on the frequency of UK cold winters has also been taken into account. According to the model, the
occurrence of extreme cold events, such as the 1962/63 winter, have decreased approximately two times during the course of the 20th century as a result of anthropogenic climate change. Furthermore, the model predicts that such an event is expected to become more uncommon, about twice less frequent, by the year 2030. Extreme cold spells in UK has been found to be heavily modulated by NAO, as well. A cold event is estimated to occur ≈3-4 times more likely during its negative than its positive phase. However, considerable uncertainty exists in these results, owing mainly to the short record length and the large
interannual variability of AFI.

## 1 Introduction

Extended periods of extreme cold weather can cause severe disruptions in human societies; on human health, by exacerbating previous medical conditions or due to reduction of food supply which can lead to famine and disease; agriculture, by devastating crops particularly if the freeze occurs early or late in the growing season; on infrastructure, e.g. severe disruptions in the
transport system, burst of residential or system water pipes (Bowman et al., 2012). All these consequences lead to important economic losses.

Of particular interest for the insurance industry are the economical losses that originate as a result of bursting of pipes due to freeze events. Water pipes burst because the water inside them expands as it gets close to freezing which causes an increase in pressure inside the pipe. Whether a pipe will break or not, depends on the water temperature (and consequently on the air

temperature), the freezing duration, the pipe diameter and composition, the wind chill effect (due to wind and air leakage on water pipes), and the presence of insulation (Gordon, 1996; McDonald et al., 2014).

Insurance losses from burst pipes have a significant impact on the UK insurance industry. They amount to more than £900 million in the last 10 years, representing around 10% of the total insured losses, mainly due to flood and windstorm, in the United Kingdom (UK) during the same period (ABI, 2017). Particular years can be very damaging, such as, for example, the winter of 2010/2011 where losses from burst pipes have exceeded £300 million in UK making it the peril with the largest losses that year (ABI, 2017). Moreover, much more extreme cold winters have actually occurred in the UK in the last 100 years, such as the winters of 1946/47 and 1962/63. It is crucial for the insurance business to be able to anticipate the likelihood of occurrence of similar and even more extreme events so that they can adequately prepare for their financial impact (AIR, 2012). In fact, the capital requirements in (re)insurance is estimated in a 1 in 200 year return period (RP) loss basis, which is usually much larger than the available historical records.

Probabilistic catastrophe modelling is generally agreed to be the most appropriate method to analyze such problems. The main goal of catastrophe models is to estimate the full spectrum of probability of loss for a specific insurance portfolio (i.e. comprised by several residential, auto, commercial or industrial risks). This requires the ability to extrapolate the possible losses at each risk to high return periods which is usually achieved by simulating synthetic events that are likely to happen in the near future (typically a year). More importantly, it requires to consider also how all risks relate to each other and their potential synergy to create catastrophic losses. Such spatial dependence between risks can result from various sources, for example due to the spatial structure of the hazard (e.g. the footprint in a windstorm or the catchment area in a flood event) or due to similar building vulnerabilities between risks in the same geographical area (e.g. due to common building practices) (Bonazzi et al., 2012).

Modelling the spatial dependence of the hazard is usually achieved by taking advantage of certain characteristic properties of the hazard footprint, like for example the track path and the radius of maximum wind for windstorms or the elevation in the case of floods. In the case of temperature, however, such a property cannot be easily defined; an alternative solution is to use multivariate copula models. Based on Sklar's theorem (Sklar, 1959), the joint distribution of all risk sources can be fully specified by the separate marginal distributions of the variables and by their copula, which defines the dependence structure between the variables.

However, one important difficulty is the limited choice of adequate copulas for more than two dimensions. For example, standard multivariate copula models such as the elliptical and Archimedean copulas do not allow for different dependency models between pairs of variables. Vine copulas provide a flexible solution to this problem based on a pairwise decomposition of a multivariate model into bivariate copulas. This approach is very flexible, as the bivariate copulas can be selected independently for each pair, from a wide range of parametric families, which enables modelling of a wide range of complex dependencies (Czado, 2010; Dißmann et al., 2013).

In this paper, the vine copula methodology is used in a novel application to develop a catastrophe model on insurance losses due to pipe bursts resulting from freeze events in the United Kingdom. The focus here is on the hazard component (Sect. 2) which is modeled using the Air Freezing Index (AFI), an index which takes account both the magnitude and duration

of air temperature below freezing, calculated from reanalysis data from the last 110 years. The statistical models utilized to extrapolate to longer return periods are described in Sect. 3. The model also accounts for two major drivers of climate variability in UK that are incorporated as predictors:

- the North Atlantic Oscillation (NAO), a leading pattern of weather and climate variability over the Northern Hemisphere mid-latitudes, which accounts for more than half of the year-to-year variability in winter surface temperature over UK.

- Anthropogenic climate change and its direct effects in the temperature profile in the UK.

Stochastic winter-seasons are simulated taking into account the correlation of the hazard between all pair-cells with the help of regular vine copulas (Sect. 3.3). The resulting return periods of extreme cold winters in UK, including the underlying uncertainties, are discussed in Sect. **??**. Concluding remarks are found in Sect. **??**.

## 2  Data

### 2.1  Temperature data sets

The hazard component of the catastrophe model is based on the European Centre for Medium-Range Weather Forecasts (ECMWF) twentieth century reanalysis (ERA-20C) covering the entire twentieth century from 1900 to 2010 (Poli et al., 2016). Reanalyses are data-assimilating weather models which are widely used as proxies for the true state of the atmosphere in the recent past. Even though centennial reanalyses, such as ERA-20C, represent the most convenient data sets for assessing the long-term historical climate, biases and uncertainties inherent in both raw observations and models mean that they should be used with caution. For example, important differences in the 2-meter temperature have been found between ERA-20C and other centennial reanalysis data sets, especially during the first half of the twentieth century as a result of the spare observational network in those early years (Poli et al., 2016; Donat et al., 2016). Furthermore, studies have suggested that long-term changes in the Arctic Oscillation, mean sea level pressure, and wintertime storminess seen in ERA-20C, may be spurious as a result of the assimilation of increasing numbers of observations (Dell'Aquila et al., 2016; Poli et al., 2016; Bloomfield et al., 2018).

ERA-20C product provides daily 3-hour forecast (i.e. eight forecast steps starting at 06:00UTC each day) of minimum and maximum temperature at 2 meters. These are used to compute daily minimum and maximum values at every grid cell for the entire period. The daily average temperatures are then computed as $0.5(\text{T}_{max}\text{-T}_{min})$ and the data are re-gridded to a $1°$x$1°$resolution, which corresponds to a total of 67 cells over land.

The coarse horizontal resolution is expected to have relatively small influence in most cases given that winter climate anomalies are often coherent across large parts of the UK as they are primarily associated with large-scale atmospheric circulation patterns (Scaife and Knight, 2008). Nevertheless, local temperature may be subtly different in certain micro-climates, such as upland and urban regions. In particular over urban regions, which are most important from an insurance perspective, lower resolution may lead to temperatures that are biased towards lower values, leading though to a conservative view on the severity of extreme freeze events. In upland regions, on the other hand, extreme cold temperatures are most probably underestimated,

although it is reasonable to expect that their damaging effects are somewhat mitigated from increased protection levels. For example, water pipes in properties located in mountainous regions are usually better protected against cold spells.

For comparison purposes, the observed daily average temperature gridded data set developed from the UK Met Office is also used (Perry et al., 2009). This data set is based on temperature data retrieved from 540 stations across UK with an average

station density of 21 x 21 km$^2$ (Perry and Hollis, 2005; Perry et al., 2009). It covers the entire UK, but for a much shorter period of 51 years (1960-2011). The original 5km x 5km resolution is re-gridded using bi-linear interpolation to 1°x1°in order to match the ERA-20C grid.

## 2.2   North Atlantic Oscillation Index

The NAO refers to a redistribution of atmospheric mass between the Arctic and the subtropical Atlantic and swings from

one phase to another producing large changes in weather, and in particular in surface air temperature, over the Atlantic and the adjacent continents (Hurrell et al., 2013). It is described by the NAO index (NAOI), a measure of the mean atmospheric pressure gradient between the Azores High and the Iceland Low. A positive NAOI is associated with depression systems taking a more northerly route across the Atlantic, which causes UK weather to be milder, while a negative NAOI is associated with depression systems taking a more southerly route, as a result of which UK weather tends to be colder and drier (Osborn, 2000).

In this study, the winter (December thru March) station-based index of the NAO from Hurrell (2003) is used, which is based on the difference of normalized sea level pressure between Lisbon, Portugal and Stykkisholmur/Reykjavik, Iceland (Fig. 1b).

## 2.3   Anthropogenic forcing

Increases in concentration of greenhouse gases, such as carbon dioxide (CO$_2$), are accompanied by increased radiative forcing, i.e. the difference between the incoming radiation from the sun and the outgoing radiation emitted from the Earth. This forcing

arises from the ability of the gases to absorb long wave radiation, thus reducing the amount of heat energy being lost to space, and increasing the warming of the earth's surface. Here we use the change in radiative forcing from CO$_2$ as a predictor for climate change. It is calculated using the simplified expression (Myhre et al., 1998):

$$\Delta F_{CO_2} = 5.35 ln \left( \frac{C_i}{C_{1990}} \right) \tag{1}$$

where $\Delta F_{CO_2}$ is the radiative forcing change (in W m$^{-2}$), $C_i$ is the concentration of atmospheric CO$_2$ at year $i$, and $C_{1900}$

is the reference 'pre-industrial' CO$_2$ concentration at year 1900. Consequently, a doubling of CO$_2$ corresponds to a change in the radiative forcing of 3.7 W m$^{-2}$. Historical observations of global mean CO$_2$ concentrations (in parts per million or ppm) are taken from Hansen et al. (2007). The temporal change in the CO$_2$ radiative forcing during the 20th century is shown in Fig. 1c. Projections of future CO$_2$ emissions are based on the Representative Concentration Pathway (RCP) scenarios adopted by the Intergovernmental Panel on Climate Change (IPCC) for its fifth Assessment Report (AR5) (Collins et al., 2013).

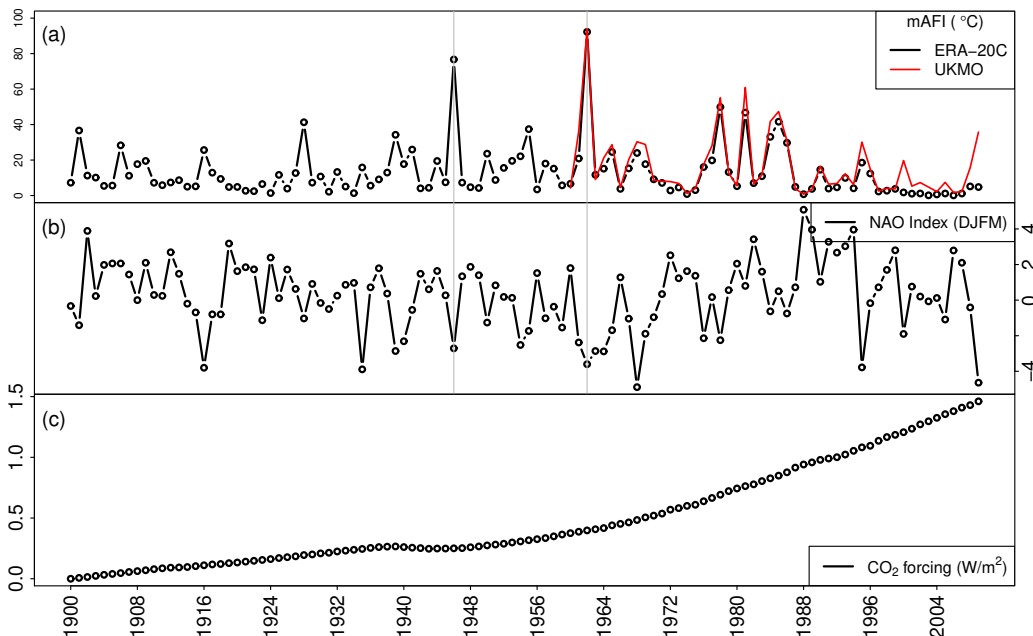

**Figure 1.** Interannual variation of (a) average AFI over UK (mAFI), (b) the North Atlantic Oscillation Index (NAOI), and (c) $CO_2$ forcing during the study period.

## 3  Methods

### 3.1  Air-Freezing Index and historical events

The daily temperature data are used to compute the AFI at each grid cell, as the sum of the absolute average daily temperatures of all days with below 0°C temperatures during the freezing period (Eq. (3)). The freezing period in this study is defined from
first of June of year to end of May of the following year, in order to include the entire winter season. Because AFI accounts both for the magnitude and duration of the freezing period, it is commonly used for determining the freezing severity of the winter season (Frauenfeld et al., 2007; Bilotta et al., 2015).

$$AFI_i = \begin{cases} \sum_{d=1}^{N} |T_d|, & \text{if } T_d < 0°C \\ 0, & \text{if } T_d \geq 0°C \text{ for all } d \end{cases} \tag{2}$$

where, $AFI_i$ is the AFI at cell $i$, N is the number of days in a winter season, $T_d$ the daily average temperature for a day $d$.
Maps of AFI values from ERA-20C for the severe winters of 1946/47, 1962/63, and 2009/10 are shown in Fig. 2. The winter of 1946/47 (i.e. season starting from 1st June 1946 to 31st of May 1947) was a harsh European winter noted for its impact in the United Kingdom. It was notable for a succession of snowstorms from late January until mid-March, mainly associated with

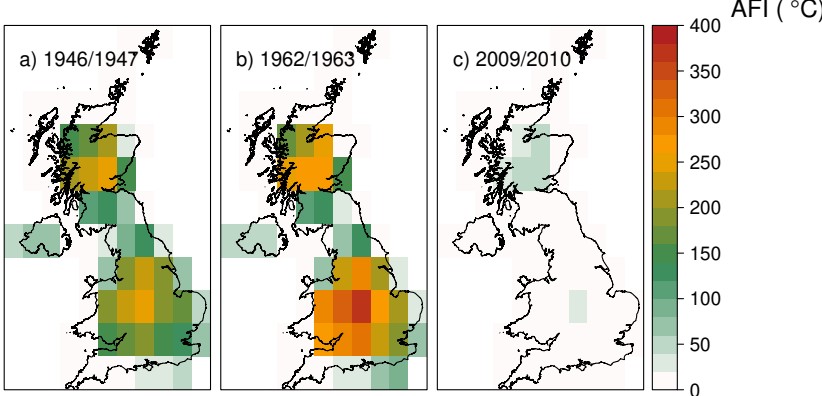

**Figure 2.** Map of AFI values (in °C) for the the winter-seasons of a) 1946/47, b) 1962/63, and c) 2009/10.

easterly airstreams (Booth, 2007). The mean AFI value (mAFI) in the entire UK (i.e. average of AFI values across all gridcells) mounted up to 75.6°C, the second largest value during the analysed period.

Based on the AFI, the 1962/1963 winter season was the most severe winter in the 20th century and one of the coldest on the record in the United Kingdom (Walsh et al., 2001). The "Big Freeze of 1962/63", as it is also known, began on the 26 of December 1962 with heavy snowfall and went on for nearly three months until March 1963. The cause of the cold conditions has been the development of a large "blocking" anticyclone over Scandinavia and north-western Russia. Easterly winds on the southern edge of this system transported cold continental air westwards, displacing the more usual mild westerly influence from the Atlantic Ocean on the British Isles. Over the Christmas period, the Scandinavian High collapsed, but a new one formed near Iceland, bringing Northerly winds. The mAFI in the entire UK (i.e. average of AFI values across all gridcells) mounted up to 90.9°C, which represents six standard deviations larger than the average of the entire 110-year period (14.0°C). The event affected more the Southern part of the country as shown in Fig. 2.

After 1962/63, a long run of mild winters followed until late 1978 and early 1979. However, temperatures in 1978/79 were not as low and the cold weather was interrupted frequently by brief periods of thaw (Cawthorne and Marchant, 1980). The mAFI value of winter 1978/79 reached 49.2°C. The 1980s stands out as a decade with several cold spells in UK, with mAFI values above 30°C for the winters 1981/82, 1984/85, and 1985/86 (46.1, 32.6, and 41.0 °C, respectively).

For the last 10 years of our study period (from 2000 to 2010), mAFI seem to be underestimated in the re-analysis data set (Fig. 1a). In particular, the winter of 2009/2010, which is well known to have brought frigid temperatures to the UK (Guirguis et al., 2011; Osborn, 2011; Seager et al., 2010; Prior and Kendon, 2011), has a mAFI value of only 4.7 °C (Fig. 2c) which is much lower than the long-term average (13 °C) and over ten times lower than mAFI value according to the UKMO dataset (59.1 °C). No clear reason is known for this bias, but it might be related to possible spurious long-term trends in the atmospheric circulation (Bloomfield et al., 2018).

As shown in Fig. 1a, the two most severe winters in the century (1946/47 and 1962/63) were associated with a negative NAO phase (Murray, 1966; Osborn, 2011). As mentioned earlier, the NAO has a profound effect on winter climate variability around

the Atlantic basin, accounting more than half of the year-to-year variability in winter surface temperature over UK (Scaife et al., 2005; Scaife and Knight, 2008). Not surprising, the ERA-20C mAFI over the entire UK is found to be significantly anti-correlated ($\rho$ = -0.49, pval=$6.510^{-8}$) with NAOI. A negative correlation is found between mAFI and $\Delta F_{CO_2}$ forcing, but it is much less significant ($\rho$ = -0.17, pval= 0.08). Both NAO and climate change effects are included in the statistical model as

predictors in order to account for their relation to cold winter spells in UK as discussed in the following section.

### 3.2   Extreme value analysis

#### 3.2.1   Stationary model

Since the historical data only extends for 110 years and our interest lies in very rare events (such as 1 in 200 years), it is necessary to extrapolate by fitting an extreme value distribution. The Generalized Extreme Value (GEV) family of distributions

has been chosen, which includes the Gumbel, the Fréchet, and Weibull distributions. An additional term was included, the probability of no hazard (P0), in order to account for the cells, mainly on the south England coast, that have years with no negative temperatures at all. The probability therefore that the AFI value ($X$) inside a cell $j$ is lower or equal than a certain value ($x$) takes the form:

$$F(x) = P(X \leq x) = P0 + (1 - P0)exp\left\{-\left(1 + \xi\frac{x - \mu}{\sigma}\right)^{-\frac{1}{\xi}}\right\} \tag{3}$$

where $\mu$, $\sigma$, and $\xi$ represent the location, scale, and shape parameters of the distribution, respectively. $F(x)$ is defined when $1 + \xi\frac{x-\mu}{\sigma} > 0$, $\mu \in \Re$, $\sigma > 0$, and $\xi \in \Re$. Its derivative, the GEV probability density function $f(x)$ is given by:

$$f(x) = f(x) = \begin{cases} P0, & \text{if } x = 0 \\ (1 - P0)\frac{1}{\sigma}\left[1 + \xi\left(\frac{x-\mu}{\sigma}\right)\right]^{-\frac{1}{\xi}-1} exp\left\{-\left[1 + \xi\left(\frac{x-\mu}{\sigma}\right)\right]^{-\frac{1}{\xi}}\right\}, & \text{if } x > 0 \end{cases} \tag{4}$$

There are various methods of parameter estimation for fitting the GEV distribution, such as least squares estimation, maximum likelihood estimation (MLE), probability weighted moments, and others. Traditional parameter estimation techniques

give equal weight to every observation in the data set. However, the focus in catastrophe modeling is mainly on the extreme outcomes and, thus, it is preferable to give more weight to the long return periods. The Tail-Weighted Maximum Likelihood Estimation (TWMLE) method developed by Kemp (2016) is employed here in order to estimate the GEV parameters. This method introduces ranking depended weights ($w_{(r)}$) in the maximum likelihood. The weights are defined for each cell based on the historical winter-season AFI values, i.e. the lowest historical AFI value in the cell (rank $r$=1 out of $n$ observations) has the lowest weight, while the largest historical AFI value (rank $r$=$n$) has the largest weight, as follows:

$$w_{(r)} = AFI_{(r)}/\sum_{r=1}^{n} AFI_{(r)} \tag{5}$$

Along with the TWMLE method described above, a second modification has been implemented in order to geographically smooth the GEV parameters. The smoothing is incorporated into the fitting process by minimizing the local (ranked) log-likelihood. More precisely, the log-likelihood at each grid cell $i$ is calculated using all grid points but weighted by their distance:

$$5 \quad LogL_i = \sum_{j=1}^{170}(k_{ij} * LogL_j) \tag{6}$$

where $k_{ij} = \frac{1}{\sqrt{2\pi}}e^{-\frac{d_{ij}^2}{2L^2}}$, $d_{ij}$ is the distance between cell $i$ and $j$, $L$ is the smoothing parameter, $LogL_j$ is the ranked log-likelihood for cell $j$.

The smoothing increases the sample size at each grid point, which thus leads to a more precise estimation of the parameters, especially for the shape parameter which is highly influential in estimating the hazard levels at high return periods. Because the data grid resolution is already coarse, a small length scale parameter L of 20 km has been used (in comparison to the grid size).

Finally, in order to avoid an over-estimate of the positive value of the shape parameter due to the small sample size (Lee et al., 2017), a modification of the maximum likelihood estimator using a penalty function is also applied for fitting the GEV. The penalty function penalizes estimates of $\xi$ that are close to, or greater than 1, following Coles and Dixon (1999).

Estimates of P0 for each grid cell are obtained by fitting a logistic regression model with intercept only (Eq. (7)). As before, the fitting is performed against all grid cells, weighted by their distance $d_{ij}$, and a length scale of 20 Km has been used. The model is extended in the non-stationary model to include covariates as described in Sect. 3.2.2.

$$ln\left(\frac{P0}{1-P0}\right) = b_0 \tag{7}$$

As an example, the GEV fit for a single cell over London is shown in Fig. 3. The curve fitted as described above (black line) is closer to the empirical estimates (black circles, computed as described in Sect. **??**) in comparison with the GEV fit with no weighting applied (grey line). As shown in Table 1, for both fits the shape parameter is positive (i.e. both fits correspond to the Fréchet distribution), but for the approach followed here (TWMLE + geographical smoothing), the shape parameter is smaller leading to a shorter tail and a curve that is nearer to the empirical estimate.

Maps of the fitted parameters are shown in Fig. 4. The probability of non-negative temperatures during a season (P0) is, as expected, larger around the coast which has milder and less variable climate due to the water influence. This also explains the lower mean (location parameter) and larger spread (scale parameter) in the AFI distributions at the coastal regions in comparison to inland. The shape parameter, which affects the skew of the distribution, shows larger values in the southern part of the UK in comparison to the north, suggesting a less rapid increase in the maximum AFI estimates.

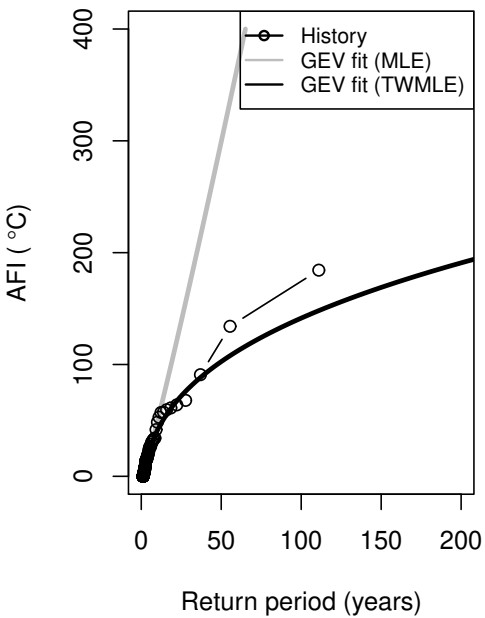

**Figure 3.** AFI return period curves for a single cell over London: empirical fit (black circles), GEV fitted with MLE (grey line), and GEV fitted with TWMLE and geographical smoothing (black line).

**Table 1.** Model parameters for a single cell over London.

| method | $b_0$ | $b_1$ | $b_2$ | $\mu$ | $\mu_0$ | $\mu_1$ | $\sigma$ | $\xi$ |
|---|---|---|---|---|---|---|---|---|
| MLE (no predictors) | -1.77 | 0 | 0 | 4.05 | 0 | 0 | 5.61 | 1.08 |
| TWMLE + geographical smoothing (no predictors) | -1.77 | 0 | 0 | 4.87 | 0 | 0 | 12.67 | 0.35 |
| TWMLE + geographical smoothing (with predictors) | -3.74 | 0.36 | 2.62 | -2.27 | -5.87 | 3.07 | 15.32 | 0.25 |

### 3.2.2 Non Stationary model

In stationary models, the distribution parameter space is assumed to be constant for the period under consideration. However, such assumption is not valid in the presence of atmospheric circulation patterns or anthropogenic changes. Regression approaches are often used to assess the influence of climatic factors on hazards and covariates such as global mean temperature, $CO_2$ concentration, and indexes of natural variability (such as NAOI) have been employed by several studies (Edwards and Challenor, 2013). In this study, a generalized linear model (GLM) is introduced into the statistical distribution

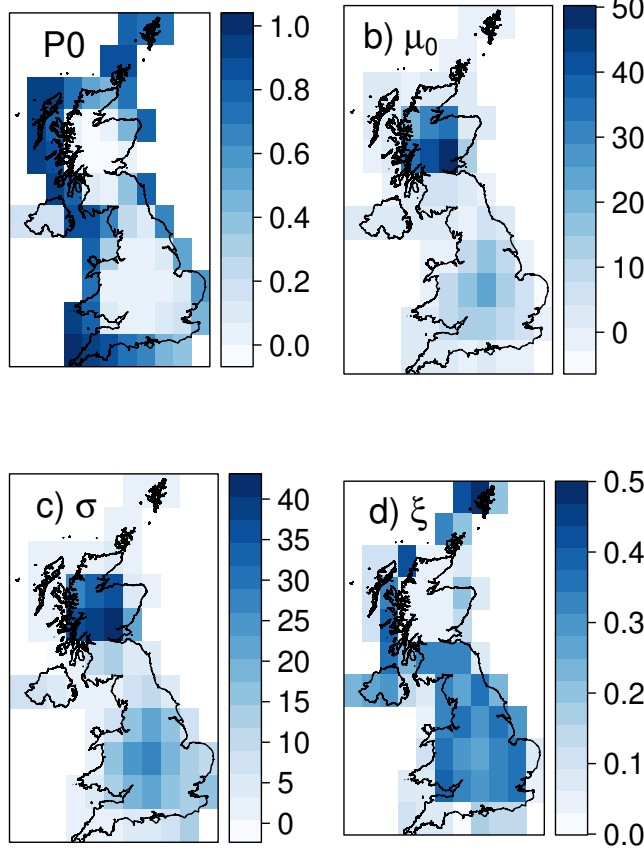

**Figure 4.** Maps showing the spatial distribution of the model fitted parameters: a) P0 calculated as $e^{b_0}/(e^{b_0}+1)$, b) location $\mu$, c) scale $\sigma$, and d) shape $\xi$.

parameter estimates in order to improve the non-stationarity representation of the model. The NAOI and the global $CO_2$ radiative forcing are chosen as covariates. There are some important caveats to this choice. First, other natural factors apart from NAO are not accounted for and hidden co-varying effects might also be present. Also, while $CO_2$ radiative forcing is linearly related to the equilibrium surface temperature, the relationship to transient surface temperatures further depends on the effi-
5   cacy of ocean heat uptake (Winton et al., 2010). Both can lead to non-linear responses of the local UK climate, especially when extrapolating far in the future.

    Despite the caveats, $CO_2$ radiative forcing and NAO have some important advantages. First both are accurately measured. Although a model that relies on global mean surface temperature may not have as strong correlations as the casual link is more indirect, it has the advantage that it does not rely on subtle regional patterns that are difficult to capture. They also provide
10  a reasonable way to isolate the human and natural influences on extreme temperatures (see for example Risser and Wehner

(2017)). While it would be possible to use a more locally defined metric (such as the change in the mean UK temperature for example), this would include more unforced naturally occurring internal variability of the climate system, making it difficult to identify the changes that are driven by anthropogenic $CO_2$ emissions. Finally, using a covariate such as the change in $CO_2$ forcing avoids the difficulty with determining the start of the trend and also results can be easily rescaled to different time period or emission scenario which is helpful for mitigation strategies.

The influence of NAO and of global warming is examined by exploring improvements to the distribution fits, after incorporating linear covariates on the distribution parameters, as follows:

– $ln\left(\frac{P0}{1-P0}\right) = b_0 + b_1 NAOI + b_2 \Delta F_{CO_2}$

– $\mu = \mu_0 + \mu_1 NAOI + \mu_2 \Delta F_{CO_2}$

where $(b_0, \mu_0)$ are the stationary model parameter estimates and $(b_1, \mu_1)$, $(b_2, \mu_2)$ are linear transformations of the covariates NAOI and $\Delta F_{CO_2}$ with respect to time, respectively.

Only non-stationarity with respect to P0 and the location parameter, $\mu$, is discussed, since modeling temporal changes in $\sigma$ and $\xi$ reliably requires long-term observations in order to be estimated accurately (Katz et al., 2002; Cheng et al., 2014). In addition, a simple linear model is selected, as this is usually preferred when searching for trends in the occurrence of extreme events (Beguería et al., 2011). Finally, even though some climate modeling studies predict changes in the nature of NAO variability in an increasing $CO_2$ climate (Rind et al., 2005; Woollings et al., 2010), the model does not include any interaction-terms, as they have been found to be non-significant.

As before, the parameters of each cell are estimated taking also into account its neighboring cells weighted by their distance. The most pertinent model is selected, for each cell, using the $\chi^2$ test, based on the change in deviance, between the null, one or two predictor model. If the significance value is less than 0.01, the model is estimated to have a significant improvement over the reduced model. A separate test is performed for the P0 and the GEV model. As an example, in the case of the London cell, the model with two predictors for both P0 and the location parameter has been chosen (table 1).

The spatial distribution of the parameters of the final model is shown in Fig. 5. Increasing NAOI or $\Delta F_{CO_2}$ are consistent with a warming trend, leading to positive values of the P0 parameters (indicating increases in the number of years with no negative temperatures) and to negative values in the location parameters (indicating lower means in the AFI distributions). The NAO is found to affect more cells in total (90%) in comparison to anthropogenic climate change (51%). Notice however that due to the internal variability of the NAO, any signal from a climate change trend can be hidden in the limited observational period.

## 3.3 Vine copula model

The stochastic behaviour of the hazard (i.e. AFI) at each cell is fully described by its corresponding GEV probability distribution, as described in Sect. 3.2. However, insurance portfolio loss analysis requires the calculation of the combined stochastic behaviour of the hazard across all the model domain (i.e. all cells). This is described by the joint distribution of the hazard which, according to Sklar's theorem, can be fully specified by the separate marginal GEV distributions and by their copula,

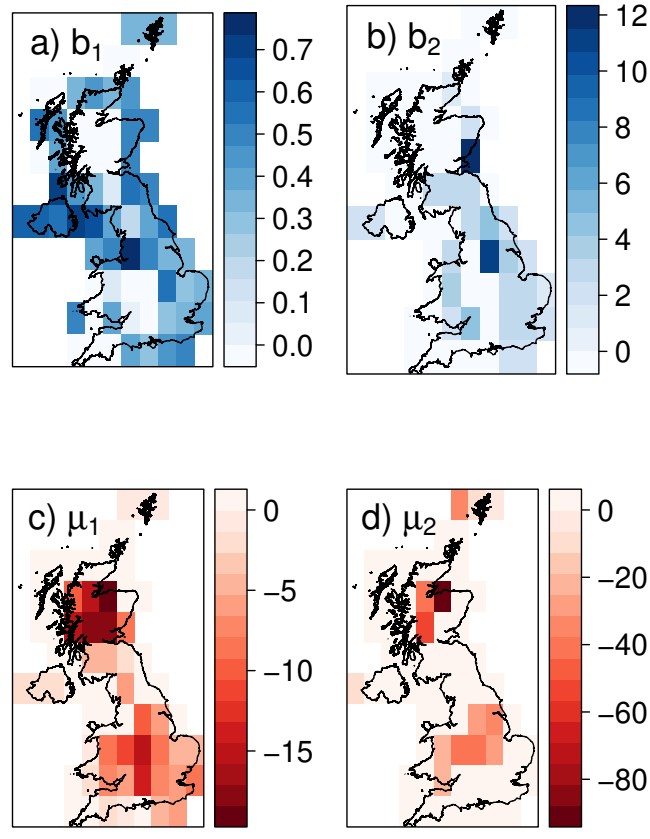

**Figure 5.** Maps showing the spatial distribution of the non-stationary model parameters: a) $b_1$, b) $b_2$, c) $\mu_1$, and d) $\mu_2$. Zero values indicate linear trends not significant at the 0.01 level.

which models the hazard dependence between the cells. Vine copulas provide a flexible solution to this problem based on a pairwise decomposition of a multivariate model into bivariate (conditional and unconditional) copulas, where each pair-copula can be chosen independently from the others. A brief introduction on the vine copula methodology can be found in Appendix **??**.

In this study, the joint multivariate hazard distribution of AFI across all the model domain (67 cells) is decomposed as a product of marginal and pair-copula pdfs. The pair-copulas are fitted using the R (https://www.r-project.org/) package VineCopula (Schepsmeier et al., 2017; Brechmann and Schepsmeier, 2013). The method follows an automatic strategy of jointly searching for an appropriate R-Vine tree structure, its pair-copula families, and estimating their parameters developed by Dißmann et al. (2013). This algorithm selects the tree structure by maximizing the empirical Kendall's $\tau$ values, based on the premise that variable pairs with high dependence should contribute significantly to the model fit and should be included in the first trees.

**Table 2.** Percentage of family types used for the first five trees of the R-Vine Model.

| Tree | Indep | Gaussian | Student t | Clayton | Gumbel | Frank | Joe | 180°Clayton | 180°Gumbel | 180°Joe | 90°Clayton | 90°Gumbel | 90°Joe | 270°Clayton | 270°Gumbel | 270°Joe |
|------|-------|----------|-----------|---------|--------|-------|-----|-------------|------------|---------|------------|-----------|--------|-------------|------------|---------|
| 1 | 0 | 3.0 | 51.5 | 0 | 34.8 | 1.5 | 1.5 | 1.5 | 0 | 6.1 | 0 | 0 | 0 | 0 | 0 | 0 |
| 2 | 9.2 | 4.6 | 36.9 | 3.1 | 6.2 | 16.9 | 1.5 | 3.1 | 1.5 | 0 | 1.5 | 4.6 | 7.7 | 0 | 0 | 3.1 |
| 3 | 25 | 1.6 | 31.2 | 4.7 | 1.6 | 7.8 | 1.6 | 0 | 1.6 | 9.4 | 6.2 | 3.1 | 1.6 | 1.6 | 0 | 3.1 |
| 4 | 27 | 6.3 | 28.6 | 4.8 | 1.6 | 9.5 | 4.8 | 3.2 | 1.6 | 4.8 | 1.6 | 1.6 | 1.6 | 0 | 1.6 | 1.6 |
| 5 | 27.4 | 8.1 | 24.2 | 4.8 | 1.6 | 9.7 | 1.6 | 6.5 | 6.5 | 1.6 | 0 | 1.6 | 1.6 | 1.6 | 0 | 3.2 |
| All | 59.2 | 3.9 | 9.4 | 2.5 | 2.1 | 8.9 | 1.4 | 2.5 | 1.0 | 1.4 | 1.6 | 0.6 | 1.3 | 1.8 | 0.8 | 1.4 |

The copula family types for each selected pair in the first tree are determined by using the Akaike information criterion (Brechmann and Schepsmeier, 2013). For computational reasons, the two-parameter Archimedean copulas are excluded from this analysis, which however has only a negligible impact in the results (not shown). The copula parameters are estimated sequentially (using maximum likelihood estimation) starting from the top tree until the last tree, as described in Czado et al. (2013). This approach only involves estimation of bivariate copulas and has been chosen since it is computationally much less demanding than joint maximum likelihood estimation of all parameters at once.

The percentage of family types used for the first few trees of the selected RVM is shown in Table 2. The large majority of the pairs in all trees are estimated to be independent (59%), but these pairs occur mainly at the higher trees, since the most important dependencies are captured in the first trees (Brechmann and Schepsmeier, 2013; Dißmann et al., 2013). Large dependencies, with Kendall's tau coefficients greater than 0.90, are found as expected between neighboring cells, but remain important across the whole model domain due to the nature of the hazard: AFI assess the freezing temperatures during the entire winter and, thus, is less associated with small scale local phenomena that can cause important spatial variation. At the first tree, 52% of the selected bivariate copulas are found to belong to the t-Student Copula and 35% to the Gumbel family, which exhibit positive dependence in the tails. Gumbel in particular has a greater dependence in the positive tail than in the negative and thus implies greater dependence at larger AFI values than at lower ones. From the third tree and onwards, the percentage of independent families is always larger than 40%.

The small sample size used (110 years of data) in conjunction with the high dimensions of the modelled pdf (67) is of concern in this study since this can lead to large uncertainties in the resulting pdf, which can also propagate in the estimated return periods. The impact of the short sample size on the uncertainties in the results is quantified using a bootstrap technique, as described in Sect. 3.4.

Goodness-of-fit (GOF) is calculated using the Cramer von Mises test, which compares the final selected RVM with the empirical copula. The RVineGofTest algorithm of the same R package implements different methods to compute the test, which however perform usually poorly in cases of small sample sizes and at higher dimensions as is the case for this work (Schepsmeier, 2013). Nevertheless, Table 3 shows the GOF results for two of these methods. The p.value is found to be larger than 0.05, which is an indication that the fitted RVM cannot be rejected at a 5% significance level. However, given also the quite

**Table 3.** Goodness-of-fit values for the Cramer von Mises statistic based on the empirical copula process (ECP) and based on the combination of probability integral transform and empirical copula process (ECP2) as implemented in the VineCopula R package.

| Method | CvM | p.val |
|--------|-----|-------|
| ECP | 9.1 | 0.7 |
| ECP2 | 0.009 | 1 |

large p.values, a Type II error cannot be excluded. Nevertheless, the suitability of the model, in comparison to the empirical data, is further discussed in the the results section as well.

In the case of the stationary model, the vine copula is employed to model the entire spatial dependence of the AFI in the UK. On the other hand, the spatial AFI structure in the case of the non-stationary model is modelled in two ways: a) by quantifying the dependence on NAO/$CO_2$ in each location, treating each location as conditionally independent, then inducing spatial dependence through the variation of NAO/$CO_2$ and b) by fitting the RVM model to all the residual dependencies associated with the AFI between the cells; these account for dependencies between cells resulting from other large-scale circulation patterns and also regional climate variability (e.g. due to effects of local orography, land-sea contrast, and small scale atmospheric features such as convective cells). Notice, however, that the effect of NAO/$CO_2$ on the residual hazard dependency structure is not taken into account here. Recently, a methodology that offers the possibility to include such meteorological predictors in a vine copula model has been developed by Bevacqua et al. (2017a); Bevacqua (2017) and is something to be addressed in a future study.

### 3.4 Stochastic simulation and uncertainty estimation via parametric bootstrap

The pdf is used to simulate 100K years of winter-seasons in the UK. For each year, the simulated AFI values at each grid cell depend on the other cells based on the fitted RVM. Long simulations are needed to obtain numerically converged results, i.e. to convergence to the "true" return period. Our focus here is the 200 year RP, which is commonly associated with capital and regulatory requirements. By repeating the simulation several times, it has been assessed that 100K years of winter seasons is long enough to neglect the Monte Carlo uncertainty. The stationary model is used to generate a stochastic set which corresponds to the current hazard experience. The non-stationary model permits us to create additional stochastic sets that represent different climate conditions. In order to assess the influence of climate change in UK cold spells, three separate stochastic sets, of 100K years each, have been created as follows:

- Pre-industrial climate ($\Delta F_{CO_2}$=0 $Wm^{-2}$), corresponding to pre-industrial (1900) concentration of $CO_2$ (296 ppm).

- Current climate ($\Delta F_{CO_2}$=1.6 $Wm^{-2}$), corresponding to a present day (2018) concentration of $CO_2$ (400 ppm).

- Future climate ($\Delta F_{CO_2} = 2.0\ Wm^{-2}$), corresponding to the future year 2030 concentration of $CO_2$ (435 ppm), according to the RCP4.5 emission scenario.

**Histogram of NAOI**

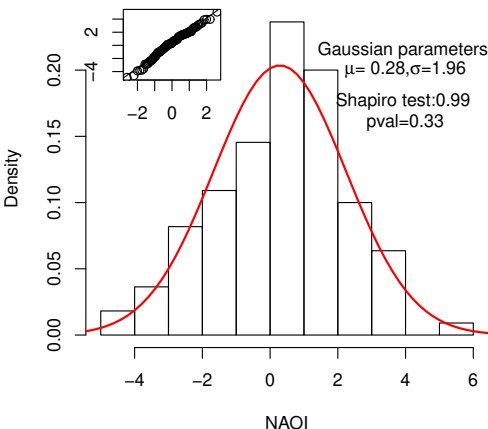

Gaussian parameters
μ= 0.28,σ=1.96

Shapiro test:0.99
pval=0.33

**Figure 6.** Histogram of the NAOI and the pdf of the fitted Gaussian distribution (red line).

The choice of year 2030 assures a relative close time distance which is more relevant for the insurance industry (UNEP, 2015). At the same time, extrapolating far in the future is particularly problematic, since it relies on the assumption of the stability of these linear relationships, even though they may be significantly altered by changing boundary conditions. The change in the $CO_2$ radiative forcing is calculated here based on the RPC4.5 scenario (2 W m$^{-2}$), but similar values are projected for 2030 by the other RCP scenarios as well (Meinshausen et al., 2011). Each year of the three stochastic sets above is associated with a random NAOI value that has been simulated assuming a Gaussian distribution, fitted to the historical NAOI dataset (see Fig. 6). The influence of NAO on each one of these sets can thus be discerned by selecting only the simulated years with negative or positive NAOI values.

The small sample size used in this study (110 years of data) together with the high dimensions of the modelled pdf (67) can lead to large uncertainties in the estimated return periods. Following Bevacqua et al. (2017a), the model uncertainty is assessed using a parametric bootstrap approach, where a large number of models are created using as basis, instead of observations, randomly simulated data from the selected RVM. In particular, confidence intervals are constructed as follows:

– A simulation with the same length as the observed data (i.e. 110 years) is repeated for B = 500 times.

– For each of these B = 500 samples, a new full model is fitted (including new GEV and logistic regression model parameters at each cell and new RVM structure, pair-copula families and parameters) following the methodology described in Sect. 3.3.

– For each of the resulting B = 500 RVMs, a simulation of 10K years of winter-seasons is performed. The uniform variables are then transformed using the (new) inverse marginal pdfs and the corresponding return period levels are estimated.

– The uncertainty in the return levels is estimated by identifying the 95% confidence interval (i.e. the range 2.5–97.5 %) from these 500 return level curves.

Due to computational constraints, confidence intervals are computed only for the stationary model. In addition, the simulation length has been reduced to 10K years (instead of 100K), which implies that part of the calculated uncertainty is due to Monte Carlo sampling variability. In order to investigate further the sources of this uncertainty, the uncertainty associated with the RVM only is separated from the uncertainty of the full model, i.e. of the joint pdf, by calculating confidence intervals with the same approach as described above, but using the same marginal pdfs in each bootstrap repetition.

## 4  Results and discussion

### 4.1  Return period maps

The obtained stochastic sets (see Sect. 3.4) are used to create return period maps for the different climatic conditions. The top panels of Fig. **??** represent the AFI values that occur once every 10, 25, and 50 years based on the stationary model. The empirical return periods are also plotted for comparison (bottom panels). These are calculated for each cell as 1/(1-P), where P represents the cumulative probabilities of the ranked values and is calculated based on the Weibull formula P=i/(n+1) (Makkonen, 2006). The spatial pattern is consistent between the empirical and stochastic sets, showing largest AFI values in high elevation areas, as expected. However, the empirical values are in general somewhat larger than the stochastic set. This difference is driven by the exceptional 1962/63 event which is estimated empirically at 1 in 110 years but is predicted to be less frequent according to the GEV fits. The probability of such an event happening today is discussed in detail in Sect. **??**.

Return period maps at higher return periods (100, 200, and 500 years) for the pre-industrial, current, and future climate stochastic sets are shown in Fig. **??**. UK in the beginning of the 20th century has been experiencing much colder winters than today. By 2030, the future climate change scenario, extreme cold events with AFI larger than 100 °C become quite rare (above 100 years RP) everywhere except at mountainous regions. At high return periods and across all scenarios, the model predicts larger AFI values for the southern part of UK in comparison to the north. The extreme AFI values in the south are driven by the exceptional 1962/63 winter which has been more severe in the South than the North (see Fig. 2b). Excluding this winter from the analysis results in almost much lower AFI values in most of the region (not shown).

### 4.2  Regional return period AFI curves

The vine copula methodology permits the estimation of the hazard return periods over aggregated regions in the UK. Since our focus is mainly on inhabited areas, for each simulation year ($y$) and for each region, the "weighted AFI" (wAFI) is computed, where the AFI value at each cell $j$ is weighted by the corresponding number of residential properties ($n_j$), as shown in Eq. (**??**). The weighted AFI thus places more weight on the hazard over large populated urban areas than agricultural or mountainous areas. The number of residential properties in the UK is taken from the PERILS Industry Exposure Database (https://www.perils.org/), which contains up-to-date high quality insurance market data at Cresta level ("Catastrophe Risk

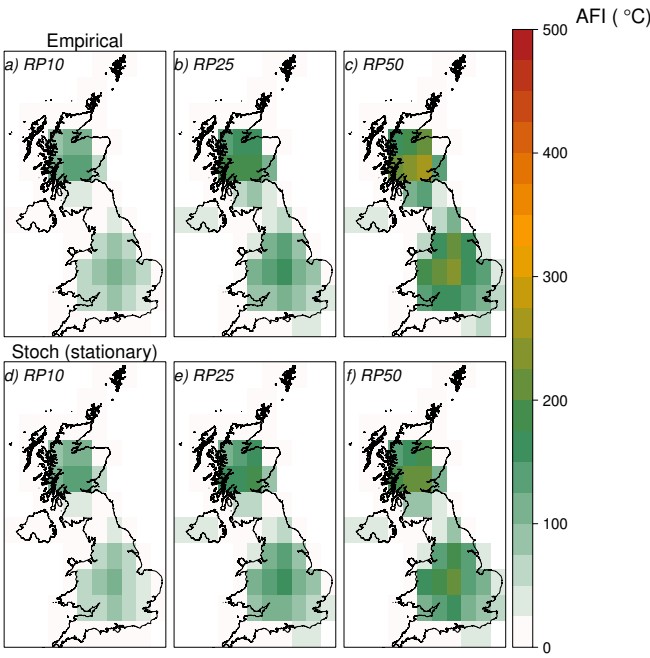

**Figure 7.** (Top panels) Maps of stochastic AFI values (in °C) for return periods of a) 10, b) 25, and c) 50 years. (Bottom panels) Maps of the corresponding empirical AFI values.

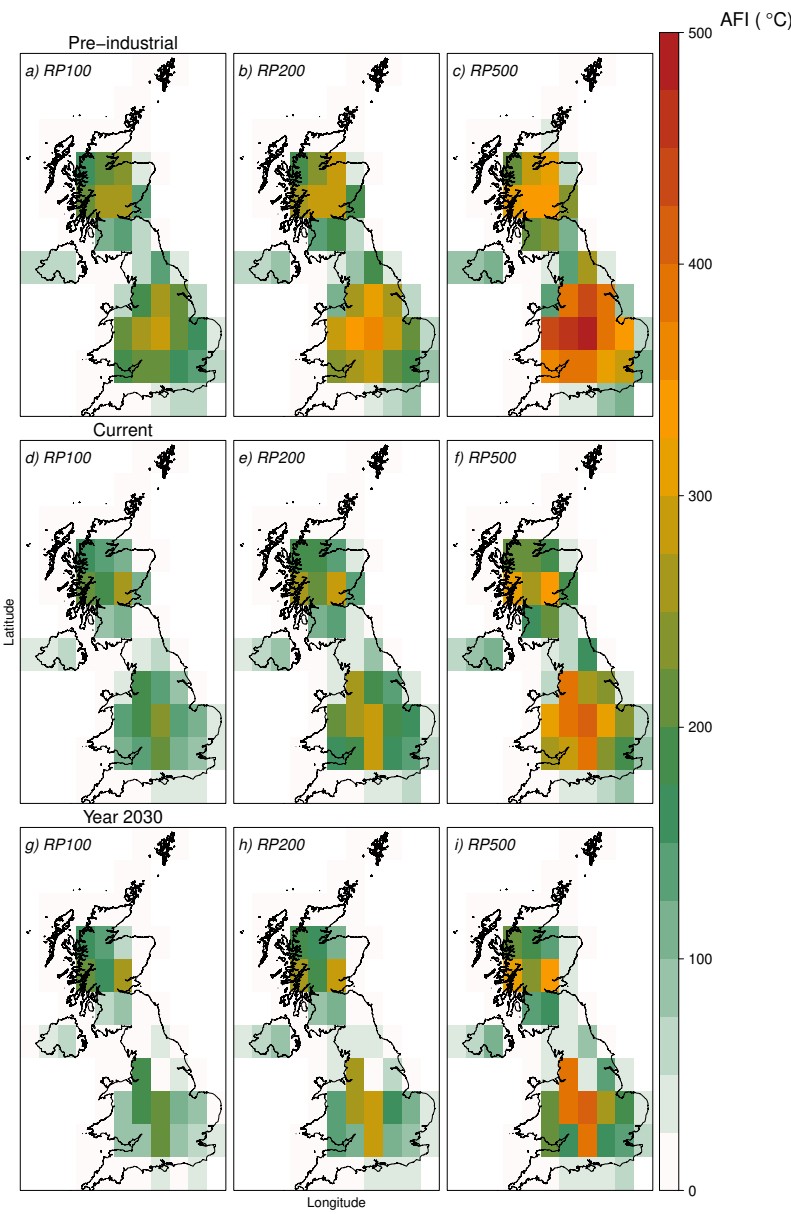

**Figure 8.** Maps of stochastic AFI values (in °C) for return periods of 100, 200, and 500 years for pre-industrial (top panels), current (middle panels), and future (bottom panels) climate.

Evaluation and Standardizing Target Accumulations", https://www.cresta.org/) based on data directly collected from insurance companies writing property business in the UK. Return period wAFI curves for both the empirical and the stochastic data are shown in Fig. **??**. Analogous return period plot based on mAFI, i.e. without weighting, can be found in the Appendix (Fig. **??**).

$$wAFI_y = \frac{\sum AFI_{j,y} \cdot n_j}{\sum n_j} \tag{8}$$

### 4.2.1 Model uncertainty

The stationary model is utilized to analyze the uncertainty in the model results and investigate its sources. Fig. **??** shows the empirical and the stochastic return period curves of wAFI for the entire UK, together with their associated uncertainties. The empirical return periods calculation is described in Sect. **??**, while their uncertainty intervals are computed from the $2.5^{th}$ and
$97.5^{th}$ quantile of the beta probability distribution function (Folland and Anderson, 2002). The stochastic curve and confidence intervals are computed as described in Sect. 3.4. The uncertainty in the model is found to be large, only marginally lower, then the empirical estimates and is associated to the short historical record length. Most of the uncertainty (around 90% for RPs greater than 50 years) appears to originate from the uncertainty in the GEV distribution parameters, with the remaining 10% to be due to the RVM model (dark shaded area in Fig. **??**). Extreme-value theory is considered as a state-of-the-art procedure
to find values for return periods that amply exceed the record length and has been utilized in this study. However, a common difficulty with extremes is that, by definition, data is rare and as a result, the shorter the record length, the more inaccurate is the estimation of the GEV parameters. The results presented in the following sections should therefore be interpreted being aware of the existing uncertainties.

### 4.2.2 The 1962/63 winter return period and climate change influence

Return period curves for the stochastic sets under pre-industrial, current and future climate conditions are shown in Fig. **??**. The 1962/63 winter, with a wAFI of 209 °C, has been the coldest in the reanalysis data in the UK and, thus, it is estimated empirically as a 1 in 110 years event (i.e. the length of the data set). This corresponds well with the Central England Temperature (CET) record, the oldest continuously running temperature data set in the world (Manley, 1974). According to the latter, only two other winters (1683/84 and 1739/40) have been colder than 1962/63 in the last 350 years, suggesting a return period in the
range of 110-120 years, as well. The stationary model overestimates this winters' return period which is estimated to 205 years across all the UK (table **??**). Especially in the South of the UK the model suggests that this event has been particularly unusual. In the Northern part of UK on the other hand, the model suggests a lower return period of 106 years, closer to the empirical estimate.

The non-stationary model suggests that under current climate conditions, such an extreme event, is approximately two times
less likely to occur than in the 1960s. This agrees with Massey et al. (2012) who used climate model simulations to demonstrate that cold December temperatures in the UK are now half as likely as they were in the 1960s. Christidis and Stott (2012) also indicate that human influence has reduced the probability of such a severe winter in UK by at least 20% and possibly by

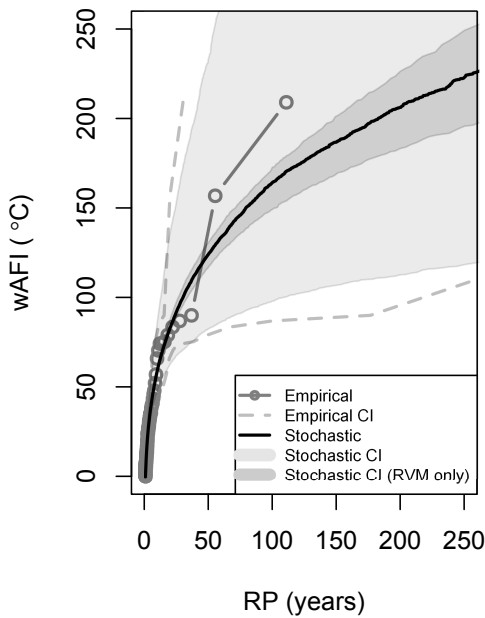

**Figure 9.** Return period curves of wAFI (in °C) based on the historical data (grey) and the stochastic model (black). The 95% confidence intervals are shown as dashed grey lines for the historical data and as a shaded grey area for the stochastic model. The dark shaded area represents the stochastic uncertainty due to the RVM model alone.

as much as 4 times, with a best estimate that the probability has been halved. On the other hand, some recent studies have argued that warming in the Arctic could favor the occurrence of cold winter extremes, and might have been also responsible for the unusually cold winters in the UK of 2009/10 and 2010/11 (Francis and Vavrus, 2012; Tang et al., 2013). This hypothesis though is still largely under debate, see for example Barnes and Screen (2015) and Wallace et al. (2014).

By the year 2030, an event of the same severity as 1962/63, is predicted to become almost twice less infrequent, having a return period of 788 years. Fig. **??**a shows an important reduction in the probability of occurrence of cold extreme events across the whole distribution as a result of the increase of anthropogenic $CO_2$ concentrations. Larger reductions are found for the most extreme events as well, which is probably related to the large increase of the probability of no negative temperatures (P0) for several cells especially around the coast (see Fig. **??**). Similar results are found in both the northern and the southern part of UK, as well (Fig. **??**b).

**Table 4.** Return period estimates (in years) for the 1962/63 winter freeze event, based on wAFI.

| Method | All UK | South UK (<55°N) | North UK (>55°N) |
|---|---|---|---|
| Empirical | 110 | 110 | 110 |
| Stationary stochastic set | 205 | 213 | 106 |
| Non-stationary stochastic sets | | | |
| pre-industrial | 204 | 209 | 102 |
| 1960s | 216 | 219 | 112 |
| current | 433 | 442 | 222 |
| 2030s | 788 | 789 | 400 |

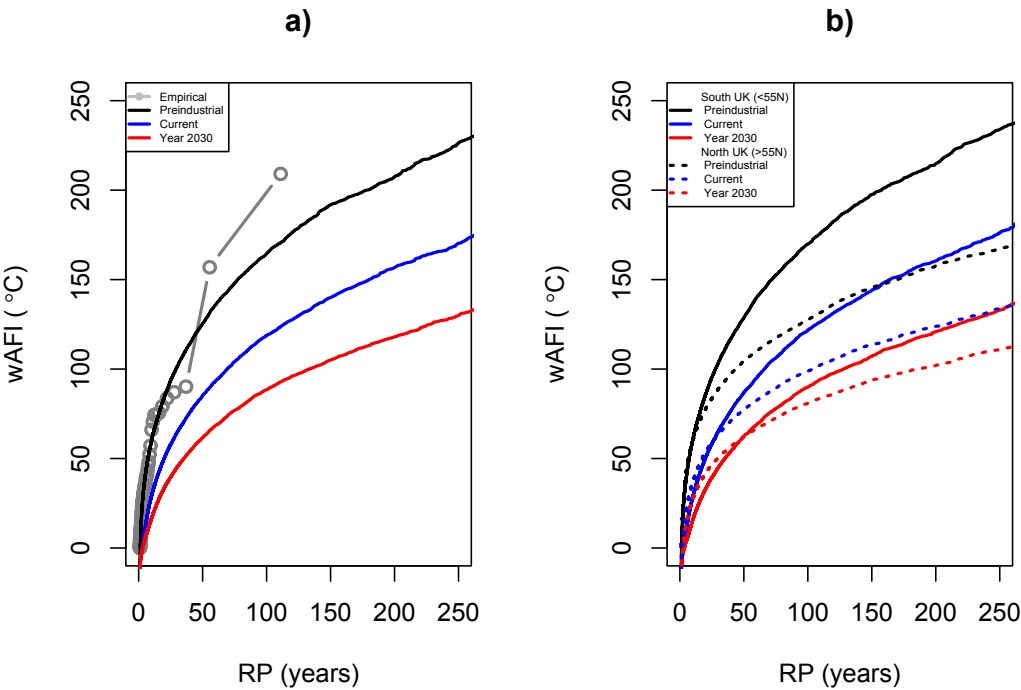

**Figure 10.** a) Return period curves of wAFI (in °C) based on the historical data (grey) and the stochastic model for three different climate conditions: pre-industrial (black), current (blue), and future (red).

b) Same as (a) but separated between South UK (full lines) and North UK (dashed lines). Only stochastic sets are shown.

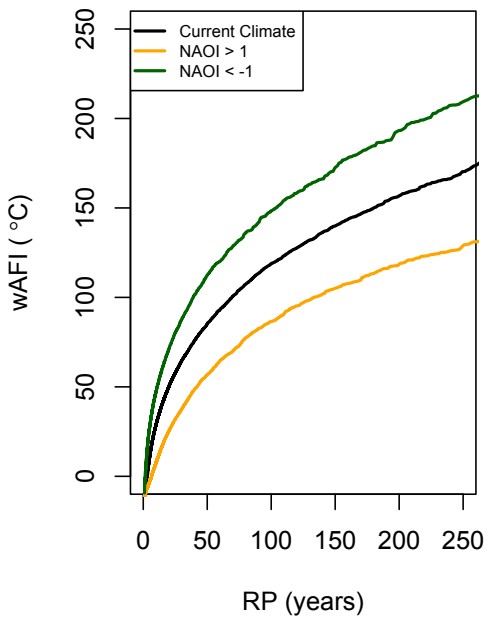

**Figure 11.** Return period curves of wAFI (in °C) based on the current climate stochastic model and assuming a variable NAOI as described in the text (black line). Return period curves based on negative (lower than -1) and positive (larger than 1) values of NAOI are shown with green and orange lines, respectively.

### 4.2.3 NAO influence

The profound effect of NAO on the winter surface temperature over UK has been reported by several studies (Scaife et al., 2005; Scaife and Knight, 2008; Osborn, 2011). In conjunction with those, the model predicts a negative (positive) NAO phase increases (decreases) substantially the probability of a cold event in the UK. Fig. **??** shows the RP curve of current climate wAFI, alongside with RP curves computed solely from simulated years with NAOI values greater than 1 (i.e. representing the positive NAO phase) or years with NAOI values lower than 1 (i.e. representing the negative NAO phase). On average, extreme cold winters are estimated to occur approximately 3 to 4 times more likely during the negative than the positive phase. As an example, an event with wAFI of 100 °C has a return period of 39 years, assuming a negative phase, and 1 in 133 years, assuming a positive phase. Because of its intrinsic chaotic behaviour, NAO is difficult (if even possible) to be predicted (Kushnir et al., 2006). Nevertheless, numerical seasonal forecast systems are currently rapidly improving and have even shown some success in the past (Graham et al., 2006; Folland et al., 2006). Incorporating such information in models could be very useful from the catastrophe risk management perspective.

As already mentioned, the effect of NAO or $CO_2$ radiative forcing in the hazard dependency structure has not been taken into account here and is something to be addressed in a future study. Another point that requires further consideration is the mechanisms that control and affect the NAO and its temporal evolution and in particular how the NAO responds to external $CO_2$ forcing (Hurrell, 2015).

## 5    Conclusions

This paper presents a probabilistic model of extreme cold winters in the United Kingdom. The hazard is modeled using the Air Freezing Index, an index which takes account both the magnitude and the duration of air temperature below freezing and is calculated from the ERA-20C reanalysis temperature data covering the period from 1900 to 2010. Extreme value theory has been applied in order to estimate the probability of extreme cold winters spatially across the UK. More importantly, the spatial dependence between regions in the UK has been assessed through a novel approach which takes advantage of the vine copula

methodology. This approach allows the modeling of concurrent high AFI values across the country which is necessary in order to assess reliably the extreme behaviour of freeze events.

Recognizing the non-stationary nature of climate extremes, the model also incorporates NAO and climate change effects as predictors. Stochastic sets of 100K years representing different climate conditions (i.e. pre-industrial, current, or future climate and positive or negative NAO) have been generated and the return periods of extreme cold winters in UK, such as the

"Big Freeze of 1962/63" have been estimated. According to the model, the occurrence of such an event is calculated to have decreased approximately two times during the course of the 20th century as a result of anthropogenic climate change. The model further predicts that by 2030s, extreme cold winters will become even more uncommon and will occur about twice less frequently under the influence of increasing $CO_2$ concentrations. The frequency of extreme cold spells in UK has been found to be heavily modulated by NAO, as well. A cold event is estimated to occur ≈3-4 times more likely during the negative than

the positive phase.

However, considerable uncertainty exists in these estimates which should be interpreted with caution. The 110-year re-analysis record used in this study is estimated to be short and the level of uncertainty in extremal estimates with long return periods is high. Additional uncertainty may also be introduced by possible spurious trends in the reanalysis data set. A longer record of temperature data would be necessary in order to reduce the uncertainty and high quality long-term reanalysis products

with multiple ensemble members could help towards this direction.

**Appendix A: Copulas and vine copulas**

According to Sklar's theorem, the joint multivariate distribution of a set of d random vectors can be fully specified by the separate marginal distributions and by their (d-dimensional) copula, which defines the dependence structure between them. More precisely, consider a vector of $X = (X_1, ..., X_d)$ of random variables with a joint probability density function (pdf), $f(x_1, ..., x_d)$. Sklar's theorem (Sklar, 1959) states that any multivariate continuous distribution function $F(x_1, ..., x_d)$ with marginals $F_1(x_1), ..., F_d(x_d)$ can be written as:

$$F(x_1, ..., x_d) = C(F_1(x_1), ..., F_d(x_d)) \tag{A1}$$

for some appropriate d-dimensional copula C, which is uniquely determined on $[0,1]^d$.

The probability density function (pdf) of X, $f(x_1, ..., x_d)$, can be found by taking the partial derivatives with respect to $X$:

$$f(x_1, ..., x_d) = c(u_1, ..., u_d) \prod_{i=1}^{d} f_i(x_i) \tag{A2}$$

where $c(u_1, ..., u_d)$ is the copula density, given by

$$c(u_1, ..., u_d) = \frac{\vartheta^d C(u_1, ..., u_d)}{\vartheta u_1 ... u_d} \tag{A3}$$

Expression **??** is important in terms of modelling because it permits to define a multivariate density as the product of marginal pdfs and a copula density function that captures the dependence between the random variables (Abbara and Zevallos, 2014). For a theoretical introduction to copulas, see Nelsen (2006); Meucci (2011); Joe (2014); Durante and Sempi (2015); for a practical/engineering approach and guidelines, see Genest and Favre (2007); Salvadori and Michele (2007); Salvadori et al. (2014, 2015)

To quantify the dependence between variables, different measures have been defined, addressing different aspects of dependence. A common measure of overall dependence is the Kendall rank correlation coefficient, commonly referred to as Kendall's $\tau$ coefficient (Genest and Favre, 2007). However, dependence of rare events cannot be measured by overall correlations: even if two variables are completely uncorrelated, there can be a significant probability of a concurrent extreme event in the two, i.e., they can still be tail dependent. Tail dependence describes the amount of dependence in the lower tail or upper tail of a bivariate distribution. For its mathematical definition see Haff et al. (2015).

One important complication is that identifying the appropriate d-dimensional copula is not an easy task. In high dimensions, the choice of adequate families is rather limited (Brechmann and Schepsmeier, 2013). Standard multivariate copulas, either do not allow for tail dependence (i.e. multivariate Gaussian) or have only a single parameter to control tail dependence of all pairs of variables (Student-t and Archimedean multivariate copulas). This is particularly problematic for catastrophe modeling applications, where a flexible modeling of tails is vital to assess reliably the extreme behaviour of natural events.

Vine copulas provide a flexible solution to this problem based on a pairwise decomposition of a multivariate model into bivariate (conditional and unconditional) copulas, where each pair-copula can be chosen independently from the others. In

particular, asymmetries and tail dependence can be taken into account as well as (conditional) independence to build more parsimonious models. Vines thus combine the advantages of multivariate copula modeling, that is separation of marginal and dependence modeling, and the flexibility of bivariate copulas (Brechmann and Schepsmeier, 2013).

As an example, in a 4-dimensional case, the joint pdf can be decomposed as a product of 6 pair-copulas (3 uncoditional and 3 conditional) and 4 marginal pdfs as shown in Eq. (**??**):

$$
\begin{aligned}
f(x_1, x_2, x_3, x_4) = {} & f(x_1)f(x_2)f(x_3)f(x_4) \\
& \times c_{12}(F_1(x_1), F_2(x_2)) \\
& \times c_{23}(F_2(x_2), F_3(x_3)) \\
& \times c_{34}(F_3(x_3), F_4(x_4)) \\
& \times c_{13|2}(F_{1|2}(x_1 \mid x_2)), F_{3|2}(x_3 \mid x_2))) \\
& \times c_{24|3}(F_{2|3}(x_2 \mid x_3)), F_{4|3}(x_4 \mid x_3))) \\
& \times c_{14|23}(F_{1|23}(x_1 \mid x_2, x_3)), F_{4|23}(x_4 \mid x_2, x_3)))
\end{aligned}
\tag{A4}
$$

The above decomposition is not unique and Bedford and Cooke (2002) introduced a graphical structure called regular vine (R-Vine) structure to represent this decomposition with a set of nested trees. The dependence structure with three trees for the 4-dimensional example above is shown in Fig. **??**. More details on vine copulas can be found in Aas et al. (2006); D. and Schirmacher (2008); Czado (2010); Schepsmeier (2013).

**A1**

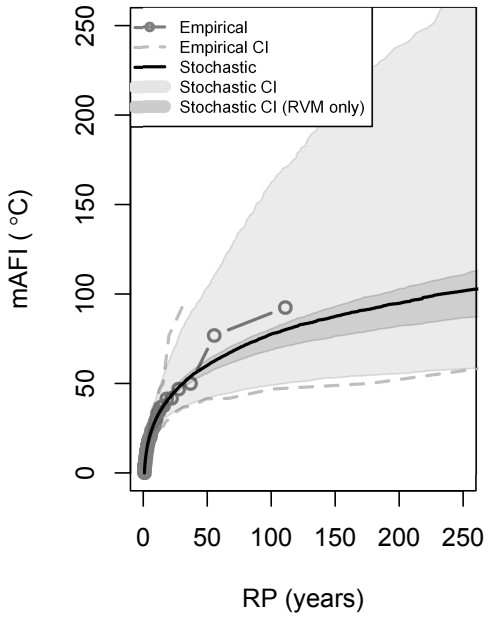

**Figure A1.** Similar to Fig. **??** but for mAFI (without weighting, in °C).

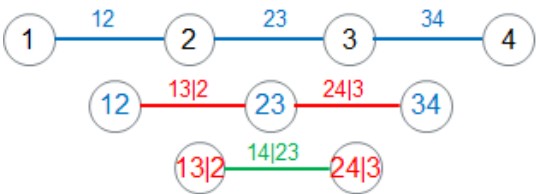

**Figure A2.** Example of 4-dimensional R-Vine trees corresponding to the decomposition shown in Eq. (**??**).

*Disclaimer.* TEXT

*Acknowledgements.* TEXT

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
