# Peer review of "A hazard model of subfreezing temperatures in the United Kingdom using vine copulas"

_Natural Hazards and Earth System Sciences, 2017_

## Referee Comment (RC1) · Anonymous Referee #1 · 21 Dec 2017

Journal: Natural Hazards and Earth System Sciences (NHESS)
Title: A hazard model of subfreezing temperatures in the United Kingdom using vine copulas
Author(s): Symeon Koumoutsaris
MS No.: nhess-2017-389
MS Type: Research article

**General comment**

The author assesses the return period of extreme cold winter across UK, based on the Air Freezing Index (AFI). The first part of the paper is based on the estimation of AFI return periods for single spatial grid boxes; in the second part the author makes use of pair-copula constructions for estimating the return period of the average AFI (or weighted AFI based on local population in grid boxes) over larger regions (i.e. UK, South England, North England & Northern Ireland, and Scotland). The latter is useful, e.g., for insurance portfolio loss analysis which requires hazard calculation across large regions. The author concludes saying that according to the model, the extreme winter 1962/63 has a return period of ~89 years, although this result may be affected by uncertainties due to the shortness of the data used for the model calibration, and to the background climate change trend. The work looks conceptually interesting to me, although the structure of the presentation, the figure descriptions, and the explanation of procedure details, would need improvements and more attention. Above all, I have the following comment.

**Main Comment**

The large (non-estimated) uncertainties due to the shortness of the data can strongly compromise the interpretation of the results. Although Vine Copula is a flexible mathematical tool for modelling multivariate probability density functions (pdfs), in general the sample size used for fitting multivariate pdfs should be large enough. The sample size needed for a reasonable fit of the pdf increases very fast (~exponentially) with the dimension of the pdf. In this study 51 observations are used to fit a 170-dimensional pdf (I hope I have not misunderstood...). I believe that this is a very small size for such a fit. Moreover, eventual serial correlations in the time series of the 170 variables would even corresponds to a reduction of the actual available data sample size (Serinaldi et al., 2015). In the case of Vine Copula, the limitation associated with the small sample size occur mostly when fitting conditional pair-copulas. The author says that he is using many independent copula: if this is a reasonable choice then it corresponds to somehow virtually reduce the dimension of the pdf. The structure of the vine is not fully specified, however we can estimate that around 2417 pair-copulas are conditional and non-independent***, which is a very large number. Serinaldi et al. (2015) and Bevacqua et al. (2017) did show that even in much better conditions (much higher ratio: sample size / vine dimension), the uncertainties of the pdf are very large, with propagation of the uncertainties to the estimated quantity of interest (return periods). Personally, I am surprised about how small is the ratio sample size / vine dimension of this work, and I am not aware about studies where a similar ratio is used. For example: (1) Brechmann and Schepsmeier (2013) (cited by the author in the paper) use 396 observations for modelling a 6-dimensional vine; (2) Hobæk Haff et al. (2015) use 40 years of daily data for modelling a 64-dimensional vine; (3) Dißmann et al. (2013) used 2337

observations for fitting a 16-dimensional vine; (4) Brechmann et al. (2012) used 1107 observation to model a 19-dimensional vine.

Here, in my opinion the small sample size can represent a strong limitation as I expect huge uncertainties associated with the results. Although the author says that there are uncertainties about the results, I think that this issue should be explicitly discussed referring to the ratio sample size / vine dimension. According to me, the outcome of this discussion should be considered as crucial for deciding whether the paper should be published. This discussion should refer to literature where similar ratio (sample size / vine dimension) were involved in the analysis (literature where they show that the approach is reasonable). If these papers are not available, I think it should be shown if the approach is reasonable, for example via quantifying the impact of the short sample size on the uncertainties of the results (via bootstrap, similarly to what has been done by, e.g., Serinaldi et al. (2015) and Bevacqua et al. (2017).

*** For a D-vine: there are totally n(n-1)/2=170(169)/2=14365 pair copulas in the Vine (169 are non-conditional copulas). As the 82% of copulas are independent, about 2586 copulas (18% of 14365) are non-independent. The non-conditional copulas are 169, therefore at least about 2586-169=2417 copulas are conditional.

References:
Serinaldi, F.: Can we tell more than we can know? The limits of bivariate drought analyses in the United States, Stoch. Env. Res. Risk A., 30, 1691, https://doi.org/10.1007/s00477-015-1124-3, 2015.

Bevacqua, E., Maraun, D., Hobæk Haff, I., Widmann, M., and Vrac, M.: Multivariate statistical modelling of compound events via pair-copula constructions: analysis of floods in Ravenna (Italy), Hydrol. Earth Syst. Sci., 21, 2701-2723, https://doi.org/10.5194/hess-21-2701-2017, 2017.

Brechmann, E. C. and Schepsmeier, U.: Modeling Dependence with C- and D-Vine Copulas: The R Package CDVine, Journal of Statistical Software, doi:10.18637/jss.v052.i03, https://www.jstatsoft.org/article/view/v052i03, 2013.

Hobæk Haff, I., A. Frigessi, and D. Maraun (2015), How well do regional climate models simulate the spatial dependence of precipitation? An application of pair-copula constructions, J. Geophys. Res. Atmos., 120, 2624–2646, doi:10.1002/2014JD022748.

Dißmann, J., Brechmann, E., Czado, C., and Kurowicka, D.: Selecting and estimating regular vine copulae and application to financial returns, Computational Statistics & Data Analysis, 59, 52 – 69, doi:https://doi.org/10.1016/j.csda.2012.08.010, http://www.sciencedirect.com/science/article/pii/S0167947312003131, 2013.

Brechmann EC, Czado C, Aas K (2012). "Truncated Regular Vines in High Dimensions with Applications to Financial Data." Canadian Journal of Statistics, 40(1), 68–85.

**Specific comments**

Structure of the paper

I think that a more typical structure would improve the manuscript, e.g. Introduction, Data, Methods, Results, Discussion, Conclusion. At the moment, the structure is not as a usual reader would expect.

About the vine (P9)

1) An equation with an example of a Vine (e.g. in 4 dimension) would be helpful for the reader. In particular this should be shown in combination with the uniform variables used for the vine fit (i.e. the "marginal variables" coming from the GEV).
2) The structure of the used vine is not clear. A table with the percentage of family types used in each tree would be appreciated by the reader.
3) There is not enough information about the procedure used for the fitting of the vine, e.g. what criteria was used for the selection of the RVM structure, what criteria was used to fit the pair copulas, or how you assigned independence to some of the par-copulas. There are references to the R-package, however this is not enough, also considering that in the package different approaches for fit can be used.

Methodology

The part where return periods of averaged AFI are computed based on the variables modelled by the copula is very similar to the so called *structural approach* used in the following references (which I suggest to cite to show similar applications to the reader):

1) Salvadori, G., Durante, F., Tomasicchio, G., and D'alessandro, F.: Practical guidelines for the multivariate assessment of the structural risk in coastal and off-shore engineering, Coast. Eng., 95, 77–83, https://doi.org/10.1016/j.coastaleng.2014.09.007, 2015.
2) Bevacqua, E., Maraun, D., Hobæk Haff, I., Widmann, M., and Vrac, M.: Multivariate statistical modelling of compound events via pair-copula constructions: analysis of floods in Ravenna (Italy), Hydrol. Earth Syst. Sci., 21, 2701-2723, https://doi.org/10.5194/hess-21-2701-2017, 2017.
3) Serinaldi, F.: Can we tell more than we can know? The limits of bivariate drought analyses in the United States, Stoch. Env. Res. Risk A., 30, 1691, https://doi.org/10.1007/s00477-015-1124-3, 2015.

P1 l1: *the third coldest winter ever recorded.* Where and according to what criteria?

P3 l5  *It is based on rigorously quality  checked station data interpolated to a regular grid using inverse-distance weighting, as desribed in Perry et al. (2009).*
It should be mentioned here or later that therefore the dependencies catched by the copulas may be partially due to the interpolation itself.

P3 l10 *Nevertheless, local temperature may be subtly different in certain micro-climates, such*
*as upland and urban regions.*

I would mention that however the resolution 5km x 5km may not always be realistic, depending on the number of stations which were available for the creation of the data set.

P3 l29 *98.3°C.*
Based on line 17, I expected negative values for the AFI. Could you mention that you take the absolute values of the temperature? Also, it would be appreciated if you would show the equation of the AFI.

P3 l32 *After 1962/63, a long run of mild winters followed until late 1978 and early 1979 (Figure 2).*
Is this in Figure 2 the AFI averaged over UK?
Please, use °C in the y label of Fig. 2.

P5 l4 *An additional term was included, the probability of no hazard (P0), in order to account for the cells mainly on the south England coast that have years with no negative temperatures at all.*
   1) **Does this mean that for some cells the GEV is fitted on very few data?** Please give information about this, and on the goodness of the fit for these cells.
   2) Please, specify how P0 is estimated, e.g. N_occurence/N_years.

FIG 3
   1) I assume that the "*historical AFI GEV fit (black circles)*" is the empirical estimate. If yes, is this computed as written in P6 L5? Please, specify this.
   2) Could you specify the estimated parameters, or also only making clear to the reader whether the difference is due to the selection of different family type (Gumbel, Frechet, and Weibull distributions)?

P5 l20 *As an example, the GEV fit for a single cell over London is shown in Figure 3. The grey line represents the GEV fit without any weighting applied, while the black curve is estimated using the TWMLE method with an improved fit towards the tail of the distribution (i.e. the more extreme events).*
I would rather say that you get a curve that is nearer to the empirical estimate.

P6 l2 *Other urban regions (e.g. Manchester or Midlands area) do not stand out as much as a result of the low grid resolution.*
Can this also be due to the original data format? For example there may be not enough stations around some urban areas.

P9 l8 *At the first level, 49% of the selected bivariate copulas are found to be Gumbel which implies greater dependence at larger AFI values.*
You refer to the tail dependence, I assume. Make it more clear, please. Greater with respect to what?

P9 l13 *The RVM is used to simulate 10,000 years of winter-seasons in the UK. This amount of realisations should be long enough in order to estimate with enough confidence the 200 year RP hazard, which is commonly associated with capital and regulatory requirements.*

The 10,000 years time series should be long enough to neglect uncertainties associated with the Monte Carlo simulations (which is the method used for extracting the return period associated with the fitted parametric pdf) (Serinaldi et al. (2015) and Bevacqua et al. (2017)). One should ensure if the sample is "long enough" via repeating the (10,000 years) simulations several times and checking if the there are differences in the estimated return period (if there are no differences, the 10,000 years sample is long enough). Performing a long enough simulations allows one to get a convergence to the true return period that one would get analytically from the fitted pdf (given the complexity of the problem it is impracticable to get an analytical derivation of the RP).

Performing a long simulation does not solve the issue about the model uncertainties (uncertainties existing about the pdf), which is there because the pdf is calibrated on a finite - very short - sample. I suggest to discuss this in a way to make difference between these different type of uncertainties.

P9 l 27 *The exceedance probability (EP) curve of wAFI is shown in Figure 7, both for the historical and the stochastic data.*
So far you talked about RP. Personally, I think that it would be better to keep the same terminology instead of introducing EP, or at least use also RP here.

P9 l27 *The uncertainty intervals in the historical data are computed as the 5th and 95th quantile of the probability density function (Folland and Anderson, 2002).*
I suggest to use: "The uncertainty intervals in the return period (estimated empirically?) of the historical data are computed via the 5th and 95th quantile of the probability density function"

P10 l2 *low tail dependence.* Gaussian and Frank copula have zero tail dependence, not "low". It may be helpful to better introduce the tail dependence in a sentence where you talk about it for the first time.

P10 l2 *On the other hand, the low impact of the other copula familes is due to the fact that the extreme hazard values are mainly driven by the large dependencies between nearby cells, especially at the first tree levels.*
Could you please argue this better?

P10 l16 *However, recent studies suggest that cold weather in the UK is likely to be less severe, to occur less frequently, and to last for a shorter period of time than was historically the case due to anthropogenic induced climate change (on Climate Change, 2017).*
I would already mention here that there is debate about this (as you then specify in the next paragraph).

P11 l7 *As shown in Figure 8, South England is in general warmer than the North England and Northern Ireland region, partially driven by the urban micro-climate effect of the London area. The 1962/63 winter was less extreme in this region (wAFI of 139° C) with an estimated return period of 1 in 79 years. On the other hand, Scotland is usually significantly colder than the rest of UK, reaching for example AFI values of 100 ∘ C almost 2 times more often.*

Please, make more clear in the text (and in the figure captions) when you talk about AFI, wAFI, averaged non-weighted AFI (and in which area is computed the average (UK, or sub-regions)). Also, when introducing eq. (3), I suggest to anticipate that you are going to use the wAFI both on UK and subregional scale.

Figure captions. Please improve the Figure captions with more information. For example in Fig 2 what is the NAOI (North Atlantic Oscillation Index)?

**Technical corrections**

P3 l6 *desribed.* Described

P4 l1 *that winter*. You may use "winter 1978/79".

Fig 4 and 5. Could you please use the same scale range, i.e. 0-400°C

P10 l2 *familes.* Families

---

## Referee Comment (RC2) · Anonymous Referee #2 · 29 Dec 2017

**REVIEW REPORT**

**Journal:** Natural Hazards and Earth System Sciences (NHESS)
**Paper:** nhess-2017-389
**Title:** A hazard model of subfreezing temperatures in the United Kingdom using vine copulas
**Author(s):** Symeon Koumoutsaris

**GENERAL COMMENTS.**

The paper is an interesting one, and outlines an original multivariate investigation concerning subfreezing temperatures. The comments posted by Referee 1 already provide an excellent, detailed review, with which I (almost) fully agree. Below, please find further notes: my objections should be read as constructive advices. Some relevant bibliography is reported at the end of this review.

1. I noticed that there is some confusion between the notions of probability distribution function and probability density function (e.g., Page 10, Lines 5–7: "The uncertainty intervals in the historical data are computed as the 5th and 95th quantile of the probability density function (Folland and Anderson, 2002)"). The probability distribution function is the integral of the probability density function (if it exists). The quantiles are the inverses of the probability distribution function (a non-decreasing one), not of the density function (which may not even be monotone). The Author must check the paper and fix all the points where such a confusion arises, otherwise the paper is not correct from a probabilistic point of view.

2. I was puzzled by the comment of Referee 1 concerning the sample size, and I ask the Author to clarify the issue: here, 170 variables are at play, each observed 51 times. To the best of my understanding, the idea revolving around Vine copulas is that any multivariate density can be decomposed into a (suitable, maybe not unique) product of univariate marginal densities and bivariate copula densities: in turn, only univariate and bivariate fits should be needed, isn't it? Thus, apparently, the fitting problem may not be so severe.

   Clearly, trying and fitting the upper tail of a GEV law using only 51 observations may be difficult (although the TWMLE escamotage is used), but it may not be impossible. Similarly, trying and fitting a bivariate copula using only 51 pairs may not be advisable, but it is not uncommon in practical applications. Overall, should my interpretation be correct, the game played by the Author may not be a "Mission Impossible", rather an "Uncertain Mission"...

   Thus, I kindly ask the Author to clearly explain the situation, and to provide estimates of the uncertainties as explained below.

3. I definitely agree with the comment of Referee 1 concerning the procedure to estimate the uncertainties (Page 9, Lines 23–ff.). As a rule of thumb, 1000 independent repetitions of the 10,000-years Monte Carlo simulations are usually suggested in literature, in order to provide "reasonable(?)" estimates of the confidence intervals of interest (clearly, it may be adjusted depending on the computational burden).

4. My main "perplexity" concerns a methodological issue. In this work, I can see the Mathematics/Statistics, but I do not see the Physics, which, instead, should be the starting point. To be clear, and to the best of my knowledge, the procedure used to construct the 170-dimensional copula finds its justification in an aggregation/clustering algorithm based solely on statistical considerations

(Page 9, Lines 13–14: "The method follows an automatic strategy of jointly searching for an appropriate R-Vine tree structure"). If I remember it correctly, the algorithm is based on the Kendall $\tau$ and/or on the Kendall Distribution Function **K**, and/or, in general, on the strength of the statistical association between the variables at play. While interesting and meaningful from a mathematical point of view, such a procedure may eventually (statistically) associate grid cells having little, or negligible, physical link (for instance, could this be the case of the grid cells corresponding to Edinburgh and London, quite far apart from a spatial and a climatic point of view?)

In other words, important information like, e.g., the latitude (corresponding to different climatic regions) may not be considered/used by the statistical procedure adopted for constructing the overall copula. The Author is kindly asked to discuss the issue, and to provide suitable justifications. Is it possible to modify the construction of the 170-dimensional copula in order to take into account the physics of the phenomenon?

5. The Author has modeled the historical data, but, should the climate be really changing, then (at least from an Insurance point of view) the Author should account for it in his model, e.g. by introducing (in the long term simulations) suitable temporal patterns in the GEV/copula parameters according to available projections of the future climate (like, e.g., in IPCC scenarios). A comment is required on this issue.

6. In Section 3.1 "Results and discussion", the Author mentions the actual debate about climate changes (already commented by Referee 1). I would suggest to take a look at a recent paper by Vezzoli et al. (2017), where the traditional validation criteria of climate models are discussed, and an advanced/thorough distributional perspective is outlined: it may partially explain why several crucial hypotheses are "still largely under debate" (as claimed by the Author and Referee 1), and may partially account for the general inability to draw up clear settlements.

**SPECIFIC COMMENTS.**

**Page(s) 2, Line(s) 23–ff.**

For the benefit of unskilled readers and practitioners, here the Author should provide general references involving seminal books, papers, and guidelines concerning copulas, like writing: "For a theoretical introduction to copulas, see Nelsen (2006); Joe (2014); Durante and Sempi (2015); for a practical/engineering approach and guidelines, see Genest and Favre (2007); Salvadori and De Michele (2007); Salvadori et al. (2007, 2014, 2015)". Instead, citations concerning Vine copulas, being more specific and related to the modeling outlined in this work, may be postponed later.

**Page(s) 9, Line(s) 20–22.**

**Author.** "Goodness-of-fit is performed for the final selected R-Vine Model (RVM) based on the RVineGofTest algorithm of the same R package (Schepsmeier, 2013). The Cramer von Mises test, which compares the empirical copula with the RVM, has a value of 0.019 and a p.value = 1, which indicates that the fitted RVM cannot be rejected at a 5% significance level."

**Referee.** I am puzzled by such a large p-Value: in my opinion, it may entail a large probability of Type II error, i.e. accepting a False Null Assumption (this a typical performance of Cramer-von-Mises and similar tests, when the sample size is insufficient). The Author is kindly asked to discuss the issue, and to provide suitable justifications.

**References**

Durante, F., Sempi, C., 2015. Principles of copula theory. CRC/Chapman & Hall, Boca Raton, FL.

Genest, C., Favre, A., 2007. Everything you always wanted to know about copula modeling but were afraid to ask. Journal of Hydrologic Engineering 12 (4), 347–368.

Joe, H., 2014. Dependence Modeling with Copulas. CRC Monographs on Statistics & Applied Probability. Chapman & Hall, London.

Nelsen, R., 2006. An introduction to copulas, 2nd Edition. Springer-Verlag, New York.

Salvadori, G., De Michele, C., 2007. On the use of copulas in hydrology: theory and practice. J. Hydrol. Eng. 12 (4), 369–380, (Special Issue: Copulas in Hydrology; doi: 10.1061/(ASCE)1084-0699(2007)12:4(369)).

Salvadori, G., De Michele, C., Kottegoda, N., Rosso, R., 2007. Extremes in Nature. An approach using Copulas. Vol. 56 of Water Science and Technology Library Series. Springer, Dordrecht, ISBN: 978-1-4020-4415-1.

Salvadori, G., Durante, F., Tomasicchio, G. R., D'Alessandro, F., 2015. Practical guidelines for the multivariate assessment of the structural risk in coastal and off-shore engineering. Coastal Engineering 95, 77–83, doi: 10.1016/j.coastaleng.2014.09.007.

Salvadori, G., Tomasicchio, G. R., D'Alessandro, F., 2014. Practical guidelines for multivariate analysis and design in coastal and off-shore engineering. Coastal Engineering 88, 1–14, doi: 10.1016/j.coastaleng.2014.01.011.

Vezzoli, R., Salvadori, G., De Michele, C., 2017. A distributional multivariate approach for assessing performance of climate-hydrology models. Scientific Reports 7:12071, doi: 10.1038/s41598-017-12343-1. URL www.nature.com/scientificreports

---

## Author Comment (AC1) · 6 Mar 2018

The comment was uploaded in the form of a supplement:
https://www.nat-hazards-earth-syst-sci-discuss.net/nhess-2017-389/nhess-2017-389-AC1-supplement.pdf

———————————————————

[Figure]

[Figure]

**Fig. 1.** Weighted AFI RP curve for all cells in UK, based on the historical AFI GEV fit (black circles), the GEV fitted

---

## Author Comment (AC2) · 6 Mar 2018

Dear Editor,

Dear reviewers,

I would like first to thank the reviewers for their constructive and very helpful comments. I am replying below to your comments in red.

With best regards, Symeon Koumoutsaris

**REVIEW 1**

**Main Comment**

This is a fair comment. Indeed the sample size is small relative to the dimension of the problem. In order to estimate the associated uncertainties, I followed a bootstrap approach, as suggested by the reviewer. In particular, similarly to Bevacqua et al. (2017a):

- I simulate B = 1000 samples with the same length as the observed data (i.e. 51 years)
- On each of the B = 1000 samples, I fit a RVM, whose structure is the same as the fitted with the observed data, while the pair-copula families are re-selected for each sample.
- For each of these B = 1000 RVMs, simulate 10K years of correlated samples and estimate the corresponding wAFI return period levels for the entire UK, and for England S, England N & N. Ireland, and Scotland shown in Figures 7 & 8 of the submitted manuscript.
- Estimate the uncertainties in the return levels by identifying the 95% confidence interval (i.e. the range 2.5– 97.5 %) of the B = 1000 return level curves.

For simplicity, let's call this uncertainty as the "RVM uncertainty".

In addition to this uncertainty and in conjunction with comment P9 l13 of the reviewer, I perform another round of re-simulations, in order to estimate the uncertainty associated with the 10K years of the Monte Carlo simulation, as follows:

- I use the "default" RVM to simulate 10K years of correlated samples x 1000 times.
- For each of these 1000 simulations, I estimate the corresponding wAFI return period levels for the entire UK and for the three sub regions, as above.
- Estimate the uncertainties in the return levels by identifying the 95% confidence interval (i.e. the range 2.5– 97.5 %) of the 1000 return level curves.
- I also take into advantage of these 1000 re-simulations in order to estimate more precisely the return period curves using the median of them.

For simplicity, let's call this uncertainty as the "10K uncertainty".

Figure 1 below shows the wAFI RP curves for the whole UK and the three sub-regions (i.e. similar to the figures 7 and 8 of the submitted manuscript), including the 95% confidence intervals (CI) of the RVM (orange dashed lines) and of the 10K (black dashed lines) as computed above.

It is clear from Figure 1 that the two uncertainty ranges are very close to each other. This means that the uncertainty in the RVM model is mainly driven by the randomness due to the 10K year sampling (the 10K uncertainty). In other words, the RVM fitting introduces only a minor additional uncertainty in comparison to the uncertainty due to the Monte Carlo sampling. This happens because a) 64% of the fitted copulas are selected to be independent and b) due to the nature of the hazard, the dependencies between neighboring cells are strong and these are driven almost entirely within the first few trees of the RVM. Both (a) and (b) above, lead to a reduction in the dimensions of the pdf, as the reviewer also mentions.

As a further confirmation, Figure 2 shows the resulting RP curves from models truncated above the first three levels of the RVM tree (using only one 10K year simulation though). Here, the truncated model above level 3 is very close to the default one with no truncation, which suggests that only the first few levels drive the dependencies between the cells.

However, uncertainty is indeed important: as shown in Table 1, the 1962/63 winter RP estimate for the entire UK ranges from 82 to 122 years. Notice as well that the more precise estimate of the RP of the 1962/63 winter derived from all the 1000 simulations of the 10K years (as described above) is 97 years (~ equivalent of 10M years of simulations) instead of 89 years in the submitted manuscript based only on one 10K years simulation. However, notice that the uncertainty is much smaller in comparison to the uncertainty in the empirical curve (i.e. directly from the observed data, grey lines in Figure 1). Because of that, I believe that the model is useful but indeed its results should be interpreted being aware of these uncertainties.

It is also important to note that uncertainties at different regions can be larger and this can be particularly important for example when calculating monetary losses of insurance portfolios where risks may be concentrated in certain areas.

I agree therefore with both reviewers and I will make sure to convey more clearly the importance of those uncertainties in the revised manuscript.

Table 1: Return periods (in years) for the 1962/63 event for regions in the UK. The 95% confidence intervals are also shown for the "10K" and "RVM" uncertainties (see text).

| Region | 1962/63 RP (in years) | | |
|---|---|---|---|
| | Median | 95% CI RVM | 95% CI 10K |
| UK | 97 | 81-120 | 82-122 |
| England S | 111 | 91-138 | 92-140 |
| England N & N. Ireland | 84 | 70-101 | 71-103 |
| Scotland | 55 | 49-65 | 49-67 |

[Figure]

Figure 1: Weighted AFI RP curves for (a) the whole UK, (b) South England, (c), North England & Northern Ireland, and (d) Scotland. Empirical estimates are shown in grey, including their confidence intervals (grey dashed lines). The black solid line respresents the stochastic curve. Confidence intervals due to the 10K years Monte Carlo simulation and due to the RVM fits are shown as the black and orange dashed lines, respectively.

[Figure]

Figure 2: Weighted AFI RP curves for the whole UK. The empirical EP is shown in grey. Results are shown using RVM with truncation above level 1 (red line), above level 2 (blue line), and above level 3 (pink line) or no truncation (default, black line).

**Specific comments**

**Structure of the paper**

I agree and I will re-structure the manuscript as suggested.

**About the vine (P9)**

1) An equation with an example of a Vine (e.g. in 4 dimension) would be helpful for the reader. In particular this should be shown in combination with the uniform variables used for the vine fit (i.e. the "marginal variables" coming from the GEV).

Agreed.

2) The structure of the used vine is not clear. A table with the percentage of family types used in each tree would be appreciated by the reader.

I rewrote this paragraph in order to make it clearer. Table 4 at the end of this document also contains the percentage of family types used in each of the tree level.

3) There is not enough information about the procedure used for the fitting of the vine, e.g. what criteria was used for the selection of the RVM structure, what criteria was used to fit the pair copulas, or how you assigned independence to some of the par-copulas. There are references to the R-package, however this is not enough, also considering that in the package different approaches for fit can be used.

I add the following paragraph in order to give more information about the fitting choices:

"The copula family types for each selected pair in the first tree are determined by using the Akaike information criterion (see Brechmann and Schepsmeier, (2013)). For computational reasons, the two-parameter Archimedean copulas are excluded from this analysis (which however has only a negligible impact in the results, see Figure 5 A1 of the Appendix). The copula parameters are estimated sequentially (using maximum likelihood estimation) starting from the top tree until the last tree, as described in Czado et al. (2013). This approach only involves estimation of bivariate copulas and has been chosen since it is computationally much simpler than joint maximum likelihood estimation of all parameters at once."

**Methodology**

P1 l1: the third coldest winter ever recorded. Where and according to what criteria?

I will update the abstract to make this clearer. This is according to the Central England Temperature (CET) record, the oldest continuously running temperature dataset in the world (Manley, 1974), only two other winters (1683/84 and 1739/40) have been colder than 1962/63 in the last 350 years (also mentioned in the submitted manuscript).

P3 l5 It is based on rigorously quality checked station data interpolated to a regular grid using inverse-distance weighting, as described in Perry et al. (2009). It should be mentioned here or later that therefore the dependencies catched by the copulas may be partially due to the interpolation itself.
Agreed

P3 l10 Nevertheless, local temperature may be subtly different in certain micro-climates, such as upland and urban regions. I would mention that however the resolution 5km x 5km may not always be realistic, depending on the number of stations which were available for the creation of the data set.

I agree. However, notice that I re-grid the data to a lower resolution of 50 x 50 km2. The station network contains 540 stations with an average density of 21 x 21 km2, with also more stations near urban areas (see Figures 1 and 2 in Perry and Hollis, 2005).

P3 l29 98.3°C. Based on line 17, I expected negative values for the AFI. Could you mention that you take the absolute values of the temperature? Also, it would be appreciated if you would show the equation of the AFI.

That's correct. The exact equation is

$$AFI = \sum_{day=1/6/Year}^{31/5/(Year+1)} |T_{day}| \qquad if\ T_{day} < 0°C$$

P3 l32 After 1962/63, a long run of mild winters followed until late 1978 and early 1979 (Figure 2). Is this in Figure 2 the AFI averaged over UK? Please, use °C in the y label of Fig. 2.

Yes, that's has been corrected.

P5 l4 An additional term was included, the probability of no hazard (P0), in order to account for the cells mainly on the south England coast that have years with no negative temperatures at all.

1) Does this mean that for some cells the GEV is fitted on very few data? Please give information about this, and on the goodness of the fit for these cells.

In order to improve the fits at those cells, I applied a geographical smoothing of the GEV parameters as well; I had erroneously missed to discuss this part from the submitted text. I will update the text to reflect in detail this methodology.

More precisely, along with the TWMLE method described in the text, I applied a second modification in order to geographically smooth the GEV parameters. The smoothing is incorporated into the fitting process by minimizing the local (ranked) log-likelihood. More precisely, the log-likelihood at each grid cell *i* is calculated using all grid points but weighted by their distance $d_{ij}$:

$$LogL_i = \sum_{j=1}^{170}(w_{ij} \cdot LogL_j), \text{ where } w_{ij} = \frac{1}{\sqrt{2\pi}}e^{-\frac{d_{ij}^2}{2L^2}}$$

where *L* here is the length scale or smoothing parameter and $LogL_j$ is the ranked log-likelihood for cell j. Because the historical gridded data are already geographically smoothed, I decided to use a small length scale parameter *L* of 15 km (in comparison to the 50km grid size).

In general, the increase of the sample size at each grid point allows for a more precise estimation of the parameters, especially for the shape parameter which is highly influential in estimating the hazard levels and high return periods. This methodology also permits the estimation of the parameters in cells with no data.

The parameters at each cell are shown in Table 3 (at the end of this document) and will also be included in the final form of the manuscript. I' also attaching a pdf which contains the GEV fits for all the 170 cells of the domain (gevPlots.pdf).

2) Please, specify how P0 is estimated, e.g. N_occurence/N_years. FIG 3

P0 = N_occurrence / (Nyears + 1)

1) I assume that the " historical AFI GEV fit (black circles) " is the empirical estimate. If yes, is this computed as written in P6 L5? Please, specify this.

Yes, that's is correct. I will make this clear in the caption.

2) Could you specify the estimated parameters, or also only making clear to the reader whether the difference is due to the selection of different family type (Gumbel, Frechet, and Weibull distributions)?

The difference between the two fitted curves in Figure 3 of the submitted manuscript is the different methodology: MLE vs. the selected TWMLE which gives more weight to the higher order observations as described in the text.

P5 l20 As an example, the GEV fit for a single cell over London is shown in Figure 3. The grey line represents the GEV fit without any weighting applied, while the black curve is estimated using the TWMLE method with an improved fit towards the tail of the distribution (i.e. the more extreme events).I would rather say that you get a curve that is nearer to the empirical estimate.

I agree with the reviewer and I will update the text.

P6 l2 Other urban regions (e.g. Manchester or Midlands area) do not stand out as much as a result of the low grid resolution. Can this also be due to the original data format? For example there may be not enough stations around some urban areas.

Figure 3 shows the AFI values for the (empirically estimated) 50 years return period based on the historical data at their original resolution (5km x 5km). Apart from the London area, other cities (e.g. Liverpool, Manchester) stand out with lower values than their surroundings. It is therefore mainly the re-gridding process to 50x50km (necessary for computing reasons) that masks the cities in this study's results.

[Figure]

Figure 3: High resolution maps of AFI values (in ∘C) for return period of 50 years.

P9 l8 At the first level, 49% of the selected bivariate copulas are found to be Gumbel which implies greater dependence at larger AFI values. You refer to the tail dependence, I assume. Make it more clear, please. Greater with respect to what?

I meant greater with respect to the low AFI values, so indeed I refer to the tail dependence.  I will rephrase it to make it more clear.

P9 l13 The RVM is used to simulate 10,000 years of winter-seasons in the UK. This amount of realisations should be long enough in order to estimate with enough confidence the 200 year RP hazard, which is commonly associated with capital and regulatory requirements.

The 10,000 years time series should be long enough to neglect uncertainties associated with the Monte Carlo simulations (which is the method used for extracting the return period associated with the fitted parametric pdf) (Serinaldi et al. (2015) and Bevacqua et al. (2017)). One should ensure if the sample is "long enough" via repeating the (10,000 years) simulations several times and checking if the there are differences in the estimated return period (if there are no differences, the 10,000 years sample is long enough). Performing a long enough simulations allows one to get a convergence to the true return period that one would get analytically from the fitted pdf (given the complexity of the problem it is impracticable to get an analytical derivation of the RP). Performing a long simulation does not solve the issue about the model uncertainties (uncertainties existing about the pdf), which is there because the pdf is calibrated on a finite - very short - sample. I suggest to discuss this in a way to make difference between these different type of uncertainties.

I agree with this comment. Please refer to my answer at the main comment where I discuss both uncertainties.

P9 l 27 The exceedance probability (EP) curve of wAFI is shown in Figure 7, both for the historical and the stochastic data. So far you talked about RP. Personally, I think that it would be better to keep the same terminology instead of introducing EP, or at least use also RP here.

I agree with the reviewer and I will update the text to reflect that.

P9 l27 The uncertainty intervals in the historical data are computed as the 5th and 95$^{th}$ quantile of the probability density function (Folland and Anderson, 2002). I suggest to use: "The uncertainty intervals in the return period (estimated empirically?) of the historical data are computed via the 5th and 95th quantile of the probability density function" Agreed.

P10 l2 low tail dependence. Gaussian and Frank copula have zero tail dependence, not "low". It may be helpful to better introduce the tail dependence in a sentence where you talk about it for the first time. Agreed. I will introduce the tail dependence definition and functions also in conjunction with the reply of the following question.

P10 l2 On the other hand, the low impact of the other copula families is due to the fact that the extreme hazard values are mainly driven by the large dependencies between nearby cells, especially at the first tree levels. Could you please argue this better?

Unfortunately I found an error in Figure 1 and 2 of the Appendix in the submitted manuscript: the curve based on the Clayton RVM was computed erroneously. It is corrected as shown in Figure 4a below. So, Gaussian, Frank and Clayton copulas now show differences in comparison to the default RVM. This is to be expected since all three of those copulas do not show right tail dependence in the limits. In

particular, away from the extremes, normal shows greater right tail concentration than Frank and Clayton copulas (see Figure 4b).

[Figure]

[Figure]

Figure 4a: Sensitivity tests for the weighted AFI RP over the entire U.K. based on RVM fitted using: all but the two-parameter Archimedean copulas (black line), only one Copula family each time, i.e. Gaussian (blue line line), Student's t (pink), Clayton (light blue), Gumbel (orange), Frank (grey), and Joe (green).

Figure 4b: Lower and upper tail dependence plot for the Gauss, Gumbel, Clayton, Joe, and Frank copulas, assuming a Kendall tau value of 0.5.

Concerning the argument that the large dependencies drive the extreme hazard values, I believe that this is to be expected: the co-occurrence of extreme AFI values in many cells (which lead to large regionally averaged AFI) is mainly driven by large dependencies among the cells.

 In order to demonstrate the large influence of the larger dependencies in the model, I manually adjust the copula parameters in order to test the sensitivity of the very large ($\tau \geq 0.5$) vs. the lower ($\tau < 0.5$) dependencies. For simplicity select a test RVM using only Gumbel (or independent) copula families (RVM04), which however matches relatively well the results from the default as shown in Figure 5 (and also described in the manuscript). I manually adjust RVM04 by reducing the Gumbel alpha parameters (at all levels) such as the new corresponding Kendall's tau ($\tau$) value would be very small (equal to 0.01). This is simple for the Gumbel copula since the Gumbel copula parameter $\alpha$ is given by: $\alpha = 1 / (1 - \tau )$. I adjust separately the copulas with very large dependencies (with $\tau \geq 0.5$ or $\alpha \geq 2$) and those with lower ($\tau$

< 0.5 or α < 2) dependencies and the resulting RP curves are shown with the blue and orange lines, respectively.

[Figure]

Figure 5: RP curves of weighted for the whole UK. The empirical EP is shown in grey. The black line shows the curve based on the default RVM while the red line is based on a RVM fitted only with Gumbel copula families. The orange and blue lines represent the curves after manually adjusting the selected Gumbel parameters as described in the text.

P10 l16 However, recent studies suggest that cold weather in the UK is likely to be less severe, to occur less frequently, and to last for a shorter period of time than was historically the case due to anthropogenic induced climate change (on Climate Change, 2017). I would already mention here that there is debate about this (as you then specify in the next paragraph).

Agreed

P11 l7 As shown in Figure 8, South England is in general warmer than the North England and Northern Ireland region, partially driven by the urban micro-climate effect of the London area. The 1962/63 winter was less extreme in this region (wAFI of 139° C) with an estimated return period of 1 in 79 years. On the other hand, Scotland is usually significantly colder than the rest of UK, reaching for example AFI values of 100 ∘ C almost 2 times more often.

Please, make more clear in the text (and in the figure captions) when you talk about AFI, wAFI, averaged non-weighted AFI (and in which area is computed the average (UK, or sub-regions)). Also, when introducing eq. (3), I suggest to anticipate that you are going to use the wAFI both on UK and subregional scale.

Agreed

Figure captions. Please improve the Figure captions with more information. For example in Fig 2 what is the NAOI (North Atlantic Oscillation Index)?

Agreed

**Technical corrections**

P3 l6 desribed. Described

P4 l1 that winter . You may use "winter 1978/79".

Fig 4 and 5. Could you please use the same scale range, i.e. 0-400°C

P10 l2 familes. Families

Agreed.

**REVIEW 2**

**GENERAL COMMENTS.**

The paper is an interesting one, and outlines an original multivariate investigation concerning subfreezing temperatures. The comments posted by Referee 1 already provide an excellent, detailed review, with which I (almost) fully agree. Below, please find further notes: my objections should be read as constructive advices. Some relevant bibliography is reported at the end of this review.

1. I noticed that there is some confusion between the notions of probability distribution function and probability density function (e.g., Page 10, Lines 5–7: "The uncertainty intervals in the historical data are computed as the 5th and 95th quantile of the probability density function (Folland and Anderson, 2002)"). The probability distribution function is the integral of the probability density function (if it exists). The quantiles are the inverses of the probability distribution function (a nondecreasing one), not of the density function (which may not even be monotone). The Author must check the paper and fix all the points where such a confusion arises, otherwise the paper is not correct from a probabilistic point of view. Agreed.

2. I was puzzled by the comment of Referee 1 concerning the sample size, and I ask the Author to clarify the issue: here, 170 variables are at play, each observed 51 times. To the best of my understanding, the idea revolving around Vine copulas is that any multivariate density can be decomposed into a (suitable, maybe not unique) product of univariate marginal densities and bivariate copula densities: in turn, only univariate and bivariate fits should be needed, isn't it? Thus, apparently, the fitting problem may not be so severe. Clearly, trying and fitting the upper tail of a GEV law using only 51 observations may be difficult (although the TWMLE escamotage is used), but it may not be impossible. Similarly, trying and fitting a bivariate copula using only 51 pairs may not be advisable, but it is not uncommon in practical applications. Overall, should my interpretation be correct, the game played by the Author may not be a "Mission Impossible", rather an "Uncertain Mission". . . Thus, I kindly ask the Author to clearly explain the situation, and to provide estimates of the uncertainties as explained below.

This question is related to the main comment of the first reviewer. Indeed there are indeed 170 variables each observed 51 times. Please see my answer to the main comment of the first reviewer at the beginning of this document.

3. I definitely agree with the comment of Referee 1 concerning the procedure to estimate the uncertainties (Page 9, Lines 23–ff.). As a rule of thumb, 1000 independent repetitions of the 10,000-years Monte Carlo simulations are usually suggested in literature, in order to provide "reasonable(?)" estimates of the confidence intervals of interest (clearly, it may be adjusted depending on the computational burden).

I have re-simulated 1000 independent repetitions of the 10K years Monte Carlo simulations as suggested by the reviewer. I used these repetitions to construct confidence intervals: please see my answer to the main comment of the first reviewer at the beginning of this document.

4. My main "perplexity" concerns a methodological issue. In this work, I can see the Mathematics/ Statistics, but I do not see the Physics, which, instead, should be the starting point. To be clear, and to the best of my knowledge, the procedure used to construct the 170-dimensional copula finds its justification in an aggregation/clustering algorithm based solely on statistical considerations (Page 9, Lines 13–14: "The method follows an automatic strategy of jointly searching for an appropriate R-Vine tree structure"). If I remember it correctly, the algorithm is based on the Kendall  and/or on the Kendall Distribution Function K, and/or, in general, on the strength of the statistical association between the variables at play. While interesting and meaningful from a mathematical point of view, such a procedure may eventually (statistically) associate grid cells having little, or negligible, physical link (for instance, could this be the case of the grid cells corresponding to Edinburgh and London, quite far apart from a spatial and a climatic point of view?) In other words, important information like, e.g., the latitude (corresponding to different climatic regions) may not be considered/used by the statistical procedure adopted for constructing the overall copula. The Author is kindly asked to discuss the issue, and to provide suitable justifications. Is it possible to modify the construction of the 170-dimensional copula in order to take into account the physics of the phenomenon?

I believe that the physics are captured via the large scale circulation, which is the driver of winter temperatures in UK, and which is modeled through the multivariate joint distribution. I don't agree with the reviewer with respect to the association of distant locations: winter temperatures are usually coherent across large part of UK as they are primarily associated with large-scale atmospheric circulation (Scaife and Knight, 2008). Notice that AFI assess the freezing temperatures at an annual temporal resolution, i.e. during the entire winter season and, thus, is less associated with small scale local phenomena which can cause differences between locations. This can be seen in Figure 6, where the 51-year long observed AFI values over London correlate significantly with all the remaining UK cells with linear correlation coefficients above 0.5. The vine copula model's role is to capture these large scale temperature associations across the UK.

Moreover, the dominant large scale mode of variability in the Euro-Atlantic region is the North Atlantic Oscillation, which I tried to include (not in the copula but) in the GEV fits (see section 3.1, page 12, lines 11-25); however, including it was not improving the model fits, possibly due to the quite noisy character of the phenomenon and the relatively short historical record used in this study.

Recently, Bevacqua et al (2017a, 2017b) has developed an R package called CDVineCopulaConditional (a reference suggested by the first reviewer in fact) which offers the possibility to include meteorological predictors for the contributing variables; including for example NAOI would be something that I would like to include in the future.

[Figure]

Correlation map

Figure 6: Linear correlation map between the 51-year long AFI values in a cell over London (blue dot) and all other cells of the model domain.

5. The Author has modeled the historical data, but, should the climate be really changing, then (at least from an Insurance point of view) the Author should account for it in his model, e.g. by introducing (in the long term simulations) suitable temporal patterns in the GEV/copula parameters according to available projections of the future climate (like, e.g., in IPCC scenarios). A comment is required on this issue.

From an Insurance point of view, the focus is mainly in the next year or sometimes a bit longer (but up to 4 years) depending on the (re) insurance contracts. Therefore, the long term changes in the climate based on future climate projections are not much of interest.

However, what is important is that the model here has been constructed under the assumption of a stationary climate, i.e. under the assumption that the climate has not changed (significantly) during the last 51 years. Although not mentioned in the original manuscript, I had tested this assumption in a similar method as for NAO (see section 3.1): I incorporated a GLM into the statistical distribution parameter estimates of the GEV fits, by defining the location parameter as a function of the year: $\mu = \beta_0 + \beta_1 \times year$. However, this was not improving the model fits (the beta parameters were not statistically different from zero) and thus it has not been included it to the final model.

6. In Section 3.1 "Results and discussion", the Author mentions the actual debate about climate changes (already commented by Referee 1). I would suggest to take a look at a recent paper by Vezzoli et al. (2017), where the traditional validation criteria of climate models are discussed, and an advanced/ thorough distributional perspective is outlined: it may partially explain why several crucial hypotheses are "still largely under debate" (as claimed by the Author and Referee 1), and may partially account for the general inability to draw up clear settlements.

Thank you I take on board this comment.

**SPECIFIC COMMENTS.**

Page(s) 2, Line(s) 23–ff.

For the benefit of unskilled readers and practitioners, here the Author should provide general references involving seminal books, papers, and guidelines concerning copulas, like writing: "For a theoretical introduction to copulas, see Nelsen (2006); Joe (2014); Durante and Sempi (2015); for a practical/engineering approach and guidelines, see Genest and Favre (2007); Salvadori and De Michele (2007); Salvadori et al. (2007, 2014, 2015)". Instead, citations concerning Vine copulas, being more specific and related to the modeling outlined in this work, may be postponed later. Page(s) 9, Line(s) 20–22.

I agree with the reviewer and I will update the manuscript as suggested.

Author. "Goodness-of-fit is performed for the final selected R-Vine Model (RVM) based on the RVineGofTest algorithm of the same R package (Schepsmeier, 2013). The Cramer von Mises test, which compares the empirical copula with the RVM, has a value of 0.019 and a p.value = 1, which indicates that the fitted RVM cannot be rejected at a 5% significance level."

Referee. I am puzzled by such a large p-Value: in my opinion, it may entail a large probability of Type II error, i.e. accepting a False Null Assumption (this a typical performance of Cramer-von- Mises and similar tests, when the sample size is insufficient). The Author is kindly asked to discuss the issue, and to provide suitable justifications.

Unfortunately, the gof tests for Vine copulas show poor behavior in small sample sizes and also at higher dimensions, as is the case for this work (Schepsmeier, 2013). I should also mention that different gof methods implemented in the VineCopula R package (via the function RVineGofTest) can show very different values for the CvM statistic as shown in the table 2 below. I have also tried to include a range in the GoF test values by resampling the historical AFI years (sampling 51 years with replacement x 10 times). Due to the long runtime, it has not been possible to sample more times; the min/max ranges are shown in parenthesis for the ECP2 and ECP methods in the table below. It is even harder to estimate the power of these tests. Thus, my main justification would be the performance of the model in comparison to the empirical estimates: as shown in Figures 7 and 8 of the submitted manuscript, the simulated RPs follow reasonable well the empirical (historical) estimates not only at the entire UK but also at smaller regions.

**Table 2:** GOF valus for the CvM statistic based on different methods implemented in the VineCopula R package

| GoF method | CvM | p.val |
|---|---|---|
| ECP2 | 0.019 (0.019-0.15) | 1 (0.95-1) |
| ECP | 2.3 (2.1-4.3) | 0.73 (0.61-0.73) |
| Breymann | 1.8 | 0.185 |
| Berg2 | 0.19 | 1 |
| Berg | 1.11 | 1 |
| White | not enough memory | |
| IR | not enough memory | |

**References**

Bevacqua, E., Maraun, D., Hobæk Haff, I., Widmann, M., and Vrac, M.: Multivariate statistical modelling of compound events via pair-copula constructions: analysis of floods in Ravenna (Italy), Hydrol. Earth Syst. Sci., 21, 2701-2723, https://doi.org/10.5194/hess-21-2701-2017, 2017a.

Bevacqua E., CDVineCopulaConditional: Sampling from Conditional C- and D-Vine Copulas. R package version 0.1.0, 2017b

Bowman, A. W. and Azzalini A., Applied smoothing techniques for data analysis, Oxford University Press, London, 1997

Brechmann, E. C. and Schepsmeier, U.: Modeling Dependence with C- and D-Vine Copulas: The R Package CDVine, Journal of Statistical Software, doi:10.18637/jss.v052.i03, https://www.jstatsoft.org/article/view/v052i03, 2013.

Czado, E., Brechmann E. C., and Gruber, I.: Selection of Vine Copulas. In: P. Jarkowski, F. Durante, and W. Haerdle, (editors). Copulae Mathematical and Quantitative Finance, 16, 775-787, 2013.

Perry, M. and Hollis, D., The generation of monthly gridded datasets for a range of climatic variables over the UK. Int. J. Climatol., 25: 1041–1054. doi:10.1002/joc.1161, 2015.

Scaife, A. A. and Knight, J. R.: Ensemble simulations of the cold European winter of 2005-2006, Quarterly Journal of the Royal Meteorological Society, 134, 1647–1659, doi:10.1002/qj.312, http://dx.doi.org/10.1002/qj.312, 2008.

Schepsmeier, U.: Estimating standard errors and efficient goodness-of- t tests for regular vine copula models, Ph.D. thesis, Fakultat fur Mathematik Technische Universitat Munchen, http://mediatum.ub.tum.de/doc/1175739/document.pdf, 2013.

| cellid | location | scale | shape | P0 | cellid | location | scale | shape | P0 |
|---|---|---|---|---|---|---|---|---|---|
| 1 | 0.89 | 2.16 | 0.02 | 0.81 | 86 | 4.13 | 11.5 | 0.22 | 0.13 |
| 2 | 0.86 | 5.09 | -0.02 | 0.58 | 87 | 5.21 | 9.29 | 0.46 | 0.15 |
| 3 | 1.28 | 4.89 | 0.13 | 0.5 | 88 | 4.28 | 11.71 | 0.29 | 0.08 |
| 4 | -0.23 | 6.3 | -0.03 | 0.54 | 89 | 3.51 | 8.35 | 0.3 | 0.23 |
| 5 | 0.53 | 6.32 | 0.02 | 0.54 | 90 | 2.22 | 4.15 | 0.16 | 0.56 |
| 6 | 3.64 | 4.02 | 0.46 | 0.48 | 91 | 1.36 | 6.05 | 0.01 | 0.44 |
| 7 | 2.98 | 7.09 | 0.35 | 0.29 | 92 | 0.49 | 9.84 | 0.04 | 0.21 |
| 8 | 4.48 | 9.23 | 0.36 | 0.23 | 93 | 14.28 | 25.62 | 0.02 | 0.02 |
| 9 | 5.13 | 7.84 | 0.45 | 0.33 | 94 | 35.76 | 39.95 | 0 | 0 |
| 10 | 4.32 | 8.43 | 0.43 | 0.21 | 95 | 10.68 | 21.56 | 0.16 | 0.02 |
| 11 | 4.57 | 6.13 | 0.52 | 0.35 | 96 | 5.76 | 16.46 | 0.16 | 0.08 |
| 12 | 3.03 | 6.58 | 0.41 | 0.42 | 97 | 5.1 | 9.74 | 0.46 | 0.1 |
| 13 | 0.67 | 10.85 | 0.21 | 0.37 | 98 | 3.24 | 11.8 | 0.18 | 0.17 |
| 14 | 2.75 | 10.5 | 0.22 | 0.42 | 99 | 1.43 | 13.42 | 0.01 | 0.23 |
| 15 | 3.29 | 6.68 | 0.47 | 0.29 | 100 | 2.5 | 3.73 | 0.06 | 0.6 |
| 16 | 5.64 | 10.84 | 0.39 | 0.15 | 101 | 2.52 | 22.72 | -0.07 | 0.06 |
| 17 | 5.37 | 10.22 | 0.42 | 0.15 | 102 | 14.03 | 27.17 | 0.01 | 0 |
| 18 | 6.95 | 13.14 | 0.37 | 0.04 | 103 | 14.94 | 23.95 | 0.12 | 0 |
| 19 | 5.49 | 12.45 | 0.34 | 0.04 | 104 | 25.7 | 35.93 | 0.02 | 0 |
| 20 | 6.38 | 13.92 | 0.29 | 0.08 | 105 | 6.5 | 16.57 | 0.14 | 0.08 |
| 21 | 5.79 | 13.5 | 0.27 | 0.08 | 106 | 0.94 | 3.86 | 0.04 | 0.58 |
| 22 | 6.59 | 13.68 | 0.27 | 0.13 | 107 | 2.51 | 7.08 | 0.01 | 0.38 |
| 23 | 3.38 | 11.79 | 0.28 | 0.1 | 108 | 2.74 | 19.94 | -0.06 | 0.1 |
| 24 | 3.1 | 3.85 | 0.37 | 0.5 | 109 | 30.46 | 44.12 | -0.12 | 0 |
| 25 | 3.66 | 4.95 | 0.3 | 0.5 | 110 | 40.49 | 43.29 | -0.01 | 0 |
| 26 | 4.34 | 9.32 | 0.35 | 0.27 | 111 | 22.2 | 30.8 | 0.09 | 0 |
| 27 | 4.56 | 10.77 | 0.4 | 0.15 | 112 | 6.83 | 19.49 | 0.09 | 0.08 |
| 28 | 7.7 | 13.55 | 0.37 | 0.08 | 113 | 1.98 | 3.2 | 0.3 | 0.54 |
| 29 | 10.6 | 17.25 | 0.32 | 0.02 | 114 | 1.97 | 9.96 | 0.05 | 0.23 |
| 30 | 8.7 | 15.98 | 0.28 | 0.02 | 115 | 3.92 | 22.25 | -0.07 | 0.04 |
| 31 | 5.66 | 11.51 | 0.31 | 0.13 | 116 | 11.66 | 25.44 | 0.03 | 0 |
| 32 | 5.33 | 12.05 | 0.28 | 0.12 | 117 | 11.47 | 26.13 | 0.02 | 0 |
| 33 | 5.19 | 9.46 | 0.36 | 0.27 | 118 | 8.01 | 23.44 | 0 | 0.02 |
| 34 | 2.59 | 7.28 | 0.26 | 0.37 | 119 | 1.64 | 2.41 | 0.05 | 0.62 |
| 35 | 3.74 | 10.93 | 0.31 | 0.12 | 120 | 2.98 | 5.56 | 0.02 | 0.44 |
| 36 | 9.46 | 17.98 | 0.26 | 0.02 | 121 | 3.57 | 16.56 | -0.05 | 0.08 |
| 37 | 11.58 | 19.04 | 0.3 | 0.02 | 122 | 33.2 | 45.78 | -0.16 | 0 |
| 38 | 8.81 | 14.29 | 0.37 | 0.06 | 123 | 35.49 | 50.13 | -0.16 | 0 |
| 39 | 11.63 | 18.16 | 0.31 | 0.02 | 124 | 12.59 | 28.93 | -0.02 | 0.02 |
| 40 | 10.62 | 18.28 | 0.3 | 0.02 | 125 | 3.74 | 13.91 | 0.04 | 0.1 |
| 41 | 7.95 | 16.99 | 0.31 | 0.02 | 126 | 1.51 | 1.57 | 0.08 | 0.79 |
| 42 | 7.97 | 15.11 | 0.31 | 0.06 | 127 | 2.69 | 4.81 | 0.02 | 0.46 |
| 43 | 5.84 | 10.18 | 0.34 | 0.19 | 128 | 12.21 | 23.27 | -0.08 | 0.06 |
| 44 | 3.66 | 8.69 | 0.33 | 0.15 | 129 | 47.6 | 53.04 | -0.2 | 0 |
| 45 | 12.45 | 21.01 | 0.21 | 0 | 130 | 95.88 | 71.01 | -0.17 | 0 |
| 46 | 14.78 | 22.14 | 0.28 | 0.02 | 131 | 84.57 | 67.4 | -0.16 | 0 |
| 47 | 10.35 | 17.45 | 0.32 | 0.04 | 132 | 12.4 | 31.34 | -0.1 | 0 |
| 48 | 10 | 18.18 | 0.28 | 0.02 | 133 | 1.92 | 3.19 | 0.04 | 0.52 |
| 49 | 11.52 | 19.94 | 0.29 | 0.02 | 134 | 4.93 | 10.12 | -0.01 | 0.15 |
| 50 | 8.63 | 15.95 | 0.34 | 0.04 | 135 | 9.63 | 19.48 | -0.06 | 0.02 |
| 51 | 8.49 | 15.65 | 0.32 | 0.02 | 136 | 56.35 | 57.26 | -0.22 | 0 |
| 52 | 6.4 | 13.47 | 0.3 | 0.08 | 137 | 63.36 | 66.14 | -0.26 | 0 |
| 53 | 1.78 | 8.65 | 0.24 | 0.19 | 138 | 63.05 | 71.33 | -0.28 | 0 |
| 54 | 1.99 | 6.11 | 0.15 | 0.48 | 139 | 17.71 | 34.76 | -0.12 | 0.02 |
| 55 | 9.48 | 19.26 | 0.22 | 0 | 140 | 3.05 | 17.2 | -0.05 | 0.04 |
| 56 | 10.23 | 18.48 | 0.29 | 0.02 | 141 | 2.81 | 3.65 | 0.04 | 0.44 |
| 57 | 10.31 | 18.12 | 0.31 | 0.02 | 142 | 2.68 | 5.45 | 0.02 | 0.23 |
| 58 | 10.51 | 17.81 | 0.28 | 0.04 | 143 | 6.83 | 14.39 | -0.04 | 0.02 |
| 59 | 10.55 | 17.28 | 0.3 | 0.04 | 144 | 46.4 | 50.68 | -0.19 | 0 |
| 60 | 8.5 | 15.03 | 0.32 | 0.06 | 145 | 17.76 | 32.1 | -0.11 | 0 |
| 61 | 7.06 | 13.64 | 0.32 | 0.06 | 146 | 12.93 | 28.71 | -0.1 | 0.02 |
| 62 | 4.32 | 10.41 | 0.34 | 0.1 | 147 | 8.22 | 22.66 | -0.07 | 0.04 |
| 63 | 1.6 | 9.04 | 0.18 | 0.25 | 148 | 4.11 | 10.89 | -0.02 | 0.19 |
| 64 | 0.25 | 8.05 | 0.07 | 0.46 | 149 | 0.87 | 1.54 | 0.52 | 0.54 |
| 65 | 4.36 | 14.45 | 0.19 | 0.12 | 150 | 2.98 | 7.33 | 0 | 0.25 |
| 66 | 4.76 | 13.42 | 0.31 | 0.08 | 151 | 2.57 | 4.06 | 0.03 | 0.33 |
| 67 | 6.62 | 15.07 | 0.25 | 0.04 | 152 | 21.01 | 32.56 | -0.11 | 0 |
| 68 | 15.97 | 22.09 | 0.23 | 0.04 | 153 | 42.9 | 46.77 | -0.17 | 0 |
| 69 | 8.33 | 15.51 | 0.3 | 0.04 | 154 | 19.89 | 28.24 | -0.1 | 0 |
| 70 | 6.9 | 15.12 | 0.26 | 0.04 | 155 | 2.18 | 7.34 | 0.01 | 0.23 |
| 71 | 4.11 | 8.48 | 0.25 | 0.19 | 156 | 2.4 | 6.37 | 0.04 | 0.23 |
| 72 | 2.89 | 12.78 | 0.25 | 0.1 | 157 | 6.8 | 18.93 | -0.06 | 0.02 |
| 73 | 9.13 | 19.5 | 0.17 | 0.02 | 158 | 10.26 | 22.43 | -0.07 | 0.02 |
| 74 | 9.53 | 16.07 | 0.3 | 0.04 | 159 | 7.81 | 12.78 | -0.03 | 0.1 |
| 75 | 7.16 | 13.97 | 0.32 | 0.08 | 160 | 2.86 | 5.06 | 0.12 | 0.21 |
| 76 | 5.33 | 10.35 | 0.28 | 0.17 | 161 | 1.24 | 2.35 | 0.06 | 0.27 |
| 77 | 4.28 | 10.04 | 0.38 | 0.13 | 162 | 1.75 | 3.2 | 0.12 | 0.27 |
| 78 | 4.59 | 9.34 | 0.33 | 0.19 | 163 | 1.06 | 1.23 | 0.23 | 0.33 |
| 79 | 4.28 | 7.99 | 0.23 | 0.31 | 164 | 2.72 | 4.03 | 0.03 | 0.23 |
| 80 | 2.46 | 3.75 | 0.25 | 0.54 | 165 | 3.75 | 5.06 | 0.02 | 0.1 |
| 81 | 2.35 | 4.23 | 0.03 | 0.48 | 166 | 2.95 | 3.96 | 0.03 | 0.13 |
| 82 | 3.48 | 16.01 | 0.11 | 0.06 | 167 | 5.23 | 9.05 | -0.01 | 0.06 |
| 83 | 18.15 | 30.48 | 0.06 | 0 | 168 | 4.42 | 6.88 | 0.01 | 0.08 |
| 84 | 15.16 | 20.59 | 0.26 | 0.02 | 169 | 6.42 | 9.35 | -0.01 | 0.06 |
| 85 | 9.4 | 16.72 | 0.26 | 0.02 | 170 | 6.35 | 9.53 | -0.01 | 0.06 |

Table 3: List of the GEV parameters, including P0 for each cell of the domain.

Table 4: List of the percentage of all selected copula families for each level of the RVM tree.

| Tree level | independence copula | Gaussian copula | Student t copula | Clayton copula | Gumbel copula | Frank copula | Joe copula | Clayton copula (180 degrees) | Gumbel copula (180 degrees) | Joe copula (180 degrees) | Clayton copula (90 degrees) | Gumbel copula (90 degrees) | Joe copula (90 degrees) | Clayton copula (270 degrees) | Gumbel copula (270 degrees) | Joe copula (270 degrees) |
|---|---|---|---|---|---|---|---|---|---|---|---|---|---|---|---|---|
| 1 | 0 | 9.5 | 13 | 0 | 48.5 | 11.8 | 7.7 | 5.3 | 2.4 | 1.8 | 0 | 0 | 0 | 0 | 0 | 0 |
| 2 | 35.1 | 8.3 | 6.5 | 4.2 | 7.7 | 5.4 | 4.2 | 9.5 | 1.8 | 3 | 3 | 0.6 | 1.2 | 4.8 | 1.8 | 3 |
| 3 | 43.7 | 3 | 3.6 | 6 | 4.2 | 16.2 | 3 | 5.4 | 1.8 | 1.2 | 2.4 | 0.6 | 2.4 | 3.6 | 1.2 | 1.8 |
| 4 | 40.4 | 4.8 | 8.4 | 1.4 | 5.4 | 6.6 | 2.4 | 4.2 | 3 | 4.2 | 1.8 | 2.4 | 4.2 | 2.4 | 1.8 | 3.6 |
| 5 | 48.5 | 7.3 | 3 | 1.2 | 1.8 | 10.3 | 6.1 | 4.8 | 3 | 1.8 | 0.6 | 0 | 3.6 | 3.6 | 1.2 | 3 |
| 6 | 51.2 | 6.7 | 3 | 4.3 | 0.6 | 7.3 | 4.3 | 3 | 0.6 | 3 | 3.7 | 0 | 4.3 | 1.8 | 2.4 | 3.7 |
| 7 | 53.4 | 4.3 | 2.5 | 2.5 | 5.5 | 6.7 | 4.9 | 4.3 | 1.8 | 3.7 | 4.3 | 0.6 | 0.6 | 1.2 | 1.2 | 2.5 |
| 8 | 58.6 | 3.1 | 3.7 | 1.2 | 3.1 | 3.1 | 4.3 | 1.9 | 3.7 | 5.6 | 2.5 | 0.6 | 3.1 | 0.6 | 1.9 | 3.1 |
| 9 | 60.9 | 3.7 | 2.5 | 4.3 | 0.6 | 5.6 | 3.1 | 3.7 | 1.2 | 1.2 | 3.7 | 0 | 3.7 | 2.5 | 1.2 | 1.9 |
| 10 | 57.5 | 3.1 | 1.9 | 3.1 | 1.2 | 9.4 | 6.2 | 2.5 | 2.5 | 3.8 | 3.1 | 1.2 | 1.9 | 0.6 | 0.6 | 1.2 |
| 11 | 61 | 1.3 | 3.1 | 5.7 | 0.6 | 5 | 3.1 | 3.8 | 1.9 | 2.5 | 5.7 | 1.3 | 1.3 | 1.9 | 0.6 | 1.3 |
| 12 | 52.5 | 5.1 | 5.1 | 5.1 | 1.3 | 5.1 | 2.5 | 6.3 | 2.5 | 3.2 | 2.5 | 1.9 | 0 | 2.5 | 1.9 | 2.5 |
| 13 | 56.1 | 6.4 | 1.9 | 0.6 | 1.3 | 9.6 | 4.5 | 1.9 | 5.1 | 3.2 | 2.5 | 1.3 | 1.3 | 2.5 | 0 | 1.9 |
| 14 | 57.7 | 5.1 | 3.8 | 1.3 | 0.6 | 5.1 | 4.5 | 1.3 | 1.3 | 4.5 | 3.2 | 2.6 | 3.8 | 1.9 | 0.6 | 2.6 |
| 15 | 63.2 | 6.5 | 1.3 | 1.9 | 2.6 | 1.9 | 3.2 | 4.5 | 0.6 | 2.6 | 2.6 | 1.3 | 2.6 | 1.9 | 0.6 | 2.6 |
| 16 | 63 | 7.1 | 1.3 | 3.2 | 1.9 | 5.2 | 1.9 | 1.3 | 0.6 | 3.2 | 0 | 1.3 | 2.6 | 2.6 | 1.9 | 2.6 |
| 17 | 58.8 | 5.9 | 2 | 3.9 | 1.3 | 5.9 | 2 | 1.3 | 1.3 | 5.2 | 1.3 | 1.3 | 2.6 | 3.3 | 2.6 | 1.3 |
| 18 | 66.4 | 5.3 | 3.9 | 3.9 | 2 | 2.6 | 2 | 4.6 | 0.7 | 1.3 | 4.6 | 0 | 0.7 | 0 | 1.3 | 0.7 |
| 19 | 62.3 | 4 | 3.3 | 2 | 1.3 | 6 | 1.3 | 4.6 | 0.7 | 2.6 | 5.3 | 0 | 1.3 | 3.3 | 0 | 2 |
| 20 | 61.3 | 3.3 | 2 | 0.7 | 0 | 7.3 | 3.3 | 3.3 | 0 | 3.3 | 5.3 | 0.7 | 2 | 2 | 0.7 | 2 |
| 21 | 63.1 | 4.7 | 4 | 2 | 0.7 | 4.7 | 0.7 | 3.4 | 0.7 | 3.4 | 3.4 | 0.7 | 2 | 2.7 | 0.7 | 3.4 |
| 22 | 64.9 | 4.1 | 2 | 1.4 | 0.7 | 6.8 | 3.4 | 0 | 2 | 4.7 | 2 | 0.7 | 4.1 | 0.7 | 0.7 | 2 |
| 23 | 63.3 | 4.1 | 1.4 | 2.7 | 1.4 | 6.1 | 0.7 | 1.4 | 0.7 | 2 | 3.4 | 2 | 2.7 | 2.7 | 2 | 3.4 |
| 24 | 74 | 4.8 | 2.1 | 2.1 | 0 | 3.4 | 1.4 | 2.1 | 0 | 2.1 | 1.4 | 0 | 0 | 3.4 | 0.7 | 2.7 |
| 25 | 62.8 | 2.1 | 2.8 | 2.8 | 0.7 | 6.9 | 2.1 | 3.4 | 0 | 2.1 | 2.8 | 2.8 | 2.1 | 4.1 | 2.1 | 0.7 |
| 26 | 61.8 | 5.6 | 0 | 3.5 | 1.4 | 7.6 | 4.2 | 1.4 | 1.4 | 0.7 | 1.4 | 2.8 | 4.2 | 0.7 | 0 | 3.5 |
| 27 | 69.9 | 2.1 | 0.7 | 3.5 | 1.4 | 4.2 | 3.5 | 0.7 | 0 | 3.5 | 2.8 | 1.4 | 1.4 | 2.1 | 0.7 | 2.1 |
| 28 | 65.5 | 0 | 1.4 | 1.4 | 2.1 | 5.6 | 3.5 | 2.8 | 1.4 | 1.4 | 7 | 1.4 | 0.7 | 2.8 | 1.4 | 1.4 |
| 29 | 59.6 | 2.1 | 4.3 | 5 | 0.7 | 5.7 | 3.5 | 4.3 | 2.8 | 2.1 | 2.8 | 0.7 | 0.7 | 0.7 | 1.4 | 3.5 |
| 30 | 67.9 | 2.1 | 1.4 | 3.6 | 0.7 | 5 | 2.9 | 2.9 | 0.7 | 2.1 | 1.4 | 0 | 2.1 | 2.9 | 2.1 | 2.1 |
| 31 | 60.4 | 3.6 | 2.2 | 2.2 | 2.9 | 7.2 | 5 | 3.6 | 0 | 2.9 | 0.7 | 1.4 | 1.4 | 3.6 | 1.4 | 1.4 |
| 32 | 58.7 | 4.3 | 1.4 | 3.6 | 0 | 4.3 | 2.9 | 4.3 | 2.9 | 2.9 | 2.9 | 1.4 | 2.9 | 2.2 | 0.7 | 4.3 |
| 33 | 59.9 | 3.6 | 2.9 | 2.9 | 0.7 | 4.4 | 4.4 | 2.9 | 1.5 | 1.5 | 2.2 | 1.5 | 2.2 | 5.8 | 0.7 | 2.9 |
| 34 | 57.4 | 5.1 | 3.7 | 2.9 | 2.2 | 6.6 | 4.4 | 3.7 | 0.7 | 2.9 | 3.7 | 2.9 | 1.5 | 1.5 | 0 | 0.7 |
| 35 | 66.7 | 0.7 | 3 | 2.2 | 0.7 | 4.4 | 6.7 | 3 | 1.5 | 2.2 | 1.5 | 0 | 1.5 | 2.2 | 1.5 | 2.2 |
| 36 | 67.9 | 2.2 | 1.5 | 2.2 | 0.7 | 6.7 | 1.5 | 4.5 | 1.5 | 1.5 | 4.5 | 0 | 2.2 | 2.2 | 0 | 0.7 |
| 37 | 66.2 | 3 | 3 | 4.5 | 1.5 | 4.5 | 3.8 | 1.5 | 0.8 | 2.3 | 0.8 | 0.8 | 1.5 | 0.8 | 2.3 | 3 |
| 38 | 67.4 | 3 | 0.8 | 2.3 | 0 | 3 | 1.5 | 5.3 | 0.8 | 3.8 | 2.3 | 0 | 4.5 | 1.5 | 1.5 | 2.3 |
| 39 | 70.2 | 1.5 | 2.3 | 2.3 | 0 | 3.8 | 2.3 | 3.1 | 2.3 | 3.8 | 1.5 | 0.8 | 1.5 | 3.1 | 0 | 1.5 |
| 40 | 65.4 | 4.6 | 1.5 | 3.8 | 0.8 | 5.4 | 2.3 | 1.5 | 0 | 1.5 | 3.1 | 0.8 | 1.5 | 2.3 | 0 | 5.4 |
| 41 | 65.9 | 4.7 | 2.3 | 0.8 | 1.6 | 4.7 | 4.7 | 3.1 | 1.6 | 0.8 | 1.6 | 0 | 3.1 | 1.6 | 2.3 | 1.6 |
| 42 | 69.5 | 5.5 | 3.1 | 0.8 | 1.6 | 3.9 | 0.8 | 3.1 | 0 | 1.6 | 2.3 | 0.8 | 2.3 | 3.9 | 0 | 0.8 |
| 43 | 64.6 | 5.5 | 0.8 | 3.9 | 0 | 6.3 | 2.4 | 2.4 | 0 | 3.9 | 3.9 | 0.8 | 0.8 | 0.8 | 0.8 | 3.1 |
| 44 | 73 | 1.6 | 5.6 | 2.4 | 0.8 | 4 | 1.6 | 0.8 | 1.6 | 3.2 | 1.6 | 1.6 | 0 | 0 | 0 | 1.6 |
| 45 | 72.8 | 3.2 | 4 | 4 | 1.6 | 1.6 | 3.2 | 3.2 | 0 | 0 | 0.8 | 0 | 0 | 1.6 | 0.8 | 3.2 |
| 46 | 62.9 | 4.8 | 1.6 | 4 | 0.8 | 10.5 | 1.6 | 3.2 | 0 | 3.2 | 1.6 | 0 | 1.6 | 3.2 | 0 | 0.8 |
| 47 | 69.9 | 2.4 | 0 | 4.1 | 0.8 | 4.1 | 1.6 | 4.9 | 1.6 | 0.8 | 2.4 | 0 | 0 | 3.3 | 0 | 4.1 |
| 48 | 68.9 | 4.9 | 2.5 | 2.5 | 1.6 | 2.5 | 1.6 | 2.5 | 0.8 | 2.5 | 0.8 | 0.8 | 4.9 | 2.5 | 0 | 0.8 |
| 49 | 59.5 | 4.1 | 2.5 | 1.7 | 0 | 9.9 | 0.8 | 2.5 | 1.7 | 1.7 | 3.3 | 1.7 | 4.1 | 3.3 | 0 | 3.3 |
| 50 | 66.7 | 4.2 | 1.7 | 1.7 | 0 | 6.7 | 4.2 | 2.5 | 0 | 2.5 | 4.2 | 0 | 0.8 | 3.3 | 0.8 | 0.8 |
| 51 | 68.1 | 2.5 | 3.4 | 0 | 0.8 | 3.4 | 3.4 | 2.5 | 1.7 | 0.8 | 2.5 | 2.5 | 0.8 | 5 | 0 | 2.5 |
| 52 | 62.7 | 1.7 | 0.8 | 3.4 | 0 | 4.2 | 1.7 | 5.1 | 0.8 | 4.2 | 0.8 | 1.7 | 2.5 | 5.9 | 1.7 | 2.5 |
| 53 | 69.2 | 6.8 | 0 | 1.7 | 0 | 5.1 | 1.7 | 3.4 | 0 | 1.7 | 1.7 | 0.9 | 0.9 | 5.1 | 0.9 | 0.9 |
| 54 | 60.3 | 6 | 0.9 | 1.7 | 1.7 | 4.3 | 4.3 | 4.3 | 0.9 | 4.3 | 3.4 | 0.9 | 1.7 | 3.4 | 0.9 | 0.9 |
| 55 | 72.2 | 0.9 | 0.9 | 0.9 | 1.7 | 6.1 | 1.7 | 1.7 | 1.7 | 1.7 | 0 | 1.7 | 5.2 | 0 | 0.9 | 2.6 |
| 56 | 75.4 | 3.5 | 2.6 | 1.8 | 0 | 0.9 | 0 | 3.5 | 2.6 | 1.8 | 0 | 0.9 | 0.9 | 3.5 | 1.8 | 1.8 |
| 57 | 66.4 | 3.5 | 0 | 1.8 | 0.9 | 6.2 | 3.5 | 1.8 | 1.8 | 1.8 | 1.8 | 1.8 | 2.7 | 2.7 | 0.9 | 2.7 |
| 58 | 70.5 | 0 | 1.8 | 1.8 | 0.9 | 7.1 | 4.5 | 0.9 | 0.9 | 2.7 | 2.7 | 0.9 | 0 | 1.8 | 1.8 | 1.8 |
| 59 | 66.7 | 2.7 | 0 | 1.8 | 0.9 | 2.7 | 1.8 | 3.6 | 0.9 | 1.8 | 2.7 | 0.9 | 4.5 | 0.9 | 0.9 | 7.2 |
| 60 | 69.1 | 3.6 | 0 | 1.8 | 1.8 | 3.6 | 7.3 | 0.9 | 0.9 | 0.9 | 2.7 | 0 | 0.9 | 4.5 | 0.9 | 0.9 |
| 61 | 67 | 2.8 | 3.7 | 7.3 | 0 | 5.5 | 0.9 | 1.8 | 0 | 2.8 | 1.8 | 0.9 | 1.8 | 1.8 | 1.8 | 0 |
| 62 | 61.1 | 0 | 2.8 | 1.9 | 2.8 | 5.6 | 0.9 | 3.7 | 2.8 | 5.6 | 0 | 0.9 | 6.5 | 0.9 | 2.8 | 1.9 |
| 63 | 68.2 | 0.9 | 2.8 | 1.9 | 0 | 4.7 | 0.9 | 2.8 | 1.9 | 2.8 | 1.9 | 0.9 | 3.7 | 1.9 | 1.9 | 2.8 |
| 64 | 65.1 | 3.8 | 2.8 | 2.8 | 0 | 5.7 | 0 | 5.7 | 0 | 3.8 | 1.9 | 0.9 | 0.9 | 3.8 | 0.9 | 1.9 |
| 65 | 70.5 | 1.9 | 1.9 | 1 | 0 | 3.8 | 2.9 | 3.8 | 1.9 | 2.9 | 5.7 | 0 | 1.9 | 1 | 0 | 1 |
| 66 | 66.3 | 4.8 | 2.9 | 3.8 | 1 | 5.8 | 2.9 | 3.8 | 1 | 0 | 0 | 1 | 1 | 4.8 | 0 | 1 |
| 67 | 60.2 | 1.9 | 3.9 | 1 | 4.9 | 5.8 | 1 | 1.9 | 1.9 | 2.9 | 2.9 | 2.9 | 1.9 | 1.9 | 1 | 3.9 |
| 68 | 69.6 | 3.9 | 2 | 3.9 | 1 | 2.9 | 2.9 | 1 | 1 | 2.9 | 3.9 | 0 | 1 | 1 | 1 | 2 |
| 69 | 72.3 | 2 | 4 | 2 | 1 | 1 | 2 | 2 | 0 | 1 | 4 | 3 | 1 | 4 | 3 | 1 |
| 70 | 64 | 6 | 1 | 2 | 0 | 4 | 2 | 0 | 0 | 5 | 5 | 0 | 2 | 5 | 1 | 3 |
| 71 | 70.7 | 4 | 5.1 | 0 | 1 | 5.1 | 2 | 3.1 | 0 | 2 | 2 | 0 | 3.1 | 5.1 | 0 | 2 |
| 72 | 63.3 | 7.1 | 4.1 | 0 | 1 | 5.1 | 2 | 2 | 2 | 0 | 3.1 | 5.1 | 0 | 2 | 2.1 | 2.1 |
| 73 | 72.2 | 2.1 | 1 | 1 | 0 | 5.2 | 1 | 2.1 | 0 | 3.1 | 4.1 | 0 | 4.1 | 0 | 2.1 | 2.1 |
| 74 | 64.6 | 2.1 | 1 | 2.1 | 3.1 | 2.1 | 1 | 2.1 | 1 | 5.2 | 5.2 | 1 | 3.1 | 3.1 | 2.1 | 1 |
| 75 | 71.6 | 3.2 | 3.2 | 1.1 | 0 | 6.3 | 2.1 | 0 | 1.1 | 4.2 | 1.1 | 0 | 2.1 | 2.1 | 1.1 | 1.1 |
| 76 | 68.1 | 3.2 | 0 | 8.5 | 0 | 6.4 | 1.1 | 1.1 | 2.1 | 1.1 | 1.1 | 0 | 2.1 | 3.2 | 2.1 | 0 |
| 77 | 74.2 | 1.1 | 0 | 1.1 | 1.1 | 2.2 | 3.2 | 1.1 | 0 | 3.2 | 3.2 | 1.1 | 1.1 | 2.2 | 1.1 | 4.3 |
| 78 | 72.8 | 4.3 | 1.1 | 3.3 | 1.1 | 4.3 | 0 | 0 | 3.3 | 3.3 | 2.2 | 1.1 | 1.1 | 1.1 | 0 | 2.2 |
| 79 | 68.1 | 2.2 | 2.2 | 1.1 | 1.1 | 7.7 | 3.3 | 0 | 3.3 | 2.2 | 3.3 | 1.1 | 0 | 1.1 | 1.1 | 2.2 |
| 80 | 61.1 | 3.3 | 3.3 | 5.6 | 1.1 | 8.9 | 2.2 | 0 | 3.3 | 1.1 | 1.1 | 1.1 | 3.3 | 0 | 1.1 | 3.3 |
| 81 | 67.4 | 3.4 | 3.4 | 4.5 | 1.1 | 5.6 | 3.4 | 1.1 | 1.1 | 2.2 | 0 | 2.2 | 0 | 0 | 2.2 | 2.2 |
| 82 | 71.6 | 3.4 | 2.3 | 0 | 1.1 | 1.1 | 4.5 | 3.4 | 1.1 | 2.3 | 1.1 | 2.3 | 2.3 | 1.1 | 1.1 | 1.1 |
| 83 | 72.4 | 2.3 | 0 | 1.1 | 0 | 3.4 | 3.4 | 2.3 | 3.4 | 2.3 | 2.3 | 0 | 2.3 | 2.3 | 1.1 | 1.1 |
| 84 | 73.3 | 3.5 | 2.3 | 2.3 | 0 | 5.8 | 1.2 | 2.3 | 0 | 1.2 | 2.3 | 1.2 | 0 | 4.7 | 0 | 0 |
| 85 | 75.3 | 2.4 | 1.2 | 2.4 | 0 | 3.5 | 2.4 | 0 | 1.2 | 1.2 | 1.2 | 0 | 1.2 | 2.4 | 1.2 | 4.7 |
| 86 | 63.1 | 2.4 | 0 | 8.3 | 0 | 6 | 2.4 | 2.4 | 0 | 1.2 | 4.8 | 1.2 | 4.8 | 1.2 | 1.2 | 1.2 |
| 87 | 60.2 | 1.2 | 1.2 | 2.4 | 0 | 3.6 | 4.8 | 2.4 | 1.2 | 3.6 | 2.4 | 2.4 | 6 | 3.6 | 0 | 4.8 |
| 88 | 72 | 4.9 | 1.2 | 0 | 0 | 4.9 | 2.4 | 1.2 | 1.2 | 1.2 | 4.9 | 1.2 | 2.4 | 1.2 | 0 | 1.2 |
| 89 | 70.4 | 3.7 | 0 | 3.7 | 1.2 | 6.2 | 1.2 | 1.2 | 1.2 | 3.7 | 0 | 0 | 0 | 1.2 | 1.2 | 4.9 |
| 90 | 72.5 | 5 | 0 | 1.2 | 1.2 | 5 | 3.8 | 2.5 | 2.5 | 2.5 | 0 | 1.2 | 1.2 | 0 | 0 | 1.2 |
| 91 | 68.4 | 1.3 | 1.3 | 6.3 | 1.3 | 2.5 | 2.5 | 0 | 0 | 3.8 | 1.3 | 2.5 | 2.5 | 1.3 | 1.3 | 5.1 |
| 92 | 48.7 | 7.7 | 1.3 | 6.4 | 1.3 | 7.7 | 1.3 | 2.6 | 2.6 | 6.4 | 5.1 | 1.3 | 1.3 | 3.8 | 2.6 | 0 |
| 93 | 66.2 | 3.9 | 3.9 | 1.3 | 0 | 2.6 | 1.3 | 1.3 | 3.9 | 2.6 | 0 | 2.6 | 3.9 | 5.2 | 0 | 1.3 |
| 94 | 69.7 | 2.6 | 1.3 | 2.6 | 1.3 | 1.3 | 2.6 | 2.6 | 1.3 | 5.3 | 2.6 | 1.3 | 1.3 | 1.3 | 0 | 2.6 |
| 95 | 77.3 | 4 | 0 | 1.3 | 0 | 5.3 | 1.3 | 2.7 | 0 | 2.7 | 0 | 1.3 | 1.3 | 2.7 | 0 | 0 |
| 96 | 77 | 4.1 | 0 | 4.1 | 0 | 4.1 | 1.4 | 2.7 | 1.4 | 1.4 | 0 | 0 | 1.4 | 1.4 | 0 | 0 |
| 97 | 58.9 | 5.5 | 4.1 | 5.5 | 0 | 8.2 | 0 | 0 | 0 | 2.7 | 2.7 | 0 | 1.4 | 4.1 | 1.4 | 5.5 |
| 98 | 73.6 | 2.8 | 0 | 1.4 | 0 | 6.9 | 2.8 | 2.8 | 0 | 0 | 2.8 | 0 | 0 | 0 | 2.8 | 4.2 |
| 99 | 63.4 | 7 | 5.6 | 4.2 | 1.4 | 4.2 | 1.4 | 1.4 | 0 | 1.4 | 1.4 | 0 | 1.4 | 2.8 | 0 | 4.2 |
| 100 | 70 | 1.4 | 4.3 | 2.9 | 0 | 0 | 2.9 | 4.3 | 0 | 1.4 | 0 | 1.4 | 0 | 4.3 | 1.4 | 5.7 |
| 101 | 68.1 | 1.4 | 0 | 0 | 1.4 | 5.8 | 0 | 4.3 | 1.4 | 2.9 | 1.4 | 2.9 | 5.8 | 0 | 0 | 4.3 |
| 102 | 57.4 | 10.3 | 0 | 4.4 | 0 | 10.3 | 2.9 | 1.5 | 0 | 1.5 | 2.9 | 0 | 0 | 2.9 | 1.5 | 4.4 |
| 103 | 76.1 | 1.5 | 0 | 0 | 0 | 4.5 | 0 | 0 | 3 | 1.5 | 4.5 | 0 | 6 | 0 | 1.5 | 1.5 |
| 104 | 68.2 | 4.5 | 1.5 | 3 | 0 | 7.6 | 0 | 4.5 | 1.5 | 3 | 0 | 0 | 3 | 1.5 | 0 | 1.5 |
| 105 | 73.8 | 1.5 | 0 | 3.1 | 0 | 4.6 | 3.1 | 3.1 | 1.5 | 1.5 | 1.5 | 0 | 4.6 | 1.5 | 0 | 0 |
| 106 | 68.8 | 3.1 | 1.6 | 1.6 | 0 | 12.5 | 0 | 3.1 | 0 | 1.6 | 4.7 | 0 | 0 | 1.6 | 0 | 1.6 |
| 107 | 73 | 6.3 | 3.2 | 0 | 1.6 | 3.2 | 1.6 | 0 | 0 | 1.6 | 1.6 | 0 | 1.6 | 3.2 | 0 | 3.2 |
| 108 | 69.4 | 3.2 | 1.6 | 0 | 1.6 | 6.5 | 0 | 8.1 | 0 | 1.6 | 1.6 | 0 | 0 | 3.2 | 3.2 | 0 |
| 109 | 67.2 | 3.3 | 1.6 | 1.6 | 0 | 9.8 | 1.6 | 0 | 1.6 | 1.6 | 3.3 | 0 | 1.6 | 3.3 | 1.6 | 1.6 |
| 110 | 66.7 | 3.3 | 0 | 0 | 0 | 5 | 0 | 3.3 | 3.3 | 1.7 | 1.7 | 1.7 | 3.3 | 6.7 | 1.7 | 1.7 |
| 111 | 76.3 | 0 | 0 | 3.4 | 0 | 3.4 | 1.7 | 3.4 | 1.7 | 0 | 5.1 | 0 | 0 | 1.7 | 0 | 3.4 |
| 112 | 69 | 3.4 | 3.4 | 1.7 | 0 | 5.2 | 0 | 1.7 | 0 | 5.2 | 1.7 | 1.7 | 3.4 | 1.7 | 0 | 1.7 |
| 113 | 59.6 | 0 | 1.8 | 3.5 | 0 | 3.5 | 7 | 5.3 | 1.8 | 5.3 | 5.3 | 1.8 | 3.5 | 0 | 0 | 1.8 |
| 114 | 76.8 | 0 | 0 | 1.8 | 1.8 | 0 | 0 | 1.8 | 1.8 | 0 | 1.8 | 0 | 5.4 | 7.1 | 0 | 1.8 |
| 115 | 72.7 | 7.3 | 0 | 1.8 | 0 | 3.6 | 0 | 1.8 | 0 | 1.8 | 0 | 0 | 5.5 | 3.6 | 0 | 1.8 |
| 116 | 70.4 | 1.9 | 0 | 1.9 | 1.9 | 1.9 | 3.7 | 1.9 | 0 | 0 | 5.6 | 1.9 | 1.9 | 5.6 | 1.9 | 0 |
| 117 | 75.5 | 3.8 | 1.9 | 1.9 | 0 | 3.8 | 0 | 1.9 | 5.7 | 1.9 | 0 | 1.9 | 0 | 0 | 0 | 1.9 |
| 118 | 67.3 | 0 | 0 | 1.9 | 3.8 | 5.8 | 3.8 | 1.9 | 0 | 1.9 | 3.8 | 1.9 | 5.8 | 1.9 | 0 | 0 |
| 119 | 70.6 | 5.9 | 0 | 2 | 0 | 9.8 | 2 | 5.9 | 0 | 0 | 0 | 0 | 2 | 2 | 0 | 0 |
| 120 | 74 | 0 | 0 | 2 | 0 | 2 | 2 | 4 | 0 | 0 | 6 | 0 | 6 | 0 | 2 | 2 |
| 121 | 75.5 | 4.1 | 2 | 0 | 0 | 4.1 | 2 | 4.1 | 0 | 0 | 2 | 2 | 2 | 2 | 0 | 2 |
| 122 | 72.9 | 4.2 | 0 | 2.1 | 0 | 0 | 0 | 4.2 | 2.1 | 2.1 | 2.1 | 2.1 | 4.2 | 2.1 | 0 | 2.1 |
| 123 | 61.7 | 8.5 | 2.1 | 2.1 | 2.1 | 8.5 | 2.1 | 2.1 | 0 | 2.1 | 4.3 | 0 | 0 | 0 | 0 | 4.3 |
| 124 | 69.6 | 0 | 0 | 2.2 | 2.2 | 4.3 | 2.2 | 2.2 | 4.3 | 2.2 | 2.2 | 2.2 | 0 | 2.2 | 2.2 | 2.2 |
| 125 | 68.9 | 2.2 | 2.2 | 2.2 | 2.2 | 6.7 | 0 | 2.2 | 2.2 | 2.2 | 0 | 2.2 | 0 | 2.2 | 0 | 2.2 |
| 126 | 70.5 | 4.5 | 0 | 2.3 | 2.3 | 4.5 | 0 | 4.5 | 0 | 2.3 | 0 | 0 | 2.3 | 4.5 | 0 | 2.3 |
| 127 | 76.7 | 2.3 | 2.3 | 2.3 | 0 | 7 | 4.7 | 0 | 0 | 2.3 | 0 | 0 | 2.3 | 0 | 0 | 0 |
| 128 | 73.8 | 0 | 0 | 0 | 0 | 2.4 | 0 | 4.8 | 0 | 2.4 | 2.4 | 0 | 4.8 | 4.8 | 0 | 4.8 |
| 129 | 78 | 2.4 | 0 | 0 | 0 | 2.4 | 2.4 | 2.4 | 0 | 0 | 2.4 | 0 | 2.4 | 4.9 | 0 | 2.4 |
| 130 | 65 | 7.5 | 5 | 0 | 0 | 10 | 0 | 5 | 2.5 | 0 | 2.5 | 0 | 2.5 | 0 | 0 | 0 |
| 131 | 69.2 | 2.6 | 2.6 | 0 | 2.6 | 7.7 | 0 | 5.1 | 0 | 0 | 2.6 | 0 | 5.1 | 0 | 2.6 | 0 |
| 132 | 76.3 | 5.3 | 0 | 2.6 | 0 | 0 | 2.6 | 0 | 0 | 7.9 | 0 | 0 | 0 | 2.6 | 2.6 | 0 |
| 133 | 73 | 5.4 | 0 | 0 | 2.7 | 8.1 | 2.7 | 2.7 | 0 | 0 | 2.7 | 0 | 0 | 0 | 0 | 2.7 |
| 134 | 61.1 | 0 | 5.6 | 0 | 2.8 | 13.9 | 2.8 | 0 | 0 | 2.8 | 0 | 0 | 0 | 5.6 | 2.8 | 2.8 |
| 135 | 71.4 | 8.6 | 0 | 0 | 0 | 5.7 | 0 | 2.9 | 0 | 0 | 0 | 0 | 2.9 | 2.9 | 0 | 5.7 |
| 136 | 73.5 | 0 | 0 | 0 | 5.9 | 0 | 2.9 | 0 | 0 | 0 | 2.9 | 2.9 | 0 | 8.8 | 2.9 | 0 |
| 137 | 69.7 | 6.1 | 3 | 0 | 6.1 | 0 | 3 | 6.1 | 3 | 0 | 0 | 0 | 0 | 3 | 0 | 0 |
| 138 | 68.8 | 0 | 0 | 3.1 | 0 | 6.2 | 3.1 | 0 | 0 | 9.4 | 0 | 6.2 | 3.1 | 0 | 0 | 0 |
| 139 | 71 | 0 | 3.2 | 3.2 | 0 | 3.2 | 0 | 6.5 | 0 | 0 | 6.5 | 0 | 3.2 | 0 | 0 | 3.2 |
| 140 | 66.7 | 6.7 | 3.3 | 0 | 3.3 | 0 | 3.3 | 0 | 0 | 3.3 | 0 | 6.7 | 3.3 | 3.3 | 0 | 0 |
| 141 | 82.8 | 0 | 0 | 0 | 0 | 6.9 | 0 | 0 | 3.4 | 3.4 | 0 | 0 | 3.4 | 0 | 0 | 0 |
| 142 | 64.3 | 0 | 0 | 7.1 | 0 | 10.7 | 0 | 0 | 3.6 | 7.1 | 0 | 3.6 | 0 | 0 | 0 | 3.6 |
| 143 | 81.5 | 0 | 3.7 | 0 | 0 | 3.7 | 0 | 0 | 7.4 | 0 | 0 | 0 | 0 | 0 | 0 | 3.7 |
| 144 | 69.2 | 11.5 | 0 | 0 | 0 | 7.7 | 0 | 3.8 | 0 | 0 | 3.8 | 3.8 | 0 | 0 | 0 | 0 |
| 145 | 92 | 8 | 0 | 0 | 0 | 0 | 0 | 0 | 0 | 0 | 0 | 0 | 0 | 0 | 0 | 0 |
| 146 | 75 | 0 | 0 | 0 | 0 | 4.2 | 0 | 0 | 4.2 | 0 | 0 | 0 | 4.2 | 8.3 | 0 | 4.2 |
| 147 | 69.6 | 0 | 0 | 4.3 | 0 | 8.7 | 4.3 | 0 | 8.7 | 0 | 0 | 0 | 4.3 | 0 | 0 | 0 |
| 148 | 59.1 | 0 | 0 | 0 | 4.5 | 9.1 | 0 | 0 | 0 | 9.1 | 4.5 | 0 | 0 | 0 | 9.1 | 4.5 |
| 149 | 66.7 | 0 | 0 | 0 | 0 | 4.8 | 0 | 4.8 | 0 | 0 | 0 | 0 | 4.8 | 4.8 | 0 | 14.3 |
| 150 | 65 | 5 | 0 | 5 | 0 | 0 | 0 | 5 | 0 | 0 | 0 | 5 | 5 | 5 | 0 | 5 |
| 151 | 78.9 | 5.3 | 0 | 0 | 0 | 10.5 | 0 | 0 | 0 | 0 | 5.3 | 0 | 0 | 0 | 0 | 0 |
| 152 | 77.8 | 0 | 0 | 0 | 5.6 | 11.1 | 0 | 0 | 0 | 0 | 0 | 0 | 5.6 | 0 | 0 | 0 |
| 153 | 70.6 | 5.9 | 0 | 0 | 0 | 0 | 11.8 | 5.9 | 0 | 0 | 0 | 0 | 0 | 0 | 0 | 5.9 |
| 154 | 81.2 | 0 | 6.2 | 0 | 0 | 6.2 | 0 | 0 | 0 | 0 | 0 | 0 | 0 | 0 | 0 | 6.2 |
| 155 | 80 | 6.7 | 0 | 6.7 | 0 | 0 | 0 | 0 | 0 | 0 | 0 | 0 | 0 | 0 | 0 | 0 |
| 156 | 71.4 | 0 | 0 | 7.1 | 0 | 7.1 | 0 | 0 | 0 | 0 | 14.3 | 0 | 0 | 0 | 0 | 0 |
| 157 | 92.3 | 0 | 0 | 0 | 0 | 0 | 0 | 0 | 0 | 0 | 0 | 0 | 7.7 | 0 | 0 | 0 |
| 158 | 75 | 8.3 | 8.3 | 0 | 0 | 0 | 0 | 0 | 0 | 8.3 | 0 | 0 | 0 | 0 | 0 | 0 |
| 159 | 63.6 | 9.1 | 9.1 | 9.1 | 0 | 0 | 0 | 0 | 0 | 0 | 0 | 0 | 0 | 0 | 0 | 9.1 |
| 160 | 70 | 0 | 0 | 10 | 0 | 0 | 0 | 0 | 0 | 10 | 0 | 10 | 0 | 0 | 0 | 0 |
| 161 | 88.9 | 0 | 0 | 11.1 | 0 | 0 | 0 | 0 | 0 | 0 | 0 | 0 | 0 | 0 | 0 | 0 |
| 162 | 57 | 0 | 0 | 0 | 0 | 25 | 12.5 | 12.5 | 0 | 0 | 0 | 12.5 | 0 | 0 | 0 | 0 |
| 163 | 85.7 | 0 | 0 | 0 | 0 | 0 | 14.3 | 0 | 0 | 0 | 0 | 0 | 0 | 0 | 0 | 0 |
| 164 | 83.3 | 0 | 0 | 0 | 0 | 0 | 0 | 16.7 | 0 | 0 | 0 | 0 | 0 | 0 | 0 | 0 |
| 165 | 80 | 0 | 0 | 0 | 0 | 0 | 0 | 0 | 0 | 0 | 0 | 0 | 0 | 20 | 0 | 0 |
| 166 | 100 | 0 | 0 | 0 | 0 | 0 | 0 | 0 | 0 | 0 | 0 | 0 | 0 | 0 | 0 | 0 |
| 167 | 66.7 | 0 | 0 | 0 | 0 | 0 | 33.3 | 0 | 0 | 0 | 0 | 0 | 0 | 0 | 0 | 0 |
| 168 | 50 | 0 | 0 | 0 | 0 | 0 | 0 | 0 | 0 | 0 | 0 | 0 | 50 | 0 | 0 | 0 |
| 169 | 100 | 0 | 0 | 0 | 0 | 0 | 0 | 0 | 0 | 0 | 0 | 0 | 0 | 0 | 0 | 0 |

---

## Author Comment (AC3) · 8 Mar 2018

AFI ( °C)

Return Period (years)

---

## Author Response (AR1)

Dear Editor,

Dear reviewers,

I would like first to thank the reviewers for their constructive and very helpful comments. I am replying point-by-point to the reviewer comments in red, discussing at the same time the relevant changes in the manuscript. Due to the significant changes in the manuscript (also in its structure), it has not been possible to include a marked-up manuscript version.

With best regards,

Symeon Koumoutsaris

**REVIEW 1**

**Main Comment**

To reply to the main comment of the first reviewer, I calculated the associated uncertainties using a bootstrap approach as suggested by the reviewer (see section 4.2.3). The uncertainty associated with the Monte Carlo simulation alone is also calculated. The procedures followed are described in section of the revised manuscript.

The calculated confidence intervals resulting from the Monte Carlo simulation almost entirely account for the model uncertainty estimated using the parametric bootstrap approach (i.e. which encompasses both (a) and (b) types of uncertainty), suggesting that the uncertainty in the RVM is negligible. The reason of the small uncertainty in the RVM is twofold: the large majority of the pairs are estimated to be independent and also the most important dependencies are captured at the first levels. Both reasons lead to a virtual reduction in the dimensions of the pdf.

An additional analysis has been performed in order to confirm that the first trees capture the most important dependencies. Figure 8a shows the return period plots for the same RVM but truncated above the first seven levels (i.e. using independent copulas above level 1, 2, 3, up to 7). The same seed as for the default RVM is used in the simulation of these truncated models in order to avoid differences associated with the Monte Carlo sampling. The return period curves are quite similar for the RVMs with truncation above level 2, indicating that the first two levels capture most of the dependency structure.

Section 4.2.3 has been added to discuss the main sources of the model uncertainty:

- the uncertainty due to the short historical record length
- the uncertainty resulting from the length of the Monte Carlo simulation (i.e. the number of simulated winters)
- the uncertainty in the joint pdf (i.e. in the RVM) due to the limited historical record
- the uncertainty due to the model assumption of a stationary climate

**Specific comments**

**Structure of the paper**

I changed the structure of the paper as suggested by the reviewer

**About the vine (P9)**

1) An equation with an example of a Vine (e.g. in 4 dimension) would be helpful for the reader. In particular this should be shown in combination with the uniform variables used for the vine fit (i.e. the "marginal variables" coming from the GEV).

The equation has been added in section 3.3.

2) The structure of the used vine is not clear. A table with the percentage of family types used in each tree would be appreciated by the reader.

The percentage of family types for the first 5 levels and overall are presented in Table 2. Due to the size of the vine, a table containing the percentages in all trees is added as the supplementary material (Table 1).

3) There is not enough information about the procedure used for the fitting of the vine, e.g. what criteria was used for the selection of the RVM structure, what criteria was used to fit the pair copulas, or how you assigned independence to some of the par-copulas. There are references to the R-package, however this is not enough, also considering that in the package different approaches for fit can be used.

I have added the following paragraph in section 3.3.1:

"The copula family types for each selected pair in the first tree are determined by using the Akaike information criterion (see Brechmann and Schepsmeier, (2013)). For computational reasons, the two-parameter Archimedean copulas are excluded from this analysis (which however has only a negligible impact in the results, see Figure 5 A1 of the Appendix). The copula parameters are estimated sequentially (using maximum likelihood estimation) starting from the top tree until the last tree, as described in Czado et al. (2013). This approach only involves estimation of bivariate copulas and has been chosen since it is computationally much simpler than joint maximum likelihood estimation of all parameters at once."

**Methodology**

P1 l1: the third coldest winter ever recorded. Where and according to what criteria?

I have updated the abstract to make this clearer. This is according to the Central England Temperature (CET) record, the oldest continuously running temperature dataset in the world (Manley, 1974), only

two other winters (1683/84 and 1739/40) have been colder than 1962/63 in the last 350 years (also mentioned in the submitted manuscript).

P3 l5 It is based on rigorously quality checked station data interpolated to a regular grid using inverse-distance weighting, as described in Perry et al. (2009). It should be mentioned here or later that therefore the dependencies catched by the copulas may be partially due to the interpolation itself.

This has been mentioned in section 2 (line 7)

P3 l10 Nevertheless, local temperature may be subtly different in certain micro-climates, such as upland and urban regions. I would mention that however the resolution 5km x 5km may not always be realistic, depending on the number of stations which were available for the creation of the data set.

I agree. However, notice that I re-grid the data to a lower resolution of 50 x 50 km2. The station network contains 540 stations with an average density of 21 x 21 km2, with also more stations near urban areas (see Figures 1 and 2 in Perry and Hollis, 2005). This has been added in section 2.

P3 l29 98.3°C. Based on line 17, I expected negative values for the AFI. Could you mention that you take the absolute values of the temperature? Also, it would be appreciated if you would show the equation of the AFI.

The exact equation is

$$AFI = \sum_{day=1/6/Year}^{31/5/(Year+1)} |T_{day}| \qquad if\ T_{day} < 0°C$$

And it has been added in section 3.1

P3 l32 After 1962/63, a long run of mild winters followed until late 1978 and early 1979 (Figure 2). Is this in Figure 2 the AFI averaged over UK? Please, use °C in the y label of Fig. 2.

Yes, that's has been corrected. Also I changed in the text and plot and specify now the mean AFI as "mAFI" to make it clearer.

P5 l4 An additional term was included, the probability of no hazard (P0), in order to account for the cells mainly on the south England coast that have years with no negative temperatures at all.

1) Does this mean that for some cells the GEV is fitted on very few data? Please give information about this, and on the goodness of the fit for these cells.

In order to improve the fits at those cells, I applied a geographical smoothing of the GEV parameters as well; I had erroneously missed to discuss this part from the submitted text. I will update the text to reflect in detail this methodology.

More precisely, along with the TWMLE method described in the text, I applied a second modification in order to geographically smooth the GEV parameters. The smoothing is incorporated into the fitting

process by minimizing the local (ranked) log-likelihood. More precisely, the log-likelihood at each grid cell $i$ is calculated using all grid points but weighted by their distance $d_{ij}$:

$$LogL_i = \sum_{j=1}^{170}(w_{ij} \cdot LogL_j), \text{ where } w_{ij} = \frac{1}{\sqrt{2\pi}}e^{-\frac{d_{ij}^2}{2L^2}}$$

where $L$ here is the length scale or smoothing parameter and $LogL_j$ is the ranked log-likelihood for cell j. Because the historical gridded data are already geographically smoothed, I decided to use a small length scale parameter $L$ of 15 km (in comparison to the 50km grid size).

In general, the increase of the sample size at each grid point allows for a more precise estimation of the parameters, especially for the shape parameter which is highly influential in estimating the hazard levels and high return periods. This methodology also permits the estimation of the parameters in cells with no data.

A table with the gev parameters in all cells is uploaded as a supplementary material (Table 2).

2) Please, specify how P0 is estimated, e.g. N_occurence/N_years. FIG 3

P0 = N_occurrence / (Nyears + 1), see section 3.2

1) I assume that the " historical AFI GEV fit (black circles) " is the empirical estimate. If yes, is this computed as written in P6 L5? Please, specify this.

Yes, that's is correct. I have added in parenthesis: "computed as described in Sect. 4.1" (page 6, line 15).

2) Could you specify the estimated parameters, or also only making clear to the reader whether the difference is due to the selection of different family type (Gumbel, Frechet, and Weibull distributions)?

I have added table 1 in the revised document which shows the parameters and discuss this in page 6, line 15 (both fits correspond to the Fréchet distribution)

P5 l20 As an example, the GEV fit for a single cell over London is shown in Figure 3. The grey line represents the GEV fit without any weighting applied, while the black curve is estimated using the TWMLE method with an improved fit towards the tail of the distribution (i.e. the more extreme events).I would rather say that you get a curve that is nearer to the empirical estimate.

Updated in the text as suggested by the reviewer.

P6 l2 Other urban regions (e.g. Manchester or Midlands area) do not stand out as much as a result of the low grid resolution. Can this also be due to the original data format? For example there may be not enough stations around some urban areas.

Figure 1 below shows the AFI values for the (empirically estimated) 50 years return period based on the historical data at their original resolution (5km x 5km). Apart from the London area, other cities (e.g. Liverpool, Manchester) stand out with lower values than their surroundings. It is therefore mainly the

re-gridding process to 50x50km (necessary for computing reasons) that masks the cities in this study's results.

**Empirical**

[Figure]

Figure 1: High resolution maps of AFI values (in ◦C) for return period of 50 years.

P9 l8 At the first level, 49% of the selected bivariate copulas are found to be Gumbel which implies greater dependence at larger AFI values. You refer to the tail dependence, I assume. Make it more clear, please. Greater with respect to what?

I meant greater with respect to the low AFI values, so indeed I refer to the tail dependence. I have rephrased it to make it more clear.

P9 l13 The RVM is used to simulate 10,000 years of winter-seasons in the UK. This amount of realisations should be long enough in order to estimate with enough confidence the 200 year RP hazard, which is commonly associated with capital and regulatory requirements.

The 10,000 years time series should be long enough to neglect uncertainties associated with the Monte Carlo simulations (which is the method used for extracting the return period associated with the fitted parametric pdf) (Serinaldi et al. (2015) and Bevacqua et al. (2017)). One should ensure if the sample is "long enough" via repeating the (10,000 years) simulations several times and checking if the there are differences in the estimated return period (if there are no differences, the 10,000 years sample is long enough). Performing a long enough simulations allows one to get a convergence to the true return period that one would get analytically from the fitted pdf (given the complexity of the problem it is impracticable to get an analytical derivation of the RP). Performing a long simulation does not solve the issue about the model uncertainties (uncertainties existing about the pdf), which is there because the pdf is calibrated on a finite - very short - sample. I suggest to discuss this in a way to make difference between these different type of uncertainties.

I have calculated the uncertainty associated solely due to the Monte Carlo sampling as suggested by the reviewer (see section 4.2.3)

- Using the selected RVM, the simulation of 10K years of winter seasons is repeated for 1,000 times.
- For each of these 1,000 simulations, the corresponding return period levels are calculated.
- The uncertainty in the return periods is estimated by identifying the 95\% confidence interval (i.e. the range 2.5–97.5 \%) from these 1,000 return level curves.

As mentioned before, I have also added Section 4.2.3 in order to discuss the main sources of the model uncertainty

P9 l 27 The exceedance probability (EP) curve of wAFI is shown in Figure 7, both for the historical and the stochastic data. So far you talked about RP. Personally, I think that it would be better to keep the same terminology instead of introducing EP, or at least use also RP here.

I agree with the reviewer and I have updated everywhere the text to reflect that.

P9 l27 The uncertainty intervals in the historical data are computed as the 5th and 95[th] quantile of the probability density function (Folland and Anderson, 2002). I suggest to use: "The uncertainty intervals in the return period (estimated empirically?) of the historical data are computed via the 5th and 95th quantile of the probability density function"

Changed (see section 4.2)

P10 l2 low tail dependence. Gaussian and Frank copula have zero tail dependence, not "low". It may be helpful to better introduce the tail dependence in a sentence where you talk about it for the first time. That has been corrected. I have also introduced the tail dependence definition in section 3.3.

P10 l2 On the other hand, the low impact of the other copula families is due to the fact that the extreme hazard values are mainly driven by the large dependencies between nearby cells, especially at the first tree levels. Could you please argue this better?

The low impact of the Gaussian, Clayton, and Frank copulas are due to the fact that these copulas do not show upper tail dependence in the limits (see page 18, lines 7-16). The figure has also been corrected (Figure 8b).

P10 l16 However, recent studies suggest that cold weather in the UK is likely to be less severe, to occur less frequently, and to last for a shorter period of time than was historically the case due to anthropogenic induced climate change (on Climate Change, 2017). I would already mention here that there is debate about this (as you then specify in the next paragraph).

This has been corrected (see section 4.2.1)

P11 l7 As shown in Figure 8, South England is in general warmer than the North England and Northern Ireland region, partially driven by the urban micro-climate effect of the London area. The 1962/63

winter was less extreme in this region (wAFI of 139° C) with an estimated return period of 1 in 79 years. On the other hand, Scotland is usually significantly colder than the rest of UK, reaching for example AFI values of 100 ◦ C almost 2 times more often.

Please, make more clear in the text (and in the figure captions) when you talk about AFI, wAFI, averaged non-weighted AFI (and in which area is computed the average (UK, or sub-regions)). Also, when introducing eq. (3), I suggest to anticipate that you are going to use the wAFI both on UK and subregional scale.

I have updated the text and added mAFI when discussing averaged non-weighed AFI and make this more clear.

Figure captions. Please improve the Figure captions with more information. For example in Fig 2 what is the NAOI (North Atlantic Oscillation Index)?

I have tried to improve all the figure captions (including Figure 2 and NAOI)

**Technical corrections**

P3 l6 desribed. Described

P4 l1 that winter. You may use "winter 1978/79".

Fig 4 and 5. Could you please use the same scale range, i.e. 0-400°C

P10 l2 familes. Families

Corrected

**REVIEW 2**

**GENERAL COMMENTS.**

The paper is an interesting one, and outlines an original multivariate investigation concerning subfreezing temperatures. The comments posted by Referee 1 already provide an excellent, detailed review, with which I (almost) fully agree. Below, please find further notes: my objections should be read as constructive advices. Some relevant bibliography is reported at the end of this review.

1. I noticed that there is some confusion between the notions of probability distribution function and probability density function (e.g., Page 10, Lines 5–7: "The uncertainty intervals in the historical data are computed as the 5th and 95th quantile of the probability density function (Folland and Anderson, 2002)"). The probability distribution function is the integral of the probability density function (if it exists). The quantiles are the inverses of the probability distribution function (a nondecreasing one), not of the density function (which may not even be monotone). The Author must check the paper and fix all the points where such a confusion arises, otherwise the paper is not correct from a probabilistic point of view.

I made sure that probability density/distribution function is not used erroneously.

2. I was puzzled by the comment of Referee 1 concerning the sample size, and I ask the Author to clarify the issue: here, 170 variables are at play, each observed 51 times. To the best of my understanding, the idea revolving around Vine copulas is that any multivariate density can be decomposed into a (suitable, maybe not unique) product of univariate marginal densities and bivariate copula densities: in turn, only univariate and bivariate fits should be needed, isn't it? Thus, apparently, the fitting problem may not be so severe. Clearly, trying and fitting the upper tail of a GEV law using only 51 observations may be difficult (although the TWMLE escamotage is used), but it may not be impossible. Similarly, trying and fitting a bivariate copula using only 51 pairs may not be advisable, but it is not uncommon in practical applications. Overall, should my interpretation be correct, the game played by the Author may not be a "Mission Impossible", rather an "Uncertain Mission". . . Thus, I kindly ask the Author to clearly explain the situation, and to provide estimates of the uncertainties as explained below.

This question is related to the main comment of the first reviewer. Please read the reply in the beginning of this document. As suggested by the reviewer, I have calculated the model uncertainty using a bootstrap approach. I have also added a section discussing the the sources of the model uncertainty in more detail (Section 4.2.3)

3. I definitely agree with the comment of Referee 1 concerning the procedure to estimate the uncertainties (Page 9, Lines 23–ff.). As a rule of thumb, 1000 independent repetitions of the 10,000-years Monte Carlo simulations are usually suggested in literature, in order to provide "reasonable(?)" estimates of the confidence intervals of interest (clearly, it may be adjusted depending on the computational burden).

I have calculated the uncertainty associated solely due to the Monte Carlo sampling as suggested by the reviewer (see section 4.2.3)

- Using the selected RVM, the simulation of 10K years of winter seasons is repeated for 1,000 times.
- For each of these 1,000 simulations, the corresponding return period levels are calculated.
- The uncertainty in the return periods is estimated by identifying the 95\% confidence interval (i.e. the range 2.5–97.5 \%) from these 1,000 return level curves.

The results are discussed in Section 4.2.3.

4. My main "perplexity" concerns a methodological issue. In this work, I can see the Mathematics/ Statistics, but I do not see the Physics, which, instead, should be the starting point. To be clear, and to the best of my knowledge, the procedure used to construct the 170-dimensional copula finds its justification in an aggregation/clustering algorithm based solely on statistical considerations (Page 9, Lines 13–14: "The method follows an automatic strategy of jointly searching for an appropriate R-Vine tree structure"). If I remember it correctly, the algorithm is based on the Kendall and/or on the Kendall Distribution Function K, and/or, in general, on the strength of the statistical association between the variables at play. While interesting and meaningful from a mathematical point of view, such a procedure may eventually (statistically) associate grid cells having little, or negligible, physical link (for instance, could this be the case of the grid cells corresponding to Edinburgh and London, quite far apart from a spatial and a climatic point of view?) In other words, important information like, e.g., the latitude (corresponding to different climatic regions) may not be considered/used by the statistical procedure adopted for constructing the overall copula. The Author is kindly asked to discuss the issue, and to provide suitable justifications. Is it possible to modify the construction of the 170-dimensional copula in order to take into account the physics of the phenomenon?

The dominant statistical associations in the model are mainly driven by a physical link: the large scale circulation, which is the driver of winter temperatures in UK. Notice that large dependencies, with Kendall's tau coefficients greater than 0.90, are found as expected between neighbouring cells, but also remain important across the whole model domain due to the nature of the hazard: AFI assess the freezing temperatures during the entire winter and, thus, is less associated with small scale local phenomena that can cause important spatial variation. This for example can be seen in Figure 2 below, where the 51-year long observed AFI values over London correlate significantly with all the remaining UK cells with linear correlation coefficients above 0.5.

Moreover, the dominant large scale mode of variability in the Euro-Atlantic region is the North Atlantic Oscillation, which I tried to include in the GEV fits (see section 4.2.2); however, including it was not improving the model fits, possibly due to the quite noisy character of the phenomenon and the relatively short historical record used in this study.

Indeed, the effect of NAO in the hazard dependency structure has not been taken into account here. Recently, Bevacqua et al. (2017); Bevacqua (2017) developed a methodology that offers the possibility

to include such meteorological predictors in a vine copula model and is something to 5 be addressed in a future study.

[Figure]

**Correlation map**

Figure 2: Linear correlation map between the 51-year long AFI values in a cell over London (blue dot) and all other cells of the model domain.

A section (4.2.2) discussing the NAO influence has been added.

5. The Author has modeled the historical data, but, should the climate be really changing, then (at least from an Insurance point of view) the Author should account for it in his model, e.g. by introducing (in the long term simulations) suitable temporal patterns in the GEV/copula parameters according to available projections of the future climate (like, e.g., in IPCC scenarios). A comment is required on this issue.

From an Insurance point of view, the focus is mainly in the next year or sometimes a bit longer (but up to 4 years) depending on the (re) insurance contracts. Therefore, the long term changes in the climate based on future climate projections are somewhat less of interest.

However, what is important is that the model here has been constructed under the assumption of a stationary climate, i.e. under the assumption that the climate has not changed (significantly) during the last 51 years. To test the non-stationarity assumption, a linear covariate is incorporated in the location parameter of the GEV distributions: $\mu = \beta_0 + \beta_1 \times$ year. The resulting parameters were not significantly different from zero, indicating an unsubstantial linear trend in AFI during the last five decades. Due to its high year-to-year variability, longer monitoring records are needed to identify 5 statistically significant trends.

I have added this discussion in Section 4.2.3 (page 18).

6. In Section 3.1 "Results and discussion", the Author mentions the actual debate about climate changes (already commented by Referee 1). I would suggest to take a look at a recent paper by Vezzoli et al. (2017), where the traditional validation criteria of climate models are discussed, and an advanced/ thorough distributional perspective is outlined: it may partially explain why several crucial hypotheses are "still largely under debate" (as claimed by the Author and Referee 1), and may partially account for the general inability to draw up clear settlements.

Thank you I take on board this comment.

**SPECIFIC COMMENTS.**

Page(s) 2, Line(s) 23–ff.

For the benefit of unskilled readers and practitioners, here the Author should provide general references involving seminal books, papers, and guidelines concerning copulas, like writing: "For a theoretical introduction to copulas, see Nelsen (2006); Joe (2014); Durante and Sempi (2015); for a practical/engineering approach and guidelines, see Genest and Favre (2007); Salvadori and De Michele (2007); Salvadori et al. (2007, 2014, 2015)". Instead, citations concerning Vine copulas, being more specific and related to the modeling outlined in this work, may be postponed later. Page(s) 9, Line(s) 20–22.

I have updated the manuscript as suggested (page 8, lines 5-7).

Author. "Goodness-of-fit is performed for the final selected R-Vine Model (RVM) based on the RVineGofTest algorithm of the same R package (Schepsmeier, 2013). The Cramer von Mises test, which compares the empirical copula with the RVM, has a value of 0.019 and a p.value = 1, which indicates that the fitted RVM cannot be rejected at a 5% significance level."

Referee. I am puzzled by such a large p-Value: in my opinion, it may entail a large probability of Type II error, i.e. accepting a False Null Assumption (this a typical performance of Cramer-von- Mises and similar tests, when the sample size is insufficient). The Author is kindly asked to discuss the issue, and to provide suitable justifications.

Unfortunately, the gof tests for Vine copulas show poor behavior in small sample sizes and also at higher dimensions, as is the case for this work (Schepsmeier, 2013). I should also mention that different gof methods implemented in the VineCopula R package (via the function RVineGofTest) can show very different values for the CvM statistic as shown in the table 2 below. I have also tried to include a range in the GoF test values by resampling the historical AFI years (sampling 51 years with replacement x 10 times). Due to the long runtime, it has not been possible to sample more times; the min/max ranges are shown in parenthesis for the ECP2 and ECP methods in the table below. It is even harder to estimate the power of these tests. Thus, my main justification would be the performance of the model in comparison to the empirical estimates: the simulated RPs follow reasonable well the empirical (historical) estimates not only at the entire UK but also at smaller regions.

This discussion has been added in section 3.3.1.

**Table 2:** GOF valus for the CvM statistic based on different methods implemented in the VineCopula R package

[revised manuscript text omitted]

---

## Referee Report (RR1)

Journal: Natural Hazards and Earth System Sciences (NHESS)
Revised version of: "A hazard model of subfreezing temperatures in the United Kingdom using vine copulas"
Author(s): Symeon Koumoutsaris
MS No.: nhess-2017-389
MS Type: Research article

**General comment**

I found the paper improved, and I appreciate the effort of the author in addressing the issues about the uncertainty. There are a couple of steps in the procedure employed for computing the uncertainties which are not fully clear to me. These steps might be potentially important. In principle, these steps might substantially affect/increase the computed model uncertainty. After these are addressed, I would suggest considering the paper for publication.

In the following, I will refer to the pages and lines of the pdf file including the corrections (in blue and red). My revision should be read, again, as a constructive advice.

**Comments related to uncertainty quantification**

P1 l8-9 "The model suggests that the extreme winter 1962/63 has a return period of approximately once every 89 years, with 95% confidence intervals between 81 to 120 years. However, the relative short record length together with the unclear effects of anthropogenic forcing on the local climate add considerable uncertainty to this estimate."

Given the "However", I am not sure that it is fully clear, here, that the uncertainty (95% CI) is due to the shortness of the data. In principle, the purpose of the uncertainty quantification is to account for the model uncertainty due to the shortness of the data. I see that you write in the following sentence "add considerably uncertainties", which might be related to acknowledging that the employed procedure to compute the model uncertainty does not account for all of the uncertainties due to the short data length. But this sentence might be improved.

P11 l13 "Both together result in a virtual reduction in the dimensions of the pdf." As I wrote in my first comment: "The author says that he is using many independent copula: if this is a reasonable choice then it corresponds to somehow virtually reduce the dimension of the pdf." I would like to observe that I employed the "somehow virtually reduce" expression in the response, however, I have never seen this used in the literature.

P12 l 1. Section 3.1.2. You might consider changing the title of the section, referring to the uncertainties. In fact, this section explains the procedure to compute the uncertainties.

P12 l2 "The RVM is used to simulate 10K years of winter-seasons in the UK. For each year, the simulated AFI values at each grid cell depend on the other cells based on the fitted RVM."

To guide the reader, I would explain why the model is used to simulate a so long sample. To reduce the uncertainties associated with the simulation (as explained in my previous comment, see the end of this document).

P12 l6 "Following Bevacqua et al. (2017), the model uncertainty is assessed using a parametric bootstrap approach..."
[Definition for the following discussion: Let's define procedure1 and procedure2 the two procedures you present on page 13.]
Procedure1. While Bevacqua et al. consider the uncertainties of the marginal pdfs during the procedure, it is not clear to me whether these are accounted for here. **Thus, I am wondering if procedure2 is used to compute the uncertainty associated with the RVM only, or the uncertainty of the full model, i.e. of the joint pdf.** Specifically, going through the first 2 steps of procedure1, it is not clear to me whether you (a) simulate the real data (real, i.e. you transform the uniform variables simulated from the vine using the inverse marginal pdfs) and fit again both the marginals and the RVM to these "real" data, or (b) you simulate only the uniform variables and fit the RVM to them only.

If the procedure is (b), then this is different from the cited Bevacqua et al., and then I think that it should be stated (note that also procedure2 is an addition with respect to Bevacqua et al., but this is not clear). Other differences that would need to be highlighted:
- P12 l7 "...data from the selected RVM." in Bevacqua this is "...data from the selected joint pdf".
- Similarly at p13 l1. "In the selected RVM" in Bevacqua this is "...in the selected joint pdf".
- Similarly at p13 l4. "A new RVM is fitted…" in Bevacqua this is "...a new joint pdf is fitted (via vines)...".

Also, if you do not account for the uncertainty of the marginals (i.e. if you follow (b)), then I recommend to not talk about "model uncertainty (e.g., in line 6), but of RVM uncertainty only. However, the following comment is relevant.

**The obtained uncertainty associated with the "model" seems very small (as you also argue later (p19 l30)). You might agree with me that this might be unexpected, given the small sample size. Thus, I am wondering if they are the uncertainty associated with the RVM only, or the uncertainty of the full model, i.e. of the joint pdf. Specifically, I am wondering about: (1) how the model uncertainty would actually be affected by the uncertainty of the marginals (if you do not account for this already, i.e. if you follow (b)); (2) how the model uncertainty increases when the RVM structure is not fixed in procedure1 (step2).**
**Is there any reason for not considering these two uncertainties? (Again, maybe you already accounted for the marginal uncertainty (1)).**
**I can see that accounting for all of these uncertainties might be cumbersome strictly following procedure1. To my understanding, an easier alternative to procedure1 (to account for all the model uncertainty, i.e. to also account for (1) the marginal uncertainties and (2) the RVM structure uncertainties), you might consider the following: Applying procedure2\*\*\*, but simulating 51 years of data (instead of 10k years). This alternative procedure should, in fact, give similar results to applying procedure1 (where also (1) the marginal uncertainties and (2) the RVM structure uncertainties are considered). In this case, it is clear that you would obtain larger**

**uncertainties than obtained via the employed procedures (as you would procedeed as done for procedure2 but employing a much shorter sample).**

***Clearly, the "real" data should be simulated, i.e. one should simulate the uniform variables from the vine, and then transform them into "real" variables employing the inverse marginal CDFs.

Consideration. To my understanding, showing the Monte Carlo uncertainty (procedure2) in the paper helps to see that the RVM uncertainty (procedure1) is almost the same as the Monte Carlo uncertainty, and therefore you can conclude that the RVM uncertainty is negligible. I see the reasoning, and in principle I like it; however, see the previous discussion about the RVM uncertainty which might become larger if computed differently. Otherwise, personally, I have difficulties in seeing a reason for describing and employing procedure2. **Thus, the reader should be helped to understand the differences between the two uncertainty procedures, e.g. explaining why they are both computed.**

P17 l24-25
According to me, a comparison between purely Monte Carlo uncertainties (obtained simulating an as long as possible sample size) and uncertainties of the "empirical curve" is not conceptually meaningful. As stated in the previous comment revision (see the end of this file), the purely Monte Carlo uncertainties (obtained simulating an as long as possible sample size) is meaningful only to quantify the uncertainty driven by the limited length of the simulation (from a given a pdf that might be assumed to be non-biased).

Instead, it makes sense to me to compare the uncertainty computed using procedure1 with the uncertainties of the "empirical curve". (As stated in the previous pages, I see a sense in comparing uncertainties form procedure1 and procedure2 for stating that the RVM uncertainties are negligible. However, I discussed potential issues of procedure2 above).

P17 l26. "The accuracy can be improved by increasing the number of simulated years, but at a computational cost". I am not comfortable with the message that might be taken from this sentence. The purely Monte Carlo uncertainty can be reduced by simulating long samples, but it should be clear that this is not related with the uncertainty of the model (in a general case).

**Other comments**
P3 l25 "(1)"
Please, write "equation (1)" or "eq. (1)".

Eq (1):
Write AFI_Year maybe?
Should the AFI_Year be defined as =0 if there are no days with negative temperatures? It is currently not exactly defined in this case, while you refer to f(x) for x=0 in equation 3.

Figure 2. Not necessary, but you might consider plotting the -NAOI rather than the NAOI (or -mAFI) to highlight the correlation between the time series.

P5 l8 "exceed"? P(X<=x)

P6 l15 "in order to geographically smooth the GEV.."
You might explain the reason for wishing to have smoothed parameters.

Fig 3 correct ":,"

P7 l4 I would write: "The largest observed AFI..."

Table1 caption. "Cell id"?

P9 l1 Please, use "The probability density function (pdf) of X, ...". Also later you talk about "densities". Later, I suggest using pdf.

P9 l6 "copula density" instead of "copula function"?

P10 l10 "eq 2 and 10" should be eq 2 and 3.

P10 l14. Is this only an intuition? Anyway, you might rephrase.

P10 l15 Please, define what a tree is, as it would help the reader. You might "use" the 4-dim example to explain what a tree and a first tree are. You might consider using the term "tree" or "level" only in the full text, as you refer to the same thing with these two different words, and this might confuse the non-expert reader (e.g., p10 l23-24).

P11 l6 "largest contribution at the second level". Add something like "after the independent copula".

P17 l15 average AFI, please: add (mAFI)

P17 l20 "However, the non-stationary fits were statistically similar to the stationary ones, with β1 parameters not significantly different from zero."
You might write: "Despite the significant anticorrelation found between the average AFI (mAFI) and the NAOI, the non-stationary fits were statistically similar to the stationary ones, with β1 parameters not significantly different from zero."
Then the next sentence ("This is probably related to the quite noisy character of the phenomenon and the relatively short historical record used in this study, which makes it difficult to discern the statistical differences in the extreme temperatures between positive and negative NAO winters") could be rephrased, maybe explicitly referring to the noise as a function of the spatial scale (in fact, the noise is not visible when looking at the average AFI (mAFI), as the correlation between mAFI and NAOI is about -0.6).

P19 l10 (a) and (c) are pretty similar: you might unify them. Furthermore, as previously discussed, the the full multivariate pdf (marginals and copula) has uncertainties, and not only the copula (RVM) (as it looks from c).

P20 l13
In these cases (fig8b), is the RVM structure always the same as the RVM structure used in the full study so far? Are there independent copulas in the RVMs used for the sensitivity study? Please, very briefly specify these details.

Here I paste a comment I gave in the previous revision. This might be useful, given the comment I have written in this review.
"The 10,000 years time series should be long enough to neglect uncertainties associated with the Monte Carlo simulations (which is the method used for extracting the return period associated with the fitted parametric pdf) (Serinaldi et al. (2015) and Bevacqua et al. (2017)). [One should ensure if the sample is "long enough" via repeating the (10,000 years) simulations several times and checking if the there are differences in the estimated return period (if there are no differences, the 10,000 years sample is long enough)]. Performing a long enough simulations allows one to get a convergence to the true return period that one would get analytically from the fitted pdf (given the complexity of the problem it is impracticable to get an analytical derivation of the RP). Performing a long simulation does not solve the issue about the model uncertainties (uncertainties existing about the pdf), which is there because the pdf is calibrated on a finite - very short - sample. I suggest to discuss this in a way to make difference between these different type of uncertainties. "

Best regards.

---

## Referee Report (RR2)

Journal: Natural Hazards and Earth System Sciences (NHESS)
Second revised version of: "A hazard model of subfreezing temperatures in the United Kingdom using vine copulas"
Author(s): Symeon Koumoutsaris
MS No.: nhess-2017-389
MS Type: Research article

Dear author and editor,

overall I found the paper improved. In particular, I am satisfied with the changes that were done with respect to my previous comments. I would suggest trying to improve the quality of the writing, maybe asking some colleague to read the paper would help (as this is a single author paper). While I think that the paper improved with respect to the previously presented analysis, the author introduced a non-stationary model. I like the idea of employing this model, but I have some main comments about it. According to me, the author should provide a discussion about some issues. However, if the discussion cannot be satisfying, I would suggest considering the publication of the paper without the non-stationary model.

**Main comments about the non-stationary model.**
- I have concerns regarding using the $CO_2$ as a predictor for the local UK warming. While global average warming is correlated with $CO_2$ concentration, the global average warming can differ from the average local warming and even more from the local (potential) warming of the extreme cold events (being the latter the focus of the study). Therefore, I think that using a predictor of global warming ($CO_2$) as a predictor for changes of the local cold extreme temperature in the UK is problematic. Especially when the author extrapolates information about the future climate based on changes in $CO_2$. The author himself highlights that under global warming (increasing $CO_2$) the Arctic amplification has been occurring, and this might lead to non-linear responses of the local UK climate. In particular, it is under debate whether very cold winters might be experienced in the UK in the future as indirect result of the increasing average global temperature. Thus, it is clear that potentially relevant physical mechanisms are not included within the selected predictor ($CO_2$). This, might lead to a mismatch between the predicted future climate and the real future climate. Thus, personally, I would not use the $CO_2$ as a predictor at all, and especially I would not present results for the future. An alternative would be to strongly highlight the limitations or - if possible - justify the use of this predictor, and in any case to have a dedicated discussion on that. I believe that the decision is up to the editor.
- Also, the non-stationary model does not consider potential the non-stationarity in the dependencies described by the joint pdf, and this would need to be highlighted better (at the moment this is only discussed at the end of the results for NAO only, and not for $CO_2$, while this would need to be said in the methods too).
  Also, according to my understanding, the dependencies described by the copula in this model are not typical spatial dependencies between the locations as in the stationary model, and this is not discussed anywhere. (The marginal pdfs include predictors through linear models. Thus, to my understanding, the uniform variables

(obtained from the marginals) modelled by the copula are not the usual ranks associated with the AFI in each location. It seems to me that the uniform variables are rather the ranks of the residual of the linear models. However, this in the parenthesis is only my interpretation, while I think that the reader should not interpret this, rather it would be better if the author provides an explanation.) I think that a discussion on this is needed, such that it is made clear what the model is actually modelling (also considering whether this might lead to limitations).

**Minor comments:**

P1 l9, rephrase

P1 l10, "such an event"?

P2 l4, I would delete "entire". As also the author says later, the NAO affects mostly the weather around the Atlantic basin. The AO, on the other hand, influences the weather over the entire Northern Hemisphere.

P3 l18, this sentence about the wind is confusing me and seems, to me, out of context.

P5 l13, at the end: "period" -> "analysed period"?

P6 l1 I would start with "Based on the AFI, the winter ..."

P10 l5, it should be made clear in the text that also the copula is stationary.

P11 l11 Please, consider to specify the percentages for mu, P0, and in total separately if you think that this would lead to some interesting additional considerations.

Paragraph P15 l4
- Should "RVM" be "pdf"?
- Sentence "Performing...", rephrase
- "long enough to neglect the Monte Carlo uncertainty"

(P15 l15 Thought: it could have been possible to employ a more standard RCP scenario for future, defined based on the DeltaF, rather than on the change in CO2.)

P16 l14
"Instead of 100K", please make a clear link to what is said at the beginning of the section, where you say that 100K would be necessary to reduce the Monte Carlo uncertainty.
Probably you could simply move the Paragraph on P15 l4 here.

P16 l14
I would start the sentence with "In order to investigate…" and then say that you separate the uncertainties.

Section 4.1,
- Please, make clear that you are referring to the mean return period when you say "stochastic set"
- I guess that you are simulating from the vine and then transforming the simulated uniform variables to the "real" via the marginal models containing the predictors/covariates. Here you fix the $CO_2$ predictors to certain values, but what value is given to the NAO predictors? Please, make this clearer. Also, please make this procedure clear in the method section.

Fig10 The grey uncertainties are not well visible on my printed version. Please, check if this is only a problem of mine.

P20 l11 "originates", I would write "appears to originate"

Table 4 missing the unit: "years"

P21 l1 see the main comment on $CO_2$ above

P21 l12 "return period of 1 in 39 years", is it not enough to write "return period of 39 years"? Also in other parts of the paper?

P21 l12 How is it a positive and negative phase defined? What is the value of NAO that is used to sample the data? In the figure it is written > or < 1, but do not you use a unique value, e.g., 1 and -1? Please, make this clear.

P23 l3. The predictor influence on the dependence is not considered. Indeed it is specified here only for the NAO, but not for the $CO_2$.

P24 l9 the occurrence has increased? The return period has increased. The same in the abstract.

P24 l16 "such extreme events", not clear which extreme events.

Best regards.

---

## Author Response (AR2)

Journal: Natural Hazards and Earth System Sciences (NHESS)
Revised version of: "A hazard model of subfreezing temperatures in the United Kingdom
using vine copulas"
Author(s): Symeon Koumoutsaris
MS No.: nhess-2017-389
MS Type: Research article

**General comment**
I found the paper improved, and I appreciate the effort of the author in addressing the issues
about the uncertainty. There are a couple of steps in the procedure employed for computing
the uncertainties which are not fully clear to me. These steps might be potentially important.
In principle, these steps might substantially affect/increase the computed model uncertainty.
After these are addressed, I would suggest considering the paper for publication.
In the following, I will refer to the pages and lines of the pdf file including the corrections (in
blue and red). My revision should be read, again, as a constructive advice.

I would like first of all to thank the reviewer for his valuable comments and corrections. In particular,
with respect to the uncertainty calculation, indeed I erroneously computed them (by simulating only
the uniform variables and fit the RVM to them only as the reviewer has thought so), which severely
underestimated the confidence intervals.

After computing the confidence intervals correctly, the resulting uncertainty is quite large, similar to
the empirical one. In order to reduce this large uncertainty, I have decided to:

a) Use a longer historical data set (reanalysis data of 110 years instead of 51 years previously)
b) Lower the resolution in order to decrease the dimension of the joint pdf (67 cells instead of
170 previously)

The resulting confidence intervals however are still quite large. Nevertheless, the longer dataset
enabled me to include some physical parameters in the model (notably the influence of NAO and of
climate change) that I found useful. I reply to the comments one-by-one below.

**Comments related to uncertainty quantification**
P1 l8-9 "The model suggests that the extreme winter 1962/63 has a return period of
approximately once every 89 years, with 95% confidence intervals between 81 to 120 years.
However, the relative short record length together with the unclear effects of anthropogenic
forcing on the local climate add considerable uncertainty to this estimate."
Given the "However", I am not sure that it is fully clear, here, that the uncertainty (95% CI) is
due to the shortness of the data. In principle, the purpose of the uncertainty quantification is
to account for the model uncertainty due to the shortness of the data. I see that you write in
the following sentence "add considerably uncertainties", which might be related to
acknowledging that the employed procedure to compute the model uncertainty does not
account for all of the uncertainties due to the short data length. But this sentence might be
improved.
I rephrased to make it more clear:
"However, the estimated uncertainty in these results is quite large and comes
from the relatively short record length. Moreover, possible spurious trends in the historical data add
considerable uncertainty to these estimates, as well."

P11 l13 "Both together result in a virtual reduction in the dimensions of the pdf." As I wrote in
my first comment: "The author says that he is using many independent copula: if this is a
reasonable choice then it corresponds to somehow virtually reduce the dimension of the
pdf ." I would like to observe that I employed the "somehow virtually reduce" expression in the

response, however, I have never seen this used in the literature.
I took out this comment.

P12 l 1. Section 3.1.2. You might consider changing the title of the section, referring to the uncertainties. In fact, this section explains the procedure to compute the uncertainties.
I have renamed the section as "Stochastic simulation and uncertainty estimation via parametric bootstrap".

P12 l2 "The RVM is used to simulate 10K years of winter-seasons in the UK. For each year, the simulated AFI values at each grid cell depend on the other cells based on the fitted RVM."
To guide the reader, I would explain why the model is used to simulate a so long sample. To reduce the uncertainties associated with the simulation (as explained in my previous comment, see the end of this document).

I have increased the simulation period (to 100K) and added the following sentences: "Performing long enough simulations is necessary in order to obtain converged numerically results, i.e. to convergence to the "true" return period. Our focus here is the 200 year RP, which is commonly associated with capital and regulatory requirements. By repeating the simulation several times, it has been assessed that 100K years of winter seasons is long enough and the Monte Carlo simulation error is negligible. "

P12 l6 "Following Bevacqua et al. (2017), the model uncertainty is assessed using a parametric bootstrap approach..."
[Definition for the following discussion: Let's define procedure1 and procedure2 the two procedures you present on page 13.]
Procedure1. While Bevacqua et al. consider the uncertainties of the marginal pdfs during the procedure, it is not clear to me whether these are accounted for here. **Thus, I am wondering if procedure2 is used to compute the uncertainty associated with the RVM only, or the uncertainty of the full model, i.e. of the joint pdf.** Specifically, going through the first 2 steps of procedure1, it is not clear to me whether you (a) simulate the real data (real, i.e. you transform the uniform variables simulated from the vine using the inverse marginal pdfs) and fit again both the marginals and the RVM to these "real" data, or (b) you simulate only the uniform variables and fit the RVM to them only.
If the procedure is (b), then this is different from the cited Bevacqua et al., and then I think that it should be stated (note that also procedure2 is an addition with respect to Bevacqua et al., but this is not clear). Other differences that would need to be highlighted:
- P12 l7 "...data from the selected RVM." in Bevacqua this is "...data from the selected joint pdf".
- Similarly at p13 l1. "In the selected RVM" in Bevacqua this is "...in the selected joint pdf".
- Similarly at p13 l4. "A new RVM is fitted…" in Bevacqua this is "...a new joint pdf is fitted (via vines)...".
Also, if you do not account for the uncertainty of the marginals (i.e. if you follow (b)), then I recommend to not talk about "model uncertainty (e.g., in line 6), but of RVM uncertainty only. However, the following comment is relevant.
**The obtained uncertainty associated with the "model" seems very small (as you also argue later (p19 l30)). You might agree with me that this might be unexpected, given the small sample size. Thus, I am wondering if they are the uncertainty associated with the RVM only, or the uncertainty of full model, i.e. of the joint pdf. Specifically, I am wondering about: (1) how the model uncertainty would actually be affected by the uncertainty of the marginals (if you do not account for this already, i.e. if you follow (b)); (2) how the model uncertainty increases when the RVM structure is not fixed in procedure1 (step2).
Is there any reason for not considering these two uncertainties? (Again, maybe you already accounted for the marginal uncertainty (1)).**

**I can see that accounting for all of these uncertainties might be cumbersome strictly following procedure1. To my understanding, an easier alternative to procedure1 (to account for all the model uncertainty, i.e. to also account for (1) the marginal uncertainties and (2) the RVM structure uncertainties), you might consider the following: Applying procedure2\*\*\*, but simulating 51 years of data (instead of 10k years). This alternative procedure should, in fact, give similar results to applying procedure1 (where also (1) the marginal uncertainties and (2) the RVM structure uncertainties are considered). In this case, it is clear that you would obtain larger uncertainties than obtained via the employed procedures (as you would procedeed as done for procedure2 but employing a much shorter sample).**

\*\*\*Clearly, the "real" data should be simulated, i.e. one should simulate the uniform variables from the vine, and then transform them into "real" variables employing the inverse marginal CDFs.

As mentioned above, indeed I had computed erroneously the confidence intervals. This has been now corrected. More precisely, as explained in the revised manuscript, the confidence intervals are computed now as follows:
- A simulation with the same length as the observed data (i.e. 110 years) is repeated for B = 500 times.
- For each of these B = 500 samples, a new full model is fitted (including new GEV and logistic regression model parameters at each cell and new RVM structure, pair-copula families and parameters) following the methodology described in sections 3.2.1 and 3.3.1.
- For each of the resulting B = 500 RVMs, a simulation of 10K years of winter-seasons is performed. The uniform variables are then transformed using the (new) inverse marginal pdfs and the corresponding return period levels are estimated.
- The uncertainty in the return levels is estimated by identifying the 95% confidence interval (i.e. the range 2.5–97.5 %) from these 500 return level curves.

Due to computational constraints, confidence intervals are computed only for the stationary model and the simulation length has been reduced to 10K years (instead of 100K). In order to separate the uncertainty associated with the RVM only from the uncertainty of the full model, i.e. of the joint pdf, confidence intervals have been also calculated with the same approach described above, but using the same marginal pdfs in each bootstrap repetition.

Furthermore, I would like to note that I have also computed the confidence intervals using the alternative procedure proposed by the reviewer above and indeed I get the similar results to the procedure above (as the reviewer also suggested). This method is less computer intensive however, the resulting confidence intervals only up to 110 year RP (the historical record length) and thus it does not give a complete picture of the (large) uncertainty. In addition, I wanted to separate the uncertainty originating from the RVM only and compare it with the full model uncertainty, which is not possible with this alternative procedure.

Consideration. To my understanding, showing the Monte Carlo uncertainty (procedure2) in the paper helps to see that the RVM uncertainty (procedure1) is almost the same as the Monte Carlo uncertainty, and therefore you can conclude that the RVM uncertainty is negligible. I see the reasoning, and in principle I like it; however, see the previous discussion about the RVM uncertainty which might become larger if computed differently. Otherwise, personally, I have difficulties in seeing a reason for describing and employing procedure2. **Thus, the reader should be helped to understand the differences between the two uncertainty procedures, e.g. explaining why they are both computed.**
I understand the reviewer's point and I have indeed increased the number of simulation years to 100K (instead of 10K) which makes the Monte Carlo uncertainty negligible.

However, some (small) Monte Carlo uncertainty might still be present in the computed confidence intervals since for computational reasons I had to use a shorter simulation period (10K).

P17 l24-25
According to me, a comparison between purely Monte Carlo uncertainties (obtained simulating an as long as possible sample size) and uncertainties of the "empirical curve" is not conceptually meaningful. As stated in the previous comment revision (see the end of this file), the purely Monte Carlo uncertainties (obtained simulating an as long as possible sample size) is meaningful only to quantify the uncertainty driven by the limited length of the simulation (from a given a pdf that might be assumed to be non-biased).
Instead, it makes sense to me to compare the uncertainty computed using procedure1 with the uncertainties of the "empirical curve". (As stated in the previous pages, I see a sense in comparing uncertainties form procedure1 and procedure2 for stating that the RVM uncertainties are negligible. However, I discussed potential issues of procedure2 above).
I agree and I do not compare the two uncertainties anymore.

P17 l26. "The accuracy can be improved by increasing the number of simulated years, but at a computational cost". I am not comfortable with the message that might be taken from this sentence. The purely Monte Carlo uncertainty can be reduced by simulating long samples, but it should be clear that this is not related with the uncertainty of the model (in a general case).
I have deleted this sentence.

**Other comments**
P3 l25 "(1)"
Please, write "equation (1)" or "eq. (1)".
Corrected

Eq (1):
Write AFI_Year maybe?
Corrected

Should the AFI_Year be defined as =0 if there are no days with negative temperatures? It is currently not exactly defined in this case, while you refer to f(x) for x=0 in equation 3.
Corrected

Figure 2. Not necessary , but you might consider plotting the -NAOI rather than the NAOI (or -mAFI) to highlight the correlation between the time series.
I have changed the plot and I believe it is more clear now

P5 l8 "exceed"? $P(X<=x)$
Corrected

P6 l15 "in order to geographically smooth the GEV.."
You might explain the reason for wishing to have smoothed parameters.
I have added the following sentence: "The smoothing increases the sample size at each grid point, which thus leads to a more precise estimation of the parameters, especially for the shape parameter which is highly influential in estimating the hazard levels and high return periods."

Fig 3 correct ":,"
Corrected

P7 l4 I would write: "The largest observed AFI..."
I have deleted this sentence since this is discussed later on.

Table1 caption. "Cell id"?
Corrected

P9 l1 Please, use "The probability density function (pdf) of X, ...". Also later you talk about "densities". Later, I suggest using pdf.
Corrected

P9 l6 "copula density" instead of "copula function"?
Corrected

P10 l10 "eq 2 and 10" should be eq 2 and 3.
Corrected

P10 l14. Is this only an intuition? Anyway, you might rephrase.
I changed this to "based on the premise".

P10 l15 Please, define what a tree is, as it would help the reader. You might "use" the 4-dim example to explain what a tree and a first tree are. You might consider using the term "tree" or "level" only in the full text, as you refer to the same thing with these two different words, and this might confuse the non-expert reader (e.g., p10 l23-24).

I have added a figure 6 showing the tree for the 4-d case to make this more clear. I am also referring always to trees and not levels in the revised manuscript to reduce the confusion.

P11 l6 "largest contribution at the second level". Add something like "after the independent copula".
I have delete this sentence.

P17 l15 average AFI, please: add (mAFI)
Corrected

P17 l20 "However, the non-stationary fits were statistically similar to the stationary ones, with β1 parameters not significantly different from zero."
You might write: "Despite the significant anticorrelation found between the average AFI (mAFI) and the NAOI, the non-stationary fits were statistically similar to the stationary ones, with β1 parameters not significantly different from zero."
Then the next sentence ("This is probably related to the quite noisy character of the phenomenon and the relatively short historical record used in this study, which makes it difficult to discern the statistical differences in the extreme temperatures between positive and negative NAO winters") could be rephrased, maybe explicitly referring to the noise as a function of the spatial scale (in fact, the noise is not visible when looking at the average AFI (mAFI), as the correlation between mAFI and NAOI is about -0.6).
I have included the NAOI in the new model – this has been possible after a bug fix, via the implementation of a P0 model, and also due to the longer Reanalysis data set.

P19 l10 (a) and (c) are pretty similar: you might unify them. Furthermore, as previously discussed, the the full multivariate pdf (marginals and copula) has uncertainties, and not only the copula (RVM) (as it looks from c).
Indeed these are now unified.
P20 l13
In these cases (fig8b), is the RVM structure always the same as the RVM structure used in the full study so far? Are there independent copulas in the RVMs used for the sensitivity study? Please, very briefly specify these details.

The sensitivity tests had different RVM structure but included the independent copulas. However, I have deleted this section from the revised manuscript mainly due to computational reasons but also I don't believe they were adding much into the article (i.e. the results were following the theoretical tail dependences).

Here I paste a comment I gave in the previous revision. This might be useful, given the comment I have written in this review.

"The 10,000 years time series should be long enough to neglect uncertainties associated with the Monte Carlo simulations (which is the method used for extracting the return period associated with the fitted parametric pdf) (Serinaldi et al. (2015) and Bevacqua et al. (2017)). [One should ensure if the sample is "long enough" via repeating the (10,000 years) simulations several times and checking if the there are differences in the estimated return period (if there are no differences, the 10,000 years sample is long enough)]. Performing a long enough simulations allows one to get a convergence to the true return period that one would get analytically from the fitted pdf (given the complexity of the problem it is impracticable to get an analytical derivation of the RP). Performing a long simulation does not solve the issue about the model uncertainties (uncertainties existing about the pdf), which is there because the pdf is calibrated on a finite - very short - sample. I suggest to discuss this in a way to make difference between these different type of uncertainties. "

Best regards.

[revised manuscript text omitted]

---

## Author Response (AR3)

**Journal: Natural Hazards and Earth System Sciences (NHESS)**
**Second revised version of: "A hazard model of subfreezing temperatures in the United Kingdom using vine copulas"**
**Author(s): Symeon Koumoutsaris**
**MS No.: nhess-2017-389**
**MS Type: Research article**

Dear reviewer and editor,

I would like to thank you both for your thoughtful comments. Please find below my answers to your comments (in red) and a copy and a copy of the manuscript where all my changes are annotated.

With best regards,
Symeon Koumoutsaris

**1. Reviewer comments**

**Main comments about the non-stationary model.**
- I have concerns regarding using the CO2 as a predictor for the local UK warming. While global average warming is correlated with CO2 concentration, the global average warming can differ from the average local warming and even more from the local (potential) warming of the extreme cold events (being the latter the focus of the study). Therefore, I think that using a predictor of global warming (CO2) as a predictor for changes of the local cold extreme temperature in the UK is problematic. Especially when the author extrapolates information about the future climate based on changes in CO2. The author himself highlights that under global warming (increasing CO2) the Arctic amplification has been occurring, and this might lead to non-linear responses of the local UK climate. In particular, it is under debate whether very cold winters might be experienced in the UK in the future as indirect result of the increasing average global temperature. Thus, it is clear that potentially relevant physical mechanisms are not included within the selected predictor (CO2). This, might lead to a mismatch between the predicted future climate and the real future climate. Thus, personally, I would not use the CO2 as a predictor at all, and especially I would not present results for the future. An alternative would be to strongly highlight the limitations or - if possible - justify the use of this predictor, and in any case to have a dedicated discussion on that. I believe that the decision is up to the editor.

A dedicated discussion discussing the caveats but justifying the choice of the predictors is added in section 3.2.2.

I believe that the choice of CO2 (and NAO) as predictor is justified for the following reasons:

- **Global $CO_2$ concentration and the subsequent global warming is an important driver of temperature change in UK.** The global increase in temperature during the last century is mirrored in the UK climate: average annual UK temperatures over land and the surrounding seas have increased in line with global observations, with a trend towards milder winters and hotter summers in recent decades (UKCCRA, 2017). In terms of extreme cold events, Massey et al. (2012) and Christidis and Stott (2012) find that human influence has significantly reduced the probability of such a severe winter in UK. This study also finds a negative

correlation is found between the average UK AFI (mAFI) and $\Delta F_{CO_2}$ forcing ($\rho$ = -0.17, pval= 0.08).  Global climate modelling studies also suggest that increased greenhouse concentrations will move UK climate towards warmer, wetter winters and hotter, drier summers. The projected warming is estimated to be larger for the high emissions scenario compared to the medium or low cases, particularly during the second half of the 21st century (UKCCRA, 2017).

- **Covariates such as global mean temperature, $CO_2$ concentration, and indexes of natural variability (ENSO, NAO etc) have been employed in many studies (Edwards and Challenor, 2013).** In particular, global average temperature is often considered as a covariate in the nonstationary GEV since it is the most robust indicator of human-induced climate change with the strongest attribution of causes (Mondal et al., 2016). Notice that using either global average temperature or $CO_2$ concertation (or radiative forcing) is equivalent as these are highly correlated: The Pearson correlation coefficient between annual global average temperature and global average $CO_2$ concentration is 0.93.  For example, Hauser et al. (2016) analyzed extreme daily temperatures in western Russia using global mean temperature and time as covariates in the GEV location parameter. Grinsted et al. (2013) used non-stationary generalized extreme value analysis and found that global average surface temperature is a better predictor of Atlantic cyclone activity than local grid cell temperatures. Mestre et al. (2009) found that extremes of maximum temperatures in future climate show a strong rise in the location GEV parameter, which is largely driven by the increasing $CO_2$ concentration. Hurricane wind speeds have been modelled with respect to global mean temperature, NAO, El Niño Southern Oscillation (ENSO) and other indexes by Jagger and Elsner (2006). Risser and Wehner (2017) analyzed observed extreme precipitation by using two time-dependent covariates, total atmospheric $CO_2$ concentration and the ENSO Index. Several studies focusing on extreme precipitation have used global warming (Oldenborgh et al. 2016; Agilan and Umamahesh, 2017) or $CO_2$ concentration (Risser and Wehner 2017; van Oldenborgh et al. 2017) as covariates of their GEV parameters.

- **Both $CO_2$ and NAO are accurately measured.** Although a model that relies on global mean surface temperature may not have as strong correlations as the casual link is more indirect, it has the advantage that it does not rely on subtle regional patterns that are difficult for models to capture.

- **They provide a reasonable way to isolate the human and natural influences on extreme temperatures.** For the same reason as well, Risser and Wehner (2017) used total atmospheric $CO_2$ concentration and the ENSO Index to analyse observed extreme precipitation. While it would be possible to use a more locally defined metric of future change (such as the change in the mean UK temperature for example), this would include more unforced naturally occurring internal variability of the climate system, making it difficult to identify the changes that are driven by anthropogenic $CO_2$ emissions.

- **Finally, using a covariate such as the change in $CO_2$ forcing avoids the difficulty with determining the start of the trend and also results can be easily rescaled to different time period or emission scenario which is helpful for mitigation strategies.**

Nevertheless, there are some caveats on the choice of predictors which I discuss this in the revised document. At the same time, I agree with the reviewer extrapolating far in the future is particularly problematic, since it assumes that the trends will remain the same in the future. For this reason, I adjust the future climate scenario to a closer year in the future (year 2030). As suggested by the reviewer further down below I also use the RCP emission scenario to estimate the radiative forcing.

- Also, the non-stationary model does not consider potential the non-stationarity in the dependencies described by the joint pdf, and this would need to be highlighted better (at the moment this is only discussed at the end of the results for NAO only, and not

for CO2, while this would need to be said in the methods too).

I have added the following phrase in the methods section:

Notice, however, that the effect of NAO/ $CO_2$ on the residual hazard dependency structure (for example the fact that the AFI between two locations might be more or less correlated as a result of changing NAO/ $CO_2$ values) is not taken into account here. Recently, a methodology that offers the possibility to include such meteorological predictors in a vine copula model has been developed by Bevacqua (2017a, 2017b) and is something to be addressed in a future study.

Also, according to my understanding, the dependencies described by the copula in this model are not typical spatial dependencies between the locations as in the stationary model, and this is not discussed anywhere. (The marginal pdfs include predictors through linear models. Thus, to my understanding, the uniform variables (obtained from the marginals) modelled by the copula are not the usual ranks associated with the AFI in each location. It seems to me that the uniform variables are rather the ranks of the residual of the linear models. However, this in the parenthesis is only my interpretation, while I think that the reader should not interpret this, rather it would be better if the author provides an explanation.) I think that a discussion on this is needed, such that it is made clear what the model is actually modelling (also considering whether this might lead to limitations).

I have added the following paragraph in the methods section:

In the case of the stationary model, the vine copula is employed to model the entire spatial dependence of the AFI in the UK. On the other hand, the spatial AFI structure in the case of the non-stationary model is modelled in two ways: a) by quantifying the dependence on NAO/ $CO_2$ in each location, treating each location as conditionally independent, then inducing spatial dependence through the variation of NAO/$CO_2$ and b) by fitting the RVM model to all the residual dependencies associated with the AFI between the cells; these account for dependencies between cells resulting from other large-scale circulation patterns, and regional climate variability (e.g. due to effects of local orography, land-sea contrast, and small scale atmospheric features such as convective cells).

**Minor comments:**
P1 l9, rephrase done
P1 l10, "such an event"? done
P2 l4, I would delete "entire". As also the author says later, the NAO affects mostly the weather around the Atlantic basin. The AO, on the other hand, influences the weather over the entire Northern Hemisphere. done
P3 l18, this sentence about the wind is confusing me and seems, to me, out of context. done
P5 l13, at the end: "period" -> "analysed period"? done
P6 l1 I would start with "Based on the AFI, the winter ..." done
P10 l5, it should be made clear in the text that also the copula is stationary. As mentioned above I have added a phrase in section 3.3.1 (at the copula section).
P11 l11 Please, consider to specify the percentages for mu, P0, and in total separately if you think that this would lead to some interesting additional considerations.

Here below you can find a table with the percentages split.

|  | **P0** | **μ** |
| --- | --- | --- |
| **NAOI** | 72% | 55% |
| **ΔF** | 39% | 21% |

As described in the text, NAO is found to affect more cells in total in comparison to anthropogenic climate change, and this is also seen both for P0 and μ. The reason for that I believe is the same (the hidden climate change trend in the limited observational period) as is already mentioned in the manuscript so I prefer to not add this in the revised manuscript.

.

Paragraph P15 l4
- Should "RVM" be "pdf"? Corrected
- Sentence "Performing...", rephrase done
- "long enough to neglect the Monte Carlo uncertainty" Corrected
(P15 l15 Thought: it could have been possible to employ a more standard RCP scenario for future, defined based on the DeltaF, rather than on the change in CO2.) Done
P16 l14
"Instead of 100K", please make a clear link to what is said at the beginning of the section, where you say that 100K would be necessary to reduce the Monte Carlo uncertainty. Probably you could simply move the Paragraph on P15 l4 here. I make a clear link:
Due to computational constraints, confidence intervals are computed only for the stationary model. In addition, the simulation length has been reduced to 10K years (instead of 100K), which implies that part of the calculated uncertainty is due to Monte Carlo sampling variability.

P16 l14
I would start the sentence with "In order to investigate…" and then say that you separate the uncertainties. Rephrased as follows: "In order to investigate further the sources of this uncertainty, the uncertainty associated with the RVM only is separated from the uncertainty of the full model, i.e. of the joint pdf, by calculating confidence intervals with the same approach as described above, but using the same marginal pdfs in each bootstrap repetition."
Section 4.1,
- Please, make clear that you are referring to the mean return period when you say "stochastic set"
- I guess that you are simulating from the vine and then transforming the simulated uniform variables to the "real" via the marginal models containing the predictors/covariates. Here you fix the $CO_2$ predictors to certain values, but what value is given to the NAO predictors? Please, make this clearer. Also, please make this procedure clear in the method section.
NAO is simulated using a Gaussian distribution. I make this more clear in the methods section (3.4):
"Each year of the three stochastic sets above is associated with a random NAOI value that has been simulated assuming a Gaussian distribution, fitted to the historical NAOI dataset (see Figure 6). The influence of NAO on each one of these sets can thus be discerned by selecting only the simulated years with negative or positive NAOI values."
Fig10 The grey uncertainties are not well visible on my printed version. Please, check if this is only a problem of mine. I have darkened the shaded areas.
P20 l11 "originates", I would write "appears to originate" Corrected
Table 4 missing the unit: "years" Corrected
P21 l1 see the main comment on CO2 above Replied above
P21 l12 "return period of 1 in 39 years", is it not enough to write "return period of 39 years"? Also in other parts of the paper? Corrected
P21 l12 How is it a positive and negative phase defined? What is the value of NAO that is used to sample the data? In the figure it is written > or < 1, but do not you use a unique value, e.g., 1 and -1? Please, make this clear. I add the following sentence:
"Fig. 11 shows the RP curve of current climate wAFI, alongside with RP curves computed solely from simulated years with NAOI values greater than 1 (i.e. representing the positive NAO phase) or years with NAOI values lower 5 than 1 (i.e. representing the negative NAO phase)."

P23 l3. The predictor influence on the dependence is not considered. Indeed it is specified here only for the NAO, but not for the CO2. As mentioned above I have added a phrase in the methods section. I've also included CO2.

P24 l9 the occurrence has increased? The return period has increased. The same in the abstract. Corrected

P24 l16 "such extreme events", not clear which extreme events. Corrected

Best regards.

**2. Editor comments**

Please consider the following main issues a) to d) to be properly clarified:

a) There is an apparent inconsistency in your manuscript

Note the following. At lines 9-10 of the abstract and lines 9-10 of page 24 in the Conclusions you write that "The model suggests that the occurrence of such extreme cold events have increased approximately two times during the course of the 20th century as a result of anthropogenic climate change". Instead, at lines 24-25 you write that "the non-stationary model suggests that under current climate conditions, such an extreme event, is approximately 2 times less likely to occur than in the 1960s". Table 4 also suggests an increased (doubled) return time of the "1962/63 winter freeze event". Please correct this inconsistency or clarify. This is a typo – it's corrected

b) In the "Methods" section, defend your choice of using the change in radiative forcing from CO2 in the nonstationary model (first main comment of the reviewer) Please see above

c) In the methods section, add that the non-stationary model does not consider the potential non-stationarity in the dependencies described by the joint pdf (first part of the second comment by the reviewer) Done

d) It is not clear which variables the nonstationary model is actually using. Please, make this clear in the methods section (second part of the second main comment of the reviewer) Done

Moreover, please consider all minor points raised by the reviewer Done

Finally consider also the following comments:

i. I agree with the reviewer that at line 9 of the abstract ""such extreme cold events" is not immediately clear and the sentence should be rephrased (I presume it refers to events as anomalous as the "1962/63 winter freeze event) I have corrected the sentence: "the occurrence of extreme cold events such as the 1962/63 winter".

ii. Line 8-9 of the conclusion "According to the model, such a cold winter is estimated to occur once every approximately 400 years under current climate conditions in the UK." I see this in table 4 (please, in the text, specify that you refer to the South UK). However, this result contradicts the empirical evidence. You write at lines 20-22, page 20 that "only two other winters (1683/84 and 1739/40) have been colder than 1962/63 in the last 350 years", suggesting a return period in the range of 110-120 years. Please clarify this inconsistency.

The return period of 400 years corresponds to the current climate, i.e. corresponding to a present day (2018) concentration of CO2 (400 ppm). I delete the sentence in the conclusion to avoid confusion.

A direct comparison with the historical Central England Temperature (CET) record, can made in comparison to the stationary model. As shown in Table 4, the stationary stochastic set seems to overestimate the return period of this event (at least in comparison to the South UK).

I mention this in section 4.2.2 and I make it more clear "According to the latter, only two other winters (1683/84 and 1739/40) have been colder than 1962/63 in the last 350 years, suggesting a return period in the range of 110-120 years, as well. The stationary model overestimates this winters' return period which is estimated to 205 years across all the UK. Especially in the South of the UK the

model suggests that this event has been particularly unusual. In the Northern part of UK on the other hand, the model suggests a lower return period of 106 years, closer to the empirical estimate."

iii. The definition of AFI in expression (2) at page 5 is not clear. Specify the meaning of the subscript "i" and specify appropriately the upper and lower limit of the sum Both corrected.

iv. Consider whether moving the first part of section 3.3 (describing copulas) to an appendix could improve readability of your manuscript and help the readers to focus on the novelty of your study I agree and I moved this section in the Appendix.

[revised manuscript text omitted]

---

## Author Response (AR4)

Dear Editor,

Thank you for recommending my paper for publication in NHESS. I am also grateful for accepting the extension of the deadline for the delivery of my revised manuscript. Finally, I would like to thank you once again for your valuable review comments.

Please find attached the revised manuscript with the corrected references.

With best regards,

Symeon Koumoutsaris